# A dynamical process-based model AMmonia–CLIMate v1.0 (AMCLIM v1.0) for quantifying global agricultural ammonia emissions – Part 1: Land module for simulating emissions from synthetic fertilizer use

Jize Jiang[1,a], David S. Stevenson[1], and Mark A. Sutton[2]

[1]School of GeoSciences, The University of Edinburgh, Crew Building. Alexander Crum Brown Road, Edinburgh, EH9 3FF, UK
[2]UK Centre for Ecology and Hydrology, Edinburgh, Bush Estate, Midlothian, Penicuik, EH26 0QB, UK
[a]now at: Institute of Agricultural Sciences/Institute of Biogeochemistry and Pollutant Dynamics, ETH Zurich, 8092 Zurich, Switzerland
[a]now at: Eawag, Swiss Federal Institute of Aquatic Science and Technology, Ueberlandstrasse 133, 8600 Dübendorf, Switzerland

*Correspondence to*: Jize Jiang (jize.jiang@usys.ethz.ch/jize.jiang@ed.ac.uk)

**Abstract.** Ammonia ($NH_3$) emissions mainly originate from agricultural practices and can have multiple adverse impacts on the environment. With the substantial increase of synthetic fertilizer use over the past decades, volatilization of $NH_3$ has become a major loss of N applied to land. Since $NH_3$ can be strongly influenced by both environmental conditions and local management practices, a better estimate of $NH_3$ emissions from fertilizer use requires improved understanding of the relevant processes. This study describes a new process-based model, AMmonia–CLIMate (AMCLIM), for quantifying agricultural $NH_3$ emissions. More specifically, the present paper focuses on the development of a module (AMCLIM–Land) that is used for simulating $NH_3$ emissions from synthetic fertilizer use. (Other modules, together termed as AMCLIM-Livestock, simulate $NH_3$ emissions from agricultural livestock, are described in Part 2). AMCLIM–Land dynamically models the evolution of N species in soils by incorporating the effects of both environmental factors and management practices to determine the $NH_3$ emissions released from the land to the atmosphere. Based on simulations for 2010, $NH_3$ emissions resulting from the synthetic fertilizer use are estimated at 15.0 Tg N yr$^{-1}$, accounting for around 17 % of applied fertilizer N. Strong spatial and seasonal variations are found. Higher emissions typically occur in agricultural intensive countries (such as China, India, Pakistan and US), and mostly reach the maximum in the summer season. Volatilization rates indicate that hotter environments can result in more N lost due to $NH_3$ emissions and show how other factors including soil moisture and pH can greatly affect volatilization of $NH_3$. The AMCLIM model also allows estimation of how application techniques and fertilizer type have impacts on the $NH_3$ emissions, pointing to the importance of improving management practice to tackle nutrient loss and of appropriate data-gathering to record management practices internationally.

# 1 Introduction

Ammonia ($NH_3$) is the one of the most critical species in the nitrogen cycle. It is the primary reduced form of reactive nitrogen ($N_r$) and also the principle alkaline gas in the atmosphere, which can have significant impacts on the environment (Sutton et al., 2020). Ammonia released to the air can react with sulfuric and nitric acids to form particulate aerosols. These aerosols not only reduce air quality and visibility, but also have implications on the Earth's radiation balance (Butterbach-Bahl et al., 2011a). Due to its high solubility, $NH_3$ readily gets deposited on wet surfaces, such as water bodies or plant surfaces. The deposition may cause eutrophication of aquatic systems and damage to vegetation, subsequently harming local biodiversity and ecosystems (Dise et al., 2011). These negative environmental consequences also pose great threats to the human society.

The largest $NH_3$-emitting sector is agriculture, via the volatilization of $NH_3$ from fertilizer and animal excreta. Estimated agricultural $NH_3$ emissions have increased rapidly from over 20 Tg N $yr^{-1}$ in the 1970s to over 45 Tg N $yr^{-1}$ in the 2010s (EDGAR, 2023) as a result of agricultural intensification. For agricultural $NH_3$ emissions, application of synthetic fertilizer is a major source. Since the mid-20th century, the exponential increase in synthetic fertilizer use has led to a substantial rise in $NH_3$ emissions (Lu and Tian, 2017; Xu et al., 2019). On the one hand, the famous Haber-Bosch process revolutionized agriculture by significantly increasing crop yields with additional N input through fertilizer application, estimated to support about half of the global population (Erisman et al., 2008). On the other hand, the use of synthetic fertilizers also causes significant environmental impacts because a large portion of applied N is leaked to the environment instead of being absorbed by the crops, contributing to the environmental problems noted above. Emission of $NH_3$ is one of the major losses of N introduced to the agricultural systems, representing a crucial waste of reactive nitrogen resources (Sutton et al., 2021).

Improving our understanding of the $NH_3$ emissions from synthetic fertilizers is important not only for evaluating their environmental impacts and resource distributions, but also for developing effective mitigation measures, especially in the face of a changing climate and growing population, which can also bring economic benefits. Although studies have been conducted to quantify $NH_3$ emissions from synthetic fertilizers, most of these estimates rely on emission factors (EFs). Emission factors are empirically derived or experimental values that summarize $NH_3$ volatilization rates that vary by specific source sectors. Emission of $NH_3$ is then calculated by combining statistical activity data with reference EFs. The simplicity in calculations is an advantage. However, these EFs may not provide an accurate representation of $NH_3$ volatilization because $NH_3$ emissions are highly sensitive to environmental conditions, such as temperature, that show large spatial and temporal variations. For example, it has been suggested that a 5 °C rise in temperature will lead $NH_3$ volatilization potential almost to double, as predicted by solubility and thermodynamics (Sutton et al., 2013). As EFs only consider the climatic effects to a limited extent, using constant values to describe the fraction of N that volatilizes as $NH_3$ from different sources may not provide realistic estimates under all environmental conditions (Jiang et al., 2021). Significant uncertainties may result from using EFs in large-scale assessment (e.g., global-scale calculations), and these also typically lack information in

the temporal characteristics of NH$_3$ emissions, although seasonal empirical corrections are sometimes used (Hellsten et al., 2008).

Compared with EFs, process-based models are a useful tool used to estimate NH$_3$ emissions, which are developed based on the theoretical understanding of relevant processes (Sutton et al., 1995; Nemitz et al., 2001; Flechard et al., 2013; Móring et al., 2016). Process-based models attempt to include the effects of environmental factors on NH$_3$ volatilization to address the limitation that EFs lack a systematic response to environmental drivers. In recent decades, more effort has been put into developing process-based models to provide better estimates for the NH$_3$ emissions. Nemitz et al. (2001) developed a two-layer canopy compensation point model (2LCCPM) to investigate the bi-directional exchange of NH$_3$ between the atmosphere, the vegetation and the ground layer below. The 2LCCPM model incorporates the "compensation point" theory (Farquhar et al., 1980; Sutton et al., 1995), which introduces a theoretical equilibrium reached by the atmospheric NH$_3$ concentrations above a given source or sink of NH$_3$. This bi-directional exchange scheme has been adapted and modified for different purposes in later studies and models, such as estimating NH$_3$ emissions from fertilized agricultural land like CMAQ-EPIC (Bash et al., 2013; Fu et al., 2015; Chen et al., 2021) and DLEM (Xu et al., 2018, 2019) and for investigating NH$_3$ volatilization from urine patches in the GAG model (Moring et al., 2016). Pinder et al. (2004) constructed a model for simulating NH$_3$ emissions from dairy cattle with a range of activities included, such as housing, grazing, manure storage and application. The Flow of Agricultural Nitrogen (FAN) model, which simulates global agricultural NH$_3$ emissions, includes response to climatic factors and has detailed soil processes (Riddick et al., 2016; Vira et al., 2020).

Although process-based models are more sophisticated compared with EFs, they also require more inputs and computational resources than studies and inventories that use the EFs. Due to the difficulties in covering all processes in a reasonable level of detail, each model has its own focus and advantage, with different levels of complexity, inputs, processes and factors emphasized. It is critical to justify which processes are important and should be included. In this study, the development, evaluation and application of a new and complete model for quantifying global agricultural NH$_3$ emissions, AMmonia–CLIMate (AMCLIM) is described. This paper presents the module of the AMCLIM model that is used for simulating NH$_3$ emissions from synthetic fertilizer use ('Fertilizer simulations'), while the other components of the model for livestock farming ('Livestock simulations') are detailed in a forthcoming paper.

## 2 Method and materials

### 2.1 Overview of the AMCLIM model

The AMmonia–CLIMate model (AMCLIM) is a dynamical emission system that quantifies climate-dependent NH$_3$ emissions from agriculture, based on process-level understanding. AMCLIM version 1.0, as described here, incorporates the effects of environmental conditions on the formation and transport of N compounds to simulate the temporal evolution of various N species, especially NH$_3$. There are three modules in AMCLIM: a) Housing, b) Manure Management and c) Land,

as shown in Fig. 1. The model focuses on synthetic fertilizer application (present paper) and livestock farming, simulating relevant physical, chemical and biological processes that govern the N flows in agricultural systems, whilst also considering agricultural management practices.

The AMCLIM model can be operated at multiple scales; it is calibrated and validated at the site scale by comparing with measurements and experimental studies and then is applied on the global scale to provide estimates of global $NH_3$ emissions. For site-scale simulations, the AMCLIM model explores various source sectors and can be run with different length time-step, depending on inputs given in the measurement datasets. For global simulations, the AMCLIM model covers all major agricultural sectors and aims to provide a comprehensive assessment of agricultural $NH_3$ emissions. Livestock sectors include pigs, poultry, cattle, sheep and goats, which together dominate the livestock $NH_3$ emissions. An earlier version of the model for poultry has been reported by Jiang et al. (2021). The AMCLIM model also simulates $NH_3$ emission from global synthetic fertilizer use on crops. In this study, the development of the Land Module of AMCLIM (AMCLIM–Land) is described, which was used to simulate $NH_3$ emissions from synthetic fertilizer use. The Land module is also used elsewhere as part of AMCLIM-Manure Management, which considers emission from the full chain of animal housing, manure storage and land application of manure.

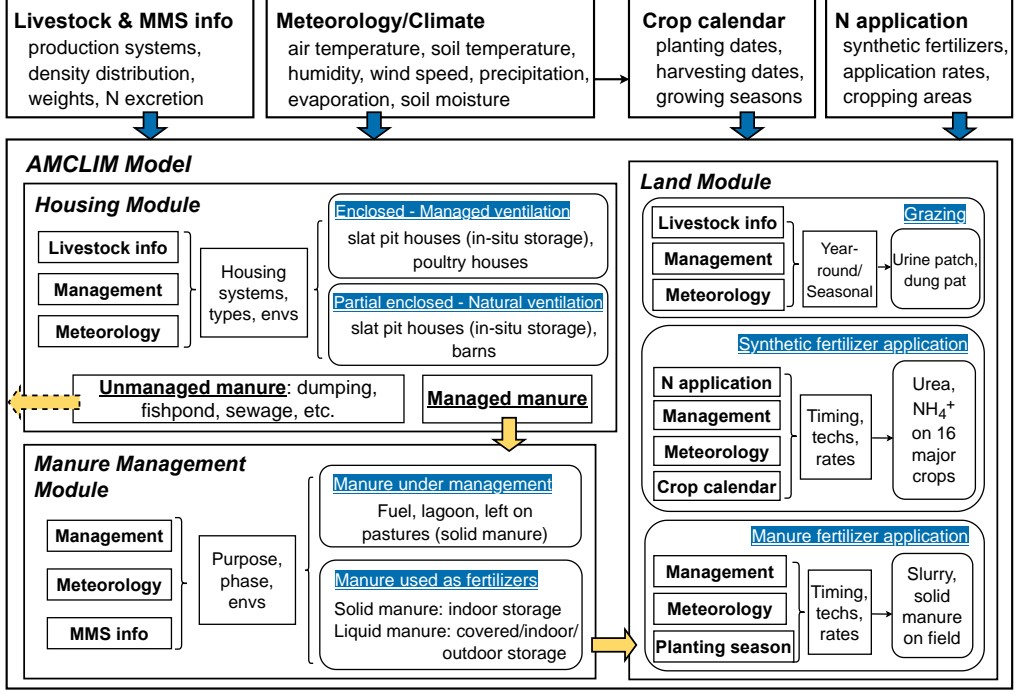

**Figure 1. Components and structure of the AMCLIM model and inputs (blue arrows) used for simulations. The dashed yellow arrows represent a fraction of unmanaged N from housing that is not simulated in the Manure Management Module (AMCLIM–**


MMS). **Solid yellow arrows represent the N flows between modules. (MMS: Manure Management System. Envs: Environments. Techs: Techniques.)**

## 2.2 Land Module of AMCLIM: AMCLIM–Land

### 2.2.1 Soil layering and simulated processes in AMCLIM–Land

AMCLIM–Land simulates $NH_3$ volatilization at the land surface and the evolution of N species in soils. AMCLIM–Land models all the crucial processes associated with fertilizer application, including $NH_3$ volatilization to the atmosphere, surface

runoff, nitrogen diffusion and leaching into deep soil, nitrification, hydrolysis of urea and plant N uptake, as illustrated in Fig. 2. While the main focus of the model is $NH_3$ emissions, these other processes are incorporated as they are relevant in determining the fate of applied nitrogen, which interacts with $NH_3$ emissions according to the mass balance.

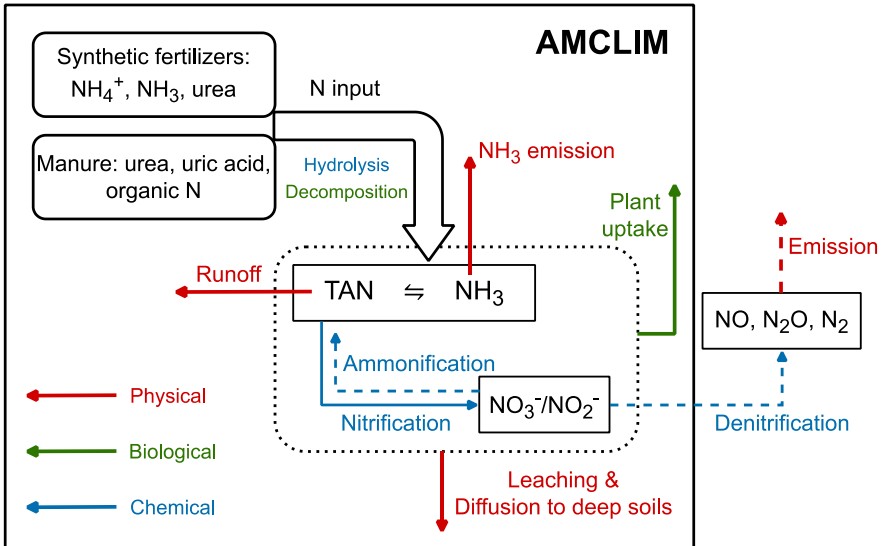

**Figure 2. Simulated N processes in AMCLIM–Land (shown in the solid black box). Ammonification of nitrate (or dissimilatory nitrate reduction to ammonium), denitrification and emission of NO, $N_2O$ and $N_2$ are not included in this study, but are shown (dashed arrows; or outside the black box) to illustrate the comprehensive concept. The dotted black box indicates soil N processes. Red arrows represent physical processes, including $NH_3$ volatilization, surface runoff, leaching and diffusion of nitrogen to deep soils. Green arrows represent biological processes, such as plant uptake of N and decomposition of organic N. Blue arrows**
**represent chemical transformations, including hydrolysis of urea and uric acid in animal excreta and nitrification.**

*Soil layers in AMCLIM–Land*

To simulate these processes, AMCLIM–Land defines four soil layers with a total depth of 28 cm, each with a specific thickness of 2 cm, 5 cm, 7 cm and 14 cm, respectively (Fig 3a). The upper two layers (0–2 cm, 2–7 cm) correspond to the

first soil layer defined by European Centre for Medium-Range Weather Forecasts Reanalysis v5 (ERA5) for which reanalysis data are used, while the lower two layers (7–14 cm, 14–28 cm) correspond to the second soil layer in ERA5 (Hersbach et al., 2020). By integrating these soil layers into its model, AMCLIM–Land can simulate the soil processes related to fertilizer applications at various depths.

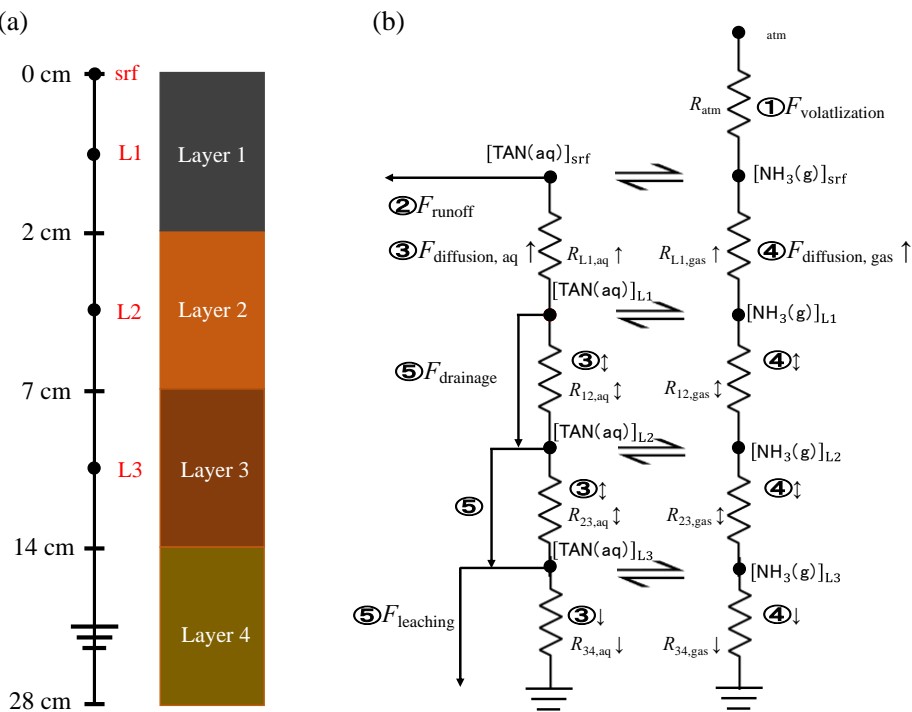

**Figure 3: Schematic view of the vertical soil layers and physical transport scheme for N species (showing TAN as an example) in the AMCLIM–Land. (a) Four soil layers in the soil column from the surface (0 cm) to a depth of 28 cm (Not scaled). (b) Physical transport of N species in soils and atmosphere. Processes include 1) NH₃ volatilization, 2) surface runoff, 3) aqueous diffusion, 4) gaseous diffusion and 5) drainage/leaching. The concentrations of aqueous TAN and gaseous NH₃ are the mean concentrations of each soil layer, represented by black dots. Soil resistances are shown between two black dots, with the numbering representing the**
**soil layers, i.e., $R_{12}$ is the soil resistance for diffusion from soil layer 1 to layer 2. Arrows represent the direction of diffusion which can be upwards, downwards or bi-directional. Transport distances for diffusion are the distance between the midpoints of two adjacent soil layers, e.g., 3.5 cm from soil layer 1 to layer 2, and 6 cm from soil layer 2 to layer 3. The concentrations of N species in the bottom soil layer are assumed to be 0, and downward fluxes take place from the above layer 3 to this layer.**

*Masses of N pools and concentrations*

    In the AMCLIM model, pools of N compounds are determined by the sources and losses. Flows between the pools (termed fluxes) are simulated, with the masses of pools calculated at every time step. The general expression for the time-dependent N pools can be simply written as:

$$\frac{\mathrm{d}M_{\mathrm{N}}}{\mathrm{d}t} = F_{\mathrm{P_N}} - F_{\mathrm{L_N}} , \qquad\qquad\qquad (1)$$

where $M_N$ is the mass of a N species (g N m$^{-2}$, given in per unit area; all masses have units of g m$^{-2}$ if not specifically explained). $F_{P_N}$ and $F_{L_N}$ represent the sum of production (including inputs) and losses of the N compound, respectively (all N flows have units of g N m$^{-2}$ s$^{-1}$ if not specifically explained).

The same mass balance approach is used to calculate the water pool in AMCLIM and which is used to solve the concentrations in the aqueous phase of each N species by dividing the mass by the volume of water, as follows:

$$[N] = \frac{M_N}{V_{H_2O}}, \tag{2}$$

where [N] (g mL$^{-1}$) is the concentration of the N species in water, and $V_{H_2O}$ (mL m$^{-2}$) is the volume of water. It must be noted that the sources and losses terms of each N pool and the water pool in different modules are not the same, and Eq. (2) is a general representation for the concentration calculations, which is modified when considering multi-phase equilibria.

*Calculation of soil TAN pool and partition*

The most important aggregated N species simulated in AMCLIM is total ammoniacal nitrogen (TAN = NH$_3$+ NH$_4^+$), which can either be partitioned into gaseous NH$_3$ ($M_{NH_3,g}$), aqueous TAN ($M_{TAN,aq}$) and adsorbed NH$_4^+$ ($M_{NH_4^+,s}$), as shown in Eq. (3):

$$M_{TAN} = M_{NH_3,g} + M_{TAN,aq} + M_{NH_4^+,s}. \tag{3}$$

The physical and biochemical processes involved in these anthropogenic activities influence the transformations between various N forms and the transport of N species, in which the abundance of TAN is the key component that governs the NH$_3$ emission potential ($\Gamma$) of a system (see Eq. 10).

In managed arable lands, the primary sources of the TAN pool are the application of N fertilizers, where ammonium is a direct source, while urea is a TAN input via hydrolysis. The TAN in soils can be depleted through multiple processes, such

as NH$_3$ volatilization, diffusion, nitrification etc. The time-dependent TAN pool ($M_{TAN}$; mass per unit area; all masses have units of g m$^{-2}$ if not specifically explained) is determined by the following equation, which includes the inputs and the depleting processes:

$$\frac{dM_{TAN}}{dt} = I_{TAN} + F_{TAN} - F_{NH_3} - F_{TAN\ runoff} - F_{diffusion} - F_{leaching} - F_{nitrif} - F_{uptake}, \tag{4}$$

where $I_{TAN}$ (g N m$^{-2}$ s$^{-1}$) represents direct input of TAN species from fertilization, such as ammonium, and $F_{TAN}$ is the TAN

production, i.e., through urea hydrolysis (see Section S1 in the Supplementary Materials). The remaining fluxes are removal processes as shown in Fig. 2 ($F_{NH_3}$ is the flux of NH$_3$ volatilization; $F_{TAN\ runoff}$ is the flux of surface TAN runoff; $F_{diffusion}$ is diffusive fluxes; $F_{leaching}$ is the flux of leaching; $F_{nitrif}$ is nitrification; $F_{uptake}$ is the flux of N uptake by plants/crops; all N fluxes/flows have units of g N m$^{-2}$ s$^{-1}$ if not specifically explained).

It should be noted that each soil layer may have different modelled processes. Equation 4 provides a general expression

for the TAN budget of the entire soil column. Nitrogen losses through volatilization and runoff occur at the land surface in the top soil layer. These losses are not included in deeper soil layers, where volatilization and surface runoff are absent.

Furthermore, it is assumed that there is no N uptake in the top soil layer (0–2 cm). In addition, diffusive and drainage fluxes considered as losses in the soil layer above become sources of nitrogen for the layer underneath. Fluxes are modified accordingly in each simulated soil layer, and detailed equations are presented in Section S1 in the Supplementary Materials.

The soil TAN pool is partitioned into three phases: gaseous $NH_3$ that exists in the air-filled porous space of soil (with its concentration expressed by $[NH_3(g)]$), aqueous TAN dissolved in the soil water (with its concentration expressed by $[TAN(aq)]$), and solid exchangeable $NH_4^+$ adsorbed onto solid particles (with its concentration expressed by $[NH_4^+](s)$, Riedo et al., 2002; Vira et al., 2020) as follows:

$$M_{TAN} = z \left( (\varepsilon - \theta)[NH_3(g)] + \theta[TAN(aq)] + (1 - \varepsilon)[NH_4^+(s)] \right), \tag{5}$$

where $\theta$ is the soil volumetric water content ($m^3$ $m^{-3}$ or m $m^{-1}$) and $\varepsilon$ is the porosity of soil (or the soil volumetric water content at saturation). The thickness of the soil layer is represented by $z$ (m).

Gaseous $NH_3$ is in equilibrium with aqueous TAN. The value of $[NH_3(g)]$ can be expressed as follows:

$$[NH_3(g)] = K_{NH_3} \cdot [TAN(aq)], \tag{6}$$

where $K_{NH_3}$ is a combined coefficient of Henry and dissociation equilibria as shown in Eqs. (9) and (10). Similarly, the

concentration of solid exchangeable $NH_4^+$ can be expressed by the following equation:

$$[NH_4^+(s)] = K_d \cdot [TAN(aq)], \tag{7}$$

where $K_d$ ($m^3$ $m^{-3}$) is the partition coefficient that represents soil adsorbed of TAN, which is dependent on soil properties (see Section S2 in the Supplementary Materials).

By combining Eqs. (5)–(7) , the concentration of aqueous TAN is given by:

$$[TAN(aq)] = \frac{M_{TAN}}{z(\theta + K_{NH_3}(\varepsilon - \theta) + K_d(1 - \varepsilon))}. \tag{8}$$

AMCLIM–Land also simulates other N species, including urea and nitrate, which have their own equations and processes that are detailed in Section S1 in the Supplementary Materials.

*Volatilization of $NH_3$*

Ammonia emission is a physiochemical process that typically takes place from wet or drying surfaces. Gas phase $NH_3$, held within the soil pore spaces (or excreta pores and at the surface in the slurry), is in dynamic equilibrium with aqueous ammonium depending on the substrate pH and temperature response of combined Henry and dissociation equilibria as follows (Nemitz et al., 2000):

$$\chi = \frac{161500}{T} \exp\left(\frac{-10378}{T}\right) \Gamma, \tag{9}$$

$$\Gamma = \frac{[NH_4^+]}{[H^+]} = \frac{[TAN]}{K_{NH_4^+} + [H^+]} \tag{10}$$

where $K_{NH4+}$ is the dissociation constant of ammonium ($NH_4^+$). and $\Gamma$ is the $NH_3$ emission potential defined as the ratio of $[NH_4^+]/[H^+]$  (Sutton et al., 1998; Nemitz et al., 2001).

The volatilization of $NH_3$ from the land surface to the atmosphere is driven by the concentration difference at two heights and is constrained by the atmospheric resistances, which is calculated as:

$$F_{NH_3} = \frac{[NH_3(g)]_{srf} - \chi_{atm}}{R_a + R_b},$$  (11)

where $[NH_3(g)]_{srf}$ and $\chi_{atm}$ are $NH_3$ concentrations at the surface and atmospheric $NH_3$ concentration at a reference height consistent with atmospheric resistances (typically 2 m). $R_a$ and $R_b$ are aerodynamic and boundary layer resistance, respectively. AMCLIM–Land simulates $NH_3$ volatilization as a uni-directional process, i.e., $NH_3$ flux is an emission only, and deposition is not simulated. There is no interaction with surface vegetation so that there is no surface resistance in Eq. (11). Although a substantial body of research has considered such bi-directional interactions (Sutton et al., 1995; Nemitz et al., 2000; Flechard et al., 2013), the present approach treats these as a separate modelling step for subsequent integration with atmospheric transport and deposition modelling (Sutton et al., 2013). The $NH_3$ concentration in Eq. (11), $[NH_3(g)]_{srf}$, is in equilibrium with the aqueous TAN concentration (as shown by Eq. (6)) at the surface, which is determined by the loss fluxes (i.e., $NH_3$ volatilization and surface runoff) from the surface and the upward diffusive fluxes from the soil underneath (details given in Section "*Physical transport of N in soils*" and Section S6 in the Supplementary Materials).

*Chemical transformations and biological processes of N in soils*

Nitrogen in soils occurs in several forms which are controlled by a range of chemical and biological processes. Nitrification and plant N uptake are crucial processes in AMCLIM–Land for simulating N dynamics in soils. Nitrification is the process by which $NH_4^+$ is converted to $NO_3^-$, which leads to depletion of the soil TAN pool. Nitrification is dependent on the abundance of $NH_4^+$, and its rate is influenced by various environmental factors, such as temperature, oxygen availability and substrate pH (Parton et al., 1996, 2001; Malhi and McGill, 1982; Bateman and Baggs, 2005; Gilmour, 1984; Norton and Stark, 2011). In AMCLIM–Land, the nitrification rate is calculated by scaling the optimum nitrification rate ($K_{nitrif,opt}$) by normalising factors that depend on temperature ($k_{nitrif,T}$), water-filled pore space ($k_{nitrif,WFPS}$) and pH ($k_{nitrif,pH}$) as follows:

$$K_{nitrif} = K_{nitrif,opt} \, k_{nitrif,T} \, k_{nitrif,WFPS} \, k_{nitrif,pH}.$$  (12)

The optimum nitrification rate and the representation of each dependence is presented in Section S3 in the Supplementary Materials.

The uptake of N by crops is a key biological process in AMCLIM–Land and can be used as a critical indicator for evaluating the fertilizer N use efficiency. However, simulating plant N uptake is complex and can be challenging; AMCLIM–Land uses a root uptake scheme derived from several studies (Riedo et al., 1998; Thornley, 1991; Thornley and Cannell, 1992; Thornley and Verberne, 1989). The scheme uses an integrated root activity parameter for N uptake ($\alpha_{root}$, g N m$^{-2}$ s$^{-1}$), the combined response factor for substrate C and N level ($J_{C,N}$), the effective available N pool for the plant, including $NH_4^+$ and $NO_3^-$ ($M_{Neff}$, g N m$^{-2}$), and the correction constant for root activity ($K_{Neff}$, g N m$^{-2}$), which is expressed by the following equation:

$$F_{\text{uptake}} = \frac{\alpha_{\text{root}}}{J_{\text{C,N}}} \frac{M_{\text{Neff}}}{M_{\text{Neff}} + K_{\text{Neff}}}, \tag{13}$$

The uptake of N by crops in AMCLIM–Land is mainly affected by temperature, and it is assumed to take place in soil layers beneath the top soil layer as explained in connection with Eq. (4). The growth of crops is represented by a set of empirical parameters reflecting the maturing state of roots. The C and N dynamics of crop growth are not modelled; constant values suggested by literature are used in this study, since the model focuses on the $NH_3$ emission process. Details are presented in Section S4 in the Supplementary Materials.

Immobilisation or microbial N uptake is a competing process against plant N uptake and is considered to be primarily regulated by available C in soils and gross ammonification (Butterbach-Bahl et al., 2011b). It remains uncertain that how microbial N uptake affects $NH_3$ emissions. Since AMCLIM does not simulate soil C dynamics, an explicit incorporation of microbial activities is beyond the scope of this study.

*Physical transport of N in soils*

The physical transport scheme of TAN and other N compounds (e.g., nitrate), is shown in Fig. 3b. Diffusion processes in soils are similar to the volatilization and are also driven by concentration gradients. AMCLIM simulates diffusion in both the aqueous and gaseous phases. Each phase is limited by soil resistances, which are functions of transport distance, molecular diffusivities and soil tortuosity factors (Móring et al., 2016; Vira et al., 2020). Detailed calculations are given in Section S5 in the Supplementary Materials. Diffusion is treated as a bi-directional process between soil layers 1 to 3, while diffusion is assumed to take place downward only from soil layer 3 to the bottom soil layer and upward only from soil layer 1 to the soil surface (Fig. 3b). It is worth noting that the surface concentrations used to calculate gaseous $NH_3$ fluxes and runoff fluxes from the surface are solved variables by assuming an equilibrium state. Upward diffusion from the first soil layer to the surface ($F_{\text{diffusion to surface}}$) is equal to the sum of $NH_3$ emission and runoff from the land surface to satisfy mass conservation (details given in Section S6 in the Supplementary Materials), as follows:

$$F_{\text{diffusion to surface}} = F_{\text{NH}_3} + F_{\text{N runoff}}. \tag{14}$$

The transport of N by movement of water includes leaching and runoff. Both processes are evaluated by multiplying the corresponding concentrations by the movement fluxes. For leaching ($F_{\text{leaching}}$), a percolation flux of water ($q_p$, m/s) is used for the calculation, as follows

$$F_{\text{leaching}} = q_p \cdot [\text{N(soil)}], \tag{15}$$

where [N(soil)] is the concentration of the N species in the soil. The loss via surface runoff ($F_{\text{N runoff}}$) is calculated similarly by the following equation:

$$F_{\text{N runoff}} = q_r \cdot [\text{N(sfc)}], \tag{16}$$

where $q_r$ (m s$^{-1}$) is the surface runoff flux of water, and [N(sfc)] is the surface concentration of the N species. It is worth noting that the surface runoff of N is assumed to ultimately enter water bodies and not to contribute to further $NH_3$ emissions.

Leaching of $NO_3^-$ occurs more frequently compared with $NH_4^+$ because $NO_3^-$ is highly mobile in soils, while $NH_4^+$ is
absorbed on the soil cation exchange complex so is less mobile (Butterbach-Bahl et al., 2011b). Annual $NH_4^+$ leaching is
usually less than 5 % of the total dissolved N in soil, but may have larger contributions in soils with heavy $NH_4^+$ loads (Dise
et al., 2009; de Vries et al., 2007). Nitrogen flows out from the simulated soil column are termed as "leaching", while N
fluxes that are transport between soil layers by water movement are termed as "drainage", as shown in Fig. 3b. The
percolation flux of water ($q_p$, in Eq. (15)) is the minimum between soil hydraulic conductivity ($K_s$, m s$^{-1}$) and water drainage
potential of a soil layer ($D_{pot}$, m s$^{-1}$), as expressed by the following equation:

$$q_p = \min(K_s, D_{pot}). \tag{17}$$

The soil hydraulic conductivity is related to the soil textures and soil water content. The water drainage potential is defined
as the excess amount of water beyond soil field capacity draining out from the soil layer. The calculation of $q_p$ is given in
Section S7 of the Supplementary Materials. In contrast, the surface runoff flux of water ($q_r$, in Eq. (16)) is not explicitly
modelled in AMCLIM–Land but is taken from the reanalysis meteorological dataset (Hersbach et al., 2018).

*Soil pH scheme in AMCLIM–Land*

Soil pH can greatly affect $NH_3$ emissions from fertilizer. Urea application is found to have strong impacts on soil pH as urea
hydrolysis consumes hydrogen ions ($H^+$) ($CO(NH_2)_2 + 2H_2O + H^+ \xrightarrow{\text{urease}} 2NH_4^+ + HCO_3^-$). Experimental studies found that
soil pH increased dramatically after urea application (including urine deposition), resulting in a peak in $NH_3$ emissions
(Chantigny et al., 2004; Curtin et al., 2020; Cabrera et al., 1991; Móring et al., 2016). Móring et al (2016) developed a
detailed chemistry scheme for soil pH dynamics in a urine patch and suggested that it is nevertheless suitable to use a fixed
pH for larger scale modelling as it is extremely difficult to explicitly simulate soil pH change on a larger scale, e.g., global
scale (Móring et al., 2017). AMCLIM–Land therefore uses an empirical relationship describing the soil pH after urea
application, which is developed based on Chantigny et al (2004) and Móring et al (2016). As shown by Fig. 4 and Eq. (18),
soil pH is assumed to reach a peak value of 8.5 within 24 to 48 hours after application, and then gradually recovers back to
the original values in the next 120 hours (Chantigny et al., 2004; Móring et al., 2016).

$$pH_{soil} = \begin{cases} \min\left(pH_{initial} + \frac{(8.5 - pH_{initial})}{24\,\text{h}}\Delta t_{app}, 8.5\right), \text{if } \Delta t_{app} \leq 48\,\text{h} \\ \max\left(8.5 - \frac{(8.5 - pH_{initial})}{120\,\text{h}}(\Delta t_{app} - 48\,\text{h}), pH_{initial}\right), \text{if } \Delta t_{app} > 48\,\text{h} \end{cases} \tag{18}$$

Due to limited experimental knowledge, for soils with higher pH than 8.5, pH values are assumed to remain unchanged at
the original values in AMCLIM–Land. Meanwhile, soil pH also does not change significantly for fertilizer applications other
than urea, e.g., $NH_4^+$, for which a database of soil pH is used in AMCLIM–Land (see Section 2.3.2). Although long-term
responses in soil pH can result from addition of different forms of N (including $NH_4^+$ and $NO_3^-$ salts) in relation to their
ultimate fate (plant uptake, storage, nitrification, leaching), such long-term effects are not considered in AMCLIM as having
a significant effect on $NH_3$ emission.

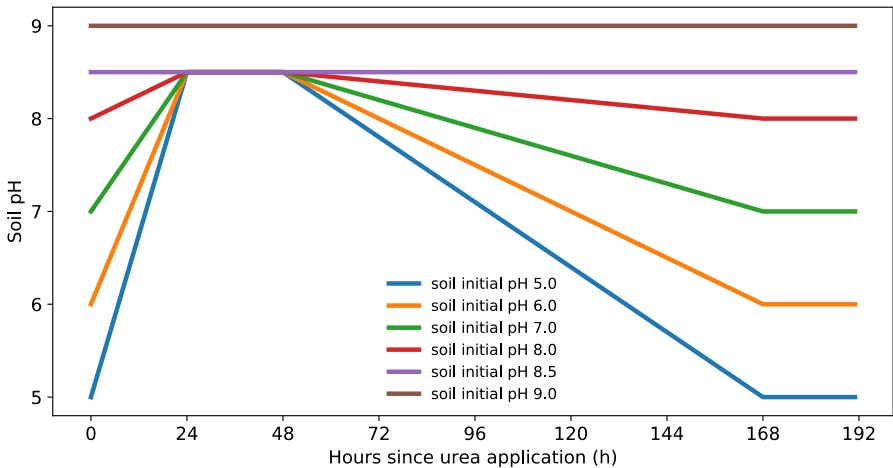


**Figure 4. Soil pH scheme used in AMCLIM–Land. Changes of soil pH for 192 hours (8 days) after urea application for soils with initial pH of six different values.**

### 2.2.2 Representations of human management in AMCLIM–Land

*Fertilizer application timing and techniques*

Nitrogen fertilizers are applied on cropland within a specific time period of the year depending on the climatic conditions, crop types and other environmental factors. AMCLIM–Land incorporates a crop calendar dataset which specifies the planting and harvesting seasons for major crops in both rain-fed and irrigated croplands (Jägermeyr et al., 2021). AMCLIM assumes a simple scheme for synthetic fertilizer application, where half the fertilizer is applied at the beginning of the

planting season and the other half in the middle of the growing period, i.e., midway between the dates of planting start and harvesting start.

AMCLIM–Land includes three techniques for fertilizer application: broadcasting, incorporation and deep placement. Different techniques refer to different locations where fertilizer N is applied, which can be reflected in AMCLIM–Land by distributing N input to the corresponding soil layer(s). "**Broadcasting**" is the easiest method and involves spreading

fertilizers broadly at the land surface, with N input added only to the first soil layer. "**Incorporation**" requires additional work to mix fertilizer deeper into the soil, and AMCLIM–Land assumes that applied N is well-mixed in the top two soil layers as a representation of immediate incorporation after fertilizer application. "**Deep placement**" is a less commonly used technique that involves burying fertilizer under the soil surface to mitigate nutrient loss to the air, and the model reflects this by adding N input to the third soil layer in simulations. By including this range of application techniques, the model can

simulate $NH_3$ emissions under different management that more closely reflect real-world situations and allows testing of potential mitigation measures. AMCLIM–Land does not simulate the impacts of fertilizer application on changing soil characteristics.

Water availability is a crucial factor that influences crop performance and determines local agricultural practices. In areas with adequate rainfall, natural precipitation is sufficient for crop growth, while in arid or semi-arid regions, additional water inputs are necessary for crop production. Croplands are classified into two categories: rain–fed and irrigated. For rain–fed croplands, it is assumed that there are no irrigation events, so soil moisture is represented using reanalysis soil moisture data, and the percolation flux ($q_P$) is represented by reanalysis subsurface runoff data. For irrigated croplands, irrigation is

assumed to occur after fertilizer application and when necessary. Consequently, the soil moisture of irrigated croplands needs to be re-estimated, as expressed by the following equation:

$$\theta_{irr,t} = \theta_{rea,t} + \Delta\theta, \tag{19}$$

where $\theta_{irr,t}$ and $\theta_{rea,t}$ represent the soil water content of irrigated croplands and the reanalysis soil water content data at time $t$, respectively. The reanalysis soil moisture data provide a reference value for "unperturbed" conditions, i.e., no irrigation. The

term $\Delta\theta$ represents an incremental change in soil moisture due to various to processes and activities, including irrigation ($w_{irr}$, m), percolation flux of water ($q_p$, m s$^{-1}$) and water uptake by crops ($W_{uptake}$, m s$^{-1}$):

$$\Delta\theta = \frac{w_{irr} - (q_p + W_{uptake})\Delta t}{z}, \tag{20}$$

where $\Delta t$ is the model time step. The amount of water applied during a single irrigation event equals the soil water content when the top two soil layers reach field capacity ($\theta_{fc}$). The water uptake by crops is described in Section S4 in the

Supplementary Materials. As mentioned, water is applied when the soil is too dry for crop growth. The threshold for initiating irrigation is when soil water content falls below the soil wilting point ($\theta_{wp}$), as expressed by the following equation:

$$w_{irr} = \begin{cases} \theta_{fc} \sum_{i=1}^{2} z_{\epsilon}, & \text{if } \theta_{irr,t} \leq \theta_{wp} \\ 0, & \text{if } \theta_{irr,t} > \theta_{wp} \end{cases}. \tag{21}$$

Irrigation is considered to have impacts on NH$_3$ emissions by influencing the leaching and altering the soil moisture. There are other processes affecting soil moisture such as evapotranspiration, which are considered to be implicitly included in the

reanalysis data. It is worth noting that the method is a simplified approximation for the soil moisture of irrigated croplands, and water uptake by plants is only simulated under this condition. A systematic simulation of soil moisture based on the underlying physics is beyond the scope of this study and is not considered in AMCLIM–Land.

## 2.3 Modelling NH$_3$ emissions from synthetic fertilizer use

**2.3.1 Application of AMCLIM–Land at site scale**

AMCLIM–Land was applied at site scale to simulate NH$_3$ emissions from a fertilized grassland. To evaluate the model, modelled emissions were compared with measurements from the GRAMINAE (GRassland AMmonia INteractions Across

Europe) experiment on $NH_3$ biosphere-atmosphere exchange conducted over intensively managed grassland in Braunschweig (52º 18´N, 10º 26´E), Germany (Sutton et al., 2009a,b). The GRAMINAE project measured $NH_3$ fluxes from managed grassland at three different stages: prior to cutting, post-cutting and after fertilization, using a combination of the aerodynamic gradient method (AGM) and relaxed eddy accumulation (REA). AMCLIM–Land was applied to simulate the $NH_3$ emissions during the third stage, in which ammonium nitrate fertilizer was broadcast onto the grassland at a rate of 100 kg N per hectare on 5 June 2000. The N input to AMCLIM–Land was then set to be 5 g $NH_4^+$–N per meter square which is equivalent to 50 kg $NH_4^+$–N per hectare for the simulation (because nitrate is assumed not to contribute to $NH_3$ emissions in the model). AMCLIM–Land was driven by meteorological variables, including air temperature, relative humidity, wind speed, precipitation and ground temperature. Soil properties and characteristics, including soil moisture, soil pH and soil textures, were also used as model input, all measured at the site by the GRAMINAE project. Measured atmospheric $NH_3$ concentrations interpolated to a reference height of 1 m were used as a reference when simulating emissions, with atmospheric resistances calculated from the measured meteorology. AMCLIM–Land was operated with a 15-min time step to match the frequency of meteorological inputs and the measured $NH_3$ fluxes. The GRAMINAE project provided the necessary level of detail for running AMCLIM–Land to simulate $NH_3$ emissions from fertilizer application, and additional information can be found in Sutton et al. (2009a,b). No irrigation event occurred after fertilizer application, so AMCLIM–Land used measured soil moisture data documented by the GRAMINAE dataset.

### 2.3.2 Global application of AMCLIM–Land: input and model setup

AMCLIM–Land was applied at the global scale to quantify $NH_3$ emissions from global synthetic fertilizer use. There were three major types of inputs: nitrogen application information, crop calendars and meteorological variables, as presented in Fig. 1. Nitrogen application data were obtained from the Global Gridded Crop Model Intercomparison Phase 3 (GGCMI3) dataset for 16 major crops, including synthetic fertilizer N application rates and total synthetic fertilizer N applied to crops (Mueller et al., 2012; Hurtt et al., 2020). The GGCMI3 datasets reported data for years from 1850 to 2015. Data for years after 2015 were then extended by a linear interpolation using data for the most recent 10 years (2005 to 2015). Data for 2010 and 2018 were used in this study. A time-series for global fertilizer use of the 21$^{st}$ century is given by Fig. A1 in the Appendix, with information on the simulated crops. The areas of croplands that use synthetic fertilizers were derived from GGCMI3, which have incorporated the harvested area from the Farming the Planet 2 (FTP2) dataset (Monfreda et al., 2008).

Nitrogen fertilizers include several types, such as urea, ammonium nitrate and ammonium phosphate etc. AMCLIM–Land considers three groups of applied N: ammonium N, urea N and nitrate N. Ammonium N directly enters the soil TAN pool and is treated as not affecting soil pH, while urea hydrolysis is treated as causing a temporary rise of soil pH after application. Nitrate is treated in AMCLIM as not contributing to $NH_3$ emission (Section 2.2.1). AMCLIM–Land combines the GGCMI3 nitrogen application data with country-level synthetic fertilizer consumption statistical data provided by the

International Fertilizer Association (IFA, 2021) to split the application rates into fractions of the three groups of applied N (see Section S8 in the Supplementary Materials). The area of cropland that uses a specific type of fertilizer is proportional to the fraction of the fertilizer used.

The crop calendars used in AMCLIM–Land were also obtained from the GGCMI3 dataset, which distinguish the planting and harvesting seasons of crops in rain-fed and irrigated systems. These calendars were used to determine the timing of fertilizer application, and each crop has a specific calendar that varies geographically. It should be noted that these crop calendars are based on climatology and therefore do not vary with years.

The hourly meteorological inputs for AMCLIM were from the ERA5 collection (Hersbach et al., 2018) and include air temperature, relative humidity (derived from dew point temperature), wind speed, rainfall, soil temperature and water content at 2 levels (0–7 cm, 7–28 cm) and runoff fluxes. Soil data inputs such as soil pH, soil texture (sand, clay and silt fraction) and soil organic matter content were obtained from the Regridded Harmonized World Soil Database (HWSD) v1.2 (FAO and IIASA, 2012; Wieder et al., 2014). The GRIPC dataset was used to classify cropland into rain–fed and irrigated systems and to determine the irrigation events and corresponding crop calendars.

For global simulations, AMCLIM–Land was applied using a longitude-latitude grid at a resolution of $0.5° \times 0.5°$. All model inputs were regridded to the model resolution if necessary. The simulations were performed at an hourly time step, and the prognostic variables at each time step were solved by the Euler method in the model. AMCLIM–Land was set up to use a one-year spin-up in order to keep an annual cycle of simulation period for each grid (as fertilization that takes place in November or December may result in $NH_3$ emissions in the following year), and was run for three rounds in which three application techniques were simulated independently and were assumed not to interact with each other. Each round was comprised of 32 full-year simulations, with urea and ammonium N run separately for the 16 major crops (i.e., one year of simulation for two types of N fertilizer applied to 16 crops). The total $NH_3$ emission from fertilizer application is calculated using the following equation:

$$F_{NH_3} = \sum_{i=1}^{3} f_{tech(i)} \sum_{j=1}^{2} f_{fert(j)} \sum_{n=1}^{16} F_{NH_3(i,j,n)}, \tag{22}$$

where $F_{NH_3(i,j,n)}$ is the component $NH_3$ emission from $n$ crop with fertilizer type $j$ by using application technique $i$, and $f_{tech(i)}$ and $f_{fert(j)}$ are the fraction of the application technique and fertilizer type used in a grid, respectively. The assumption in AMCLIM–Land is that the fraction of a specific fertilizer application technique used is related to the country income level, with higher income countries assumed to use more incorporation and deep placement compared with lower income countries. The income classification is provided by World Bank statistics (WB, 2022). The details are presented in Table A1 in the Appendix.

**3 Results**

**3.1 Site simulations for NH$_3$ emissions from synthetic fertilizer application**

Figures 5 shows the results of simulated NH$_3$ emissions over 10 days from the fertilized post-cutting grassland (the GRAMINAE campaign site), along with comparisons with measurements. Meteorological conditions are also given in Fig. 5, which shows that the ground temperature at the study site varied between 10 and 25 °C, with three particularly hot days on 9, 10 and 13 June. It is notable that ground temperature, relative humidity (RH) and friction velocity (which is dependent on wind speed and atmospheric stability) showed large diurnal variations. During the daytime, ground temperature and friction velocity were high while RH was low, with the opposite occurring at night time. Atmospheric resistances (including aerodynamic and boundary layer resistance) are inversely related to the friction velocity, of which values were small during the day and much larger at night. A few precipitation events occurred during the study period, with the largest rainfall occurring on 10 June (Fig. 5c). Soil water content was measured every two days at two depths of 0.15 and 0.30 m. The grassland was watered prior to the fertilization on 15 to 17 May, and there was no irrigation between 5 to 15 June. Therefore, subsurface runoff fluxes retrieved from the ERA5 reanalysis data were used as the percolation fluxes to determine drainage and leaching, ranging from 0.60 to 0.75 mm d$^{-1}$ in the simulated days. The shallower layer had lower soil water content than deeper soils, and moisture levels at both depths decreased from 12 % to 10 % in the 10 days following fertilizer application. The extent to which rainfall affected soil water content was uncertain based on the available measurements.

Ammonium nitrate fertilizer was applied on the grassland by broadcasting at 5:00 – 6:00 am on 5 June, 2000. Emissions of NH$_3$ occurred immediately after the fertilization, with maximum values of over 3000 ng m$^{-2}$ s$^{-1}$ observed on the same day. The measured emissions then gradually decreased over the following days, but showed strong diurnal variability, peaking around the midday, when temperature and friction velocity were high, and declining to a minimum at night when these variables were low. By 7 June, the third day after fertilization, the highest emission was 1500 ng m$^{-2}$ s$^{-1}$, which was only half of the peak value observed on the first day. The NH$_3$ emissions increased substantially on 8 June relative to the previous day (7 June) and declined again. From 12 June, the measured emissions were generally less than 500 ng m$^{-2}$ s$^{-1}$, which was significantly lower compared with the first week.

Figure 5d demonstrates that the AMCLIM model is capable of capturing the predominant features of the measured NH$_3$ emissions throughout the simulated period and producing estimates for daily NH$_3$ emissions and sub-hourly fluctuations comparable with the measurements. However, there are some differences between modelled and measured NH$_3$ fluxes, particularly on the first day and during night time simulations. Simulated emissions are higher than measurements in the afternoon and evening of the first day and night time on 6, 7, 9 and 10 June. Meanwhile, the highest measured emissions on 8 June are over 2500 ng m$^{-2}$ s$^{-1}$, but AMCLIM is unable to replicate these values and underestimates the peak emissions by about 40%. It should be noted that particularly large standard errors (shown as shaded grey area) also exist in measured NH$_3$ fluxes during 8–10 June, which is likely due to instrument uncertainties (Sutton et al., 2009b). Overall, AMCLIM

overestimates cumulative NH₃ emissions by 50 % from 5 June to 15 June (when there are available measurements). The modelled cumulative NH₃ flux is 0.49 g m⁻² compared with 0.32±0.07 g m⁻² by the measurements (Sutton et al., 2009b). Other diagnostic variables are shown in Section S9 in the Supplementary Materials.

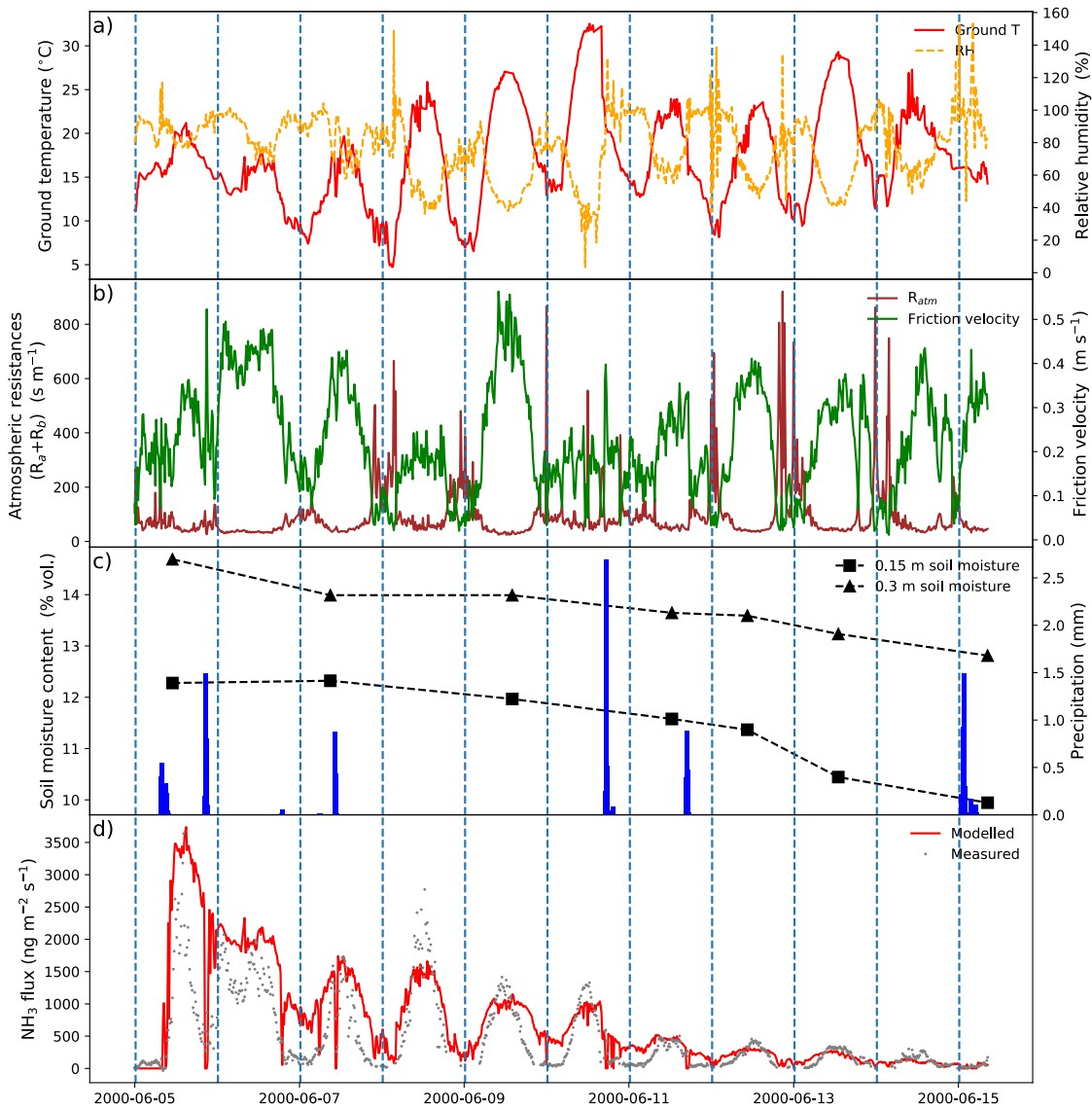

**Figure 5. Meteorological variables measured by GRAMINAE and site simulations for NH₃ emissions from a post-cutting grassland after fertilization in Braunschweig, Germany, from 5 June 2000 to 15 June 2000 by AMCLIM–Land. (a) Surface temperature and relative humidity. (b) Atmospheric resistances and friction velocity. (c) Soil volumetric water content at 0.15 m and 0.30 m depth and precipitation. (d) Modelled and measured NH₃ emissions.**

To evaluate the performance of the model against the measurements, AMCLIM–Land operated 12 runs with varying model parameters, variables and processes with different levels of complexity. In general, there were four groups of simulations, with correlation coefficients ("r" value) of each run and measurements, standard deviations normalized to measurements and normalized root mean square error (NRMSE) shown in Fig. 6. The base run provides the "best" fitting as the closest point to the measurement. All runs show similar correlation coefficients, ranging between 0.7 to 0.85, which

demonstrates the model's robustness as over 50 % of the variability in the measured $NH_3$ flux can be explained by the model. Varying the thickness of the top soil layer leads to the largest changes in standard deviation and the NRMSE (circles in Fig. 6). Changing z1 to 1 cm results in much larger NRMSE than the base value of 2 cm, and also overestimates the variability of the measurement. In contrast, simulations using different atmospheric $NH_3$ concentrations at 1 m does not show significant changes in NRMSE (triangles in Fig. 6), and the standard deviations of these simulations are close to those of the

measurement. A range of tortuosity correction was tested (stars in Fig. 6). Lowering tortuosity (no tortuosity and $j=6$) results in large increase in NRMSE compared with the base run using $j$ of 8.5, with an overestimation of the variations of measured fluxes. By comparison, higher tortuosity ($j=10$) leads to comparable NRMSE but much smaller standard deviations. AMCLIM–Land was also run by switching off several N processes, including drainage flux to the soil layer underneath, surface runoff and nitrification ("P"s in Fig. 6). Excluding the drainage of N in the model results in larger NRMSE than the

base run, while excluding runoff or nitrification only leads to small change.

     Based on the comparison with the GRAMINAE measurements, AMCLIM–Land provided an overall reasonable estimate for the $NH_3$ emission from a fertilized field and generally captured the variations of $NH_3$ at a high temporal resolution. The Taylor plot (Fig. 6) shows that the base model set up produces the best fitting to the measurement compared with several model runs with varying parameters. Moreover, assuming a zero background $NH_3$ concentration for global simulations is

justified as only limited impacts were found on the overall model performance. The model performance at various temporal resolutions was tested, and an hourly time step was found to be acceptable for global simulations given the modelling results and computational costs. More details for testing the model temporal resolution are given in Section S10 in the Supplementary Materials. AMCLIM–Land was then applied at the global scale and the results are presented in the following sections.

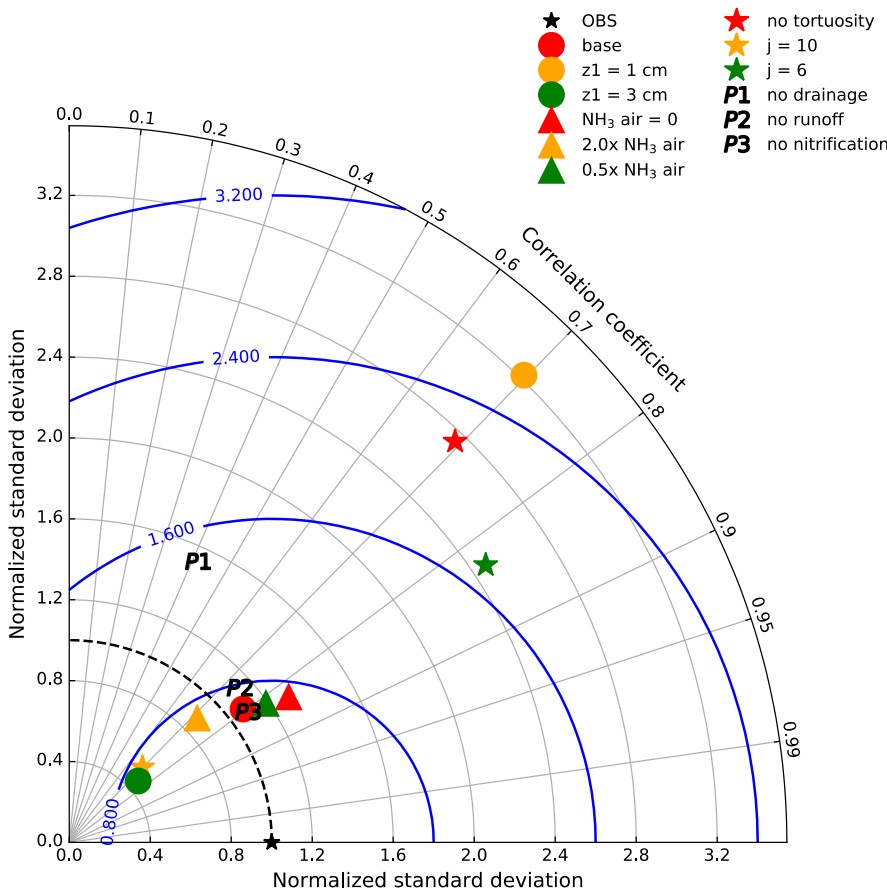

**Figure 6. A Taylor plot of correlation of simulated and measured NH$_3$ emissions by GRAMINAE, and normalized standard deviation of the model and measurements for four groups of model runs. Circles: base run (red); top soil layer thickness z1 = 1 cm (orange); z1 = 3 cm (green). Triangles: atmospheric NH$_3$ concentration set to 0 (red); 2.0x measured NH$_3$ concentration (orange); 0.5x measured NH$_3$ concentration (green). Stars: no soil tortuosity correction for diffusion (red); tortuosity correction factor $j$=1.0 (orange); $j$=0.65 (green). P1–P3 represent simulations without drainage, surface runoff and nitrification, respectively. The blue contours represent the root mean square error normalized by measurements (NRMSE).**

## 3.2 Global NH$_3$ emissions from synthetic fertilizer use

According to simulations using AMCLIM–Land, the global NH$_3$ emissions from synthetic fertilizer use are 15.0 Tg N yr[-1] in 2010 and 16.8 Tg N yr[-1] in 2018. The use of synthetic fertilizer increases from 102.3 Tg N yr[-1] to 120.5 Tg N yr[-1] during this period. The overall volatilization rates, which represent the percentage of applied N in ammonium and urea fertilizer that volatilizes as NH$_3$, are 17.2 % in 2010 and 16.7 % in 2018, respectively. Additional details about the use of three types of fertilizer are summarised in Table 1.

**Table 1. Use of three types of synthetic fertilizers, corresponding NH$_3$ emissions and percentage of volatilization ($P_V$) simulated by AMCLIM–Land in 2010 and 2018. Data for synthetic fertilizer use are derived from GGCMI3 and IFA. [a]Nitrate fertilizer is assumed not to contribute to NH$_3$ emissions in AMCLIM–Land. [b]Percentage of volatilization when not including nitrate fertilizers. [c]Percentage of volatilization when including nitrate fertilizers. [x]Percentage of volatilization from ammonium input (rather than of total N applied)**

| Year | | Ammonium | Urea | Nitrate[a] | Total |
|---|---|---|---|---|---|
| 2010 | Fertilizer use (Tg N yr$^{-1}$) | 31.9 | 55.2 | 15.1 | 102.3 |
| | NH$_3$ emission (Tg N yr$^{-1}$) | 6.2 | 8.9 | -- | 15.0 |
| | $P_v$ (%) | 19.3[x] | 16.1 | -- | 17.2[b] (14.6[c]) |
| 2018 | Fertilizer use (Tg N yr$^{-1}$) | 39.8 | 61.3 | 19.5 | 120.5 |
| | NH$_3$ emission (Tg N yr$^{-1}$) | 7.2 | 9.6 | -- | 16.8 |

The geographical distributions of NH$_3$ emissions and the volatilization rates for 2010 and 2018 are shown in Fig. 7 and Fig. A2 (see Appendix). The spatial patterns are similar for both years, with the highest emissions occurring in some parts of South Asia (mainly India and Pakistan), the Northern China Plain (NCP) and north-eastern China, and mid US and southern Canada. Regions including Europe, the Middle East and South America also had high emissions in some countries such as France, Spain, Turkey, and Argentina.

For the volatilization rates ($P_V$), the highest rates are found in eastern Africa (e.g., Kenya, Ethiopia and Somalia), southern Africa (e.g., Namibia), part of East Asia (e.g., Mongolia and northern China), exceeding 50 %. High $P_V$ values are also found in several regions in western US, the southern part of South America, the Sahel region, Ukraine, southwestern Russia and western Australia. It should be noted that regions with high volatilization rates do not always coincide with high emissions. Countries with high emissions often have moderate $P_V$ rates. In particular, the NCP and northeastern China show $P_V$ values of around 20 %, with high volatilization hot spots. The estimated volatilization rates of India are approximately 24 %, while some regions in the middle part of India and southern India show higher $P_V$ values. In most parts of Europe, estimated $P_V$ rates range from low to moderate (6 % to 18 %). However, both relatively high emissions and high volatilization rates are observed in Argentina and mid US.

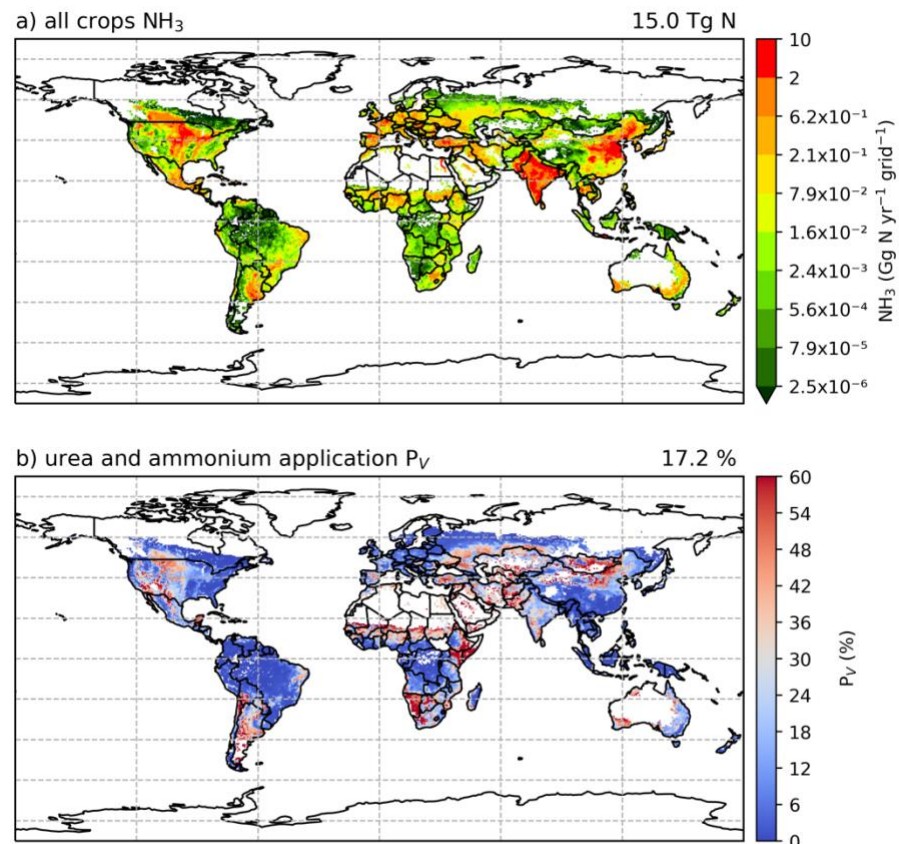

**Figure 7. Simulated (a) annual global NH₃ emissions (Gg N yr⁻¹ grid⁻¹) from synthetic fertilizer use in 2010. The colour bar represents 5th, 15th, 25th, 35th, 50th, 65th, 75th, 85th, 95th and 99th percentile of NH₃ emissions from synthetic fertilizer application in 2010. (b) Percentage of applied N in synthetic fertilizers (urea and ammonium fertilizers) that volatilizes ($P_V$) as NH₃ in 2010. The resolution is 0.5° × 0.5°. Maps of global fertilizer use in 2010 are shown in Figure S1 (see Section S8 in the Supplementary Materials.**

Figures 8 and A3 show the NH₃ emission from ammonium and urea fertilizer and the corresponding volatilization rates. In both 2010 and 2018, urea application results in more emissions than ammonium application due to its widespread use, although the emissions from each type of fertilizer are comparable. In 2010, about 40 % of emissions are from the use of ammonium and 60 % are from urea, and the relative contribution of ammonium to NH₃ emissions increases to 43 % in 2018. The volatilization rates of both fertilizers are similar, with ammonium application resulting in slightly higher volatilization rates. The overall volatilization rate from ammonium application decreases from 19.3 % in 2010 to 18.1 % in 2018, while the rate for urea also decreases from 16.1 % in 2010 to 15.7 % in 2018. These differences are directly linked to meteorological differences between these years.

As shown in Fig. 8, ammonium application in 2010 shows higher volatilization rates than urea application in most of the regions, especially in Argentina, mid US, the Middle East (Iran and Turkey), South Asia (Pakistan, note that there was no urea application in Pakistan in 2010 according to IFA), while urea application shows higher volatilization than ammonium in northern China, Mongolia and Ukraine (Fig. 8b and Fig. 8d). In 2018, the spatial variations of the volatilization rates for both fertilizers are very similar (Fig. A3b and Fig. A3d).

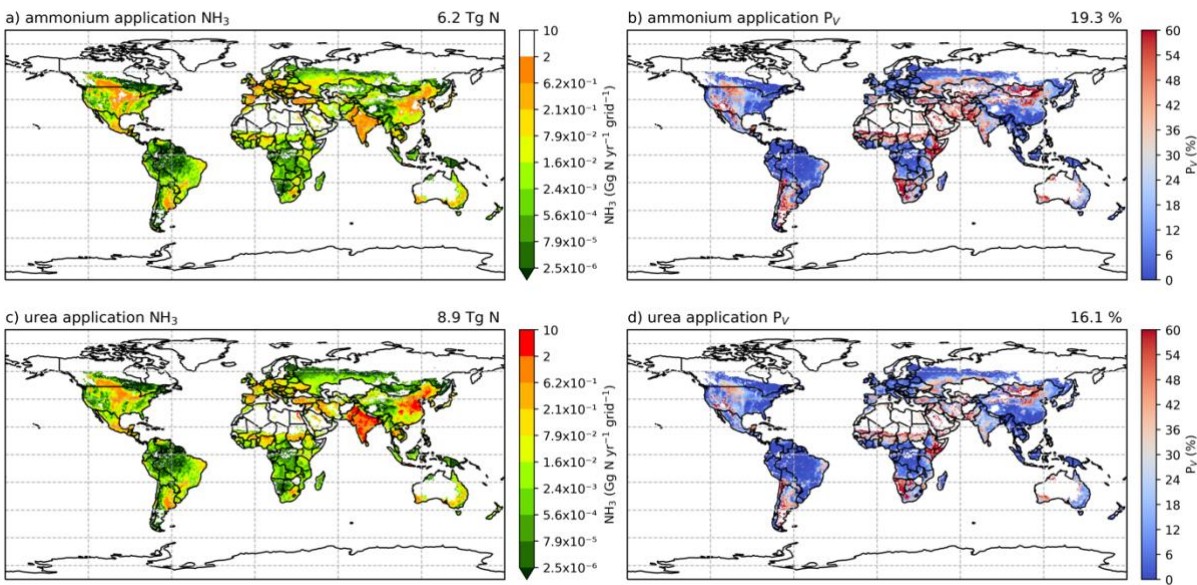

**Figure 8. Simulated NH$_3$ emissions (Gg N yr$^{-1}$) from two main types of fertilizers (global maps on the left) and the corresponding volatilization rates ($P_V$) in 2010 (global maps on the right). Ammonia emissions from (a) ammonium application and (c) urea application. Percentage of applied N that volatilizes as NH$_3$ from (b) ammonium application and (d) urea.**

Simulated NH$_3$ emissions from individual crops are shown in Fig. A7 and A8 (see Appendix). Among the 16 major crops, wheat, maize and rice are the top three emitter crops. The NH$_3$ emissions associated with wheat are the largest, with NH$_3$ increasing from 4.6 Tg N yr$^{-1}$ 2010 to 5.3 Tg N yr$^{-1}$ in 2018. Maize contributes to 2.9 and 3.2 Tg N yr$^{-1}$ in 2010 and 2018, respectively, and emissions from rice increase from 2.4 Tg N yr$^{-1}$ in 2010 to 2.5 Tg N yr$^{-1}$ in 2018. For the other crops, emissions range from 41.9 (rye) to 843.4 (cotton) Gg N yr$^{-1}$ in 2010, and from 45.4 (rye) to 1116.3 (cotton) Gg N yr$^{-1}$ in 2018.

The total global NH$_3$ emissions from synthetic fertilizer use in 2018 are generally higher than 2010 as a result of increasing synthetic fertilizer, and most of crops have higher emissions in 2018 compared to 2010. Cotton and groundnut, in particular, have a 32 % increase in NH$_3$ emissions, which are the topmost increase over time among these crops. By comparison, rapeseed is the only crop of which emissions decreased, with around 3 % less NH$_3$ emitted in 2018 than in 2010.

Figures A9 and A10 show the component $NH_3$ emissions from fertilization by different techniques for the two years. Broadcasting is responsible for more than 90 % of the estimated $NH_3$ emission, whereas incorporation and deep placement together contribute less than 10 % of the estimated global emissions (cf. Table A1 for assumptions). The geographical distributions of the volatilization rates for broadcasting are consistent with the global totals, given that broadcasting is the primary method used in fertilizer applications (Riddick et al., 2016). By comparison, incorporation and deep placement

result in lower volatilization rates. Specifically, incorporation reduces simulated emissions by more than 50 % based on the $P_V$ rates, while deep placement could potentially reduce emissions by almost 98 % (Fig. A9 and A10), although this reduction needs to be further investigated as it may be an overestimation. Regions with high volatilization rates for broadcasting also have high rates even when fertilizers were assumed to be incorporated into the soils in the simulations, such as Argentina, northern China, Mongolia, Namibia and mid US (Fig. A9b and A9d; Fig. A10b and A10d).

The fate of applied N in fertilizers for 2010 and 2018 are shown in Fig. 9. For both years, N uptake by crops is the largest among all processes, equivalent to 46.6 % (40.6 Tg N yr$^{-1}$) and 45.1 % (45.6 Tg N yr$^{-1}$) of fertilizer N applied in 2010 and 2018, respectively. Surface runoff is responsible for the smallest estimated N loss, which is only 3.6 % (3.1 Tg N yr$^{-1}$) in 2010 and 2.4 % (2.4 Tg N yr$^{-1}$) in 2018. The amounts of N losses (in the form of ammonium and urea) due to volatilization, nitrification and dissolved in soils through diffusion and leaching are comparable. In 2010, around 16.8 % (14.6 Tg N yr$^{-1}$) of

N is estimated to undergo nitrification, with 14.5 % (12.6 Tg N yr$^{-1}$) transferred to deeper soils. Nitrification is estimated at 16.2 Tg N yr$^{-1}$ in 2018, accounting for 16.2 % of the total pathways, which is similar to 2010. The diffusive fluxes and leaching in 2018 are approximately 50 % higher than the 2010 values (12.6 Tg N yr$^{-1}$), which together account for 11.0 % (18.5 Tg N yr$^{-1}$) of the N in applied synthetic fertilizers.

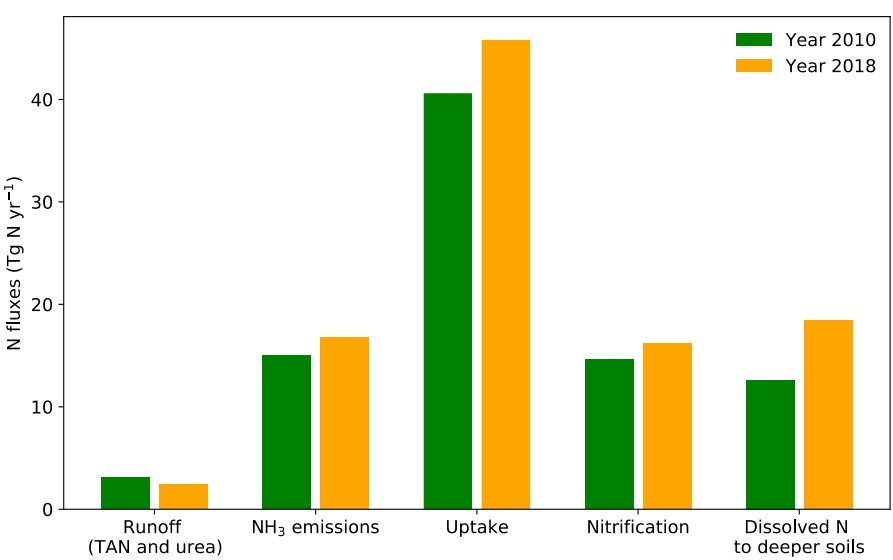


**Figure 9. The fate of N of ammonium and urea application in 2010 and 2018 simulated by AMCLIM–Land. Note that the runoff only includes surface runoff of TAN and urea, while nitrate runoff is excluded. "Uptake" refers to plant uptake of N.**

### 3.3 Seasonal and regional NH$_3$ emissions from synthetic fertilizer use

As NH$_3$ emissions are greatly influenced by climatic conditions and local management, NH$_3$ emissions exhibit strong seasonality that varies across the globe. Figures 10 and A4 (see Appendix) show the seasonal NH$_3$ emissions from fertilizer applications for 2010 and 2018, respectively. The seasonal emissions in both years are similar, with over 50 % of NH$_3$ occurring in the Northern Hemisphere (NH) summer months and about 25 % in March-April-May (MAM). September-October-November (SON) and December-January-February (DJF) both contribute slightly over 10 % of the annual

emissions. In the NH, more than 70 % of annual emissions are from June-July-August (JJA), while emissions in SON and DJF are significant in the Southern Hemisphere (SH). For example, Brazil and central African countries have predominantly SON emissions, while Argentina and southern Africa have the emissions largely occurred in JJA. Countries with high annual NH$_3$ emissions such as China, India and US generally show similar seasonal patterns, with the highest emissions occurring in JJA and lower emissions in other months.


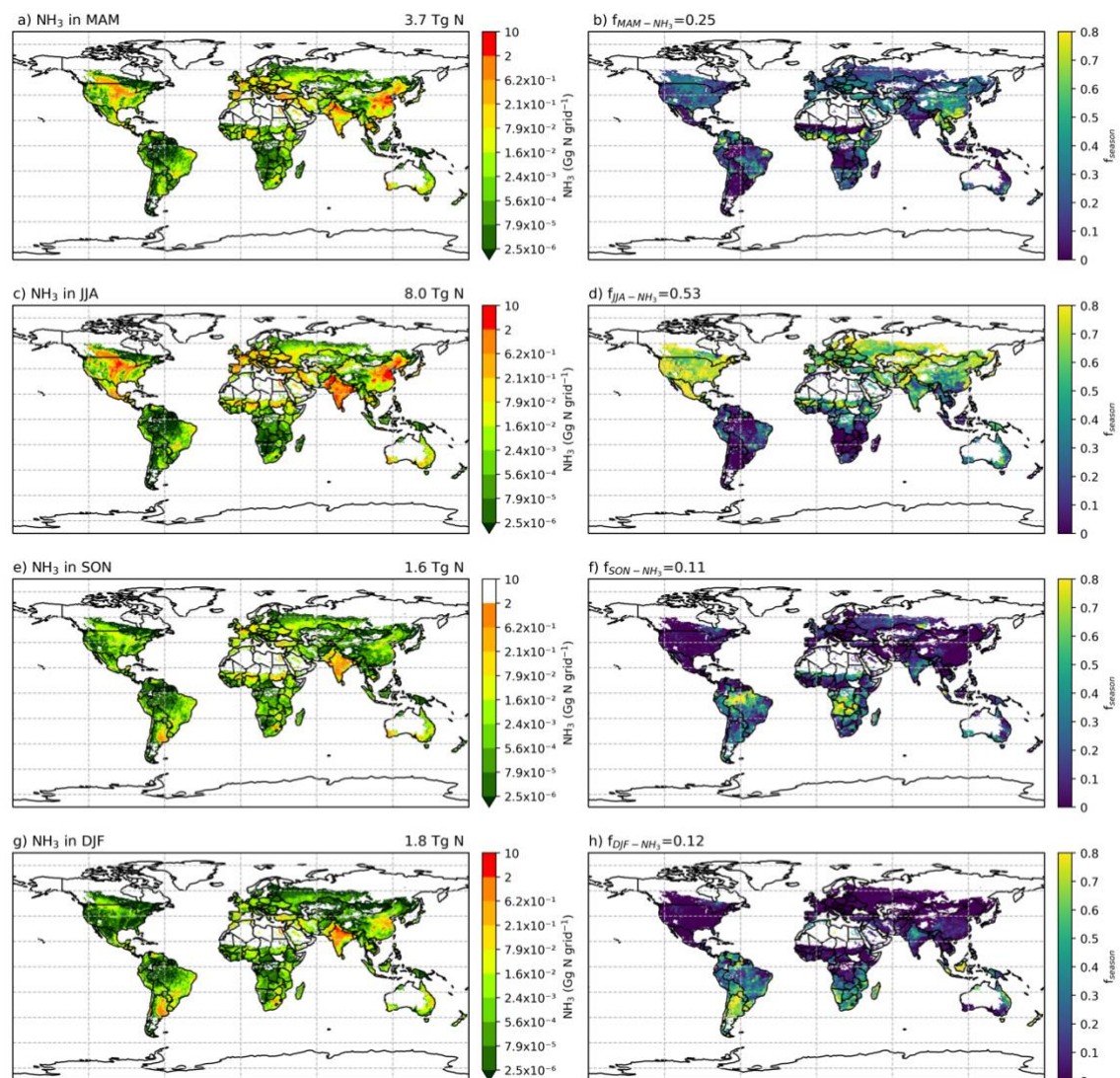

**Figure 10. Seasonal NH₃ emissions (Gg N grid⁻¹) from ammonium and urea fertilizer application (global maps on the left) and the relative fraction of annual emissions that are from the corresponding season ($f_{season}$) in 2010 simulated by AMCLIM–Land (global maps on the right). Ammonia emissions in (a) March, April and May (MAM), (c) June, July and August (JJA) (e) September, October and November (SON), and (g) December, January and February (DJF). Percentage of annual emissions in the season of (b) MAM, (d) JJA, (f) SON and (h) DJF.**

Global monthly emissions of NH₃ from synthetic fertilizer use categorized between the 16 crops are shown in Fig. 11 and Fig. A5 (see Appendix). The seasonal trends for both 2010 and 2018 are generally the same. The highest emission of around 4.0 Tg N month⁻¹ occurs in July, with August showing the second highest emission of around 2.5 Tg N month⁻¹. Large emissions take place in between April and August. The first emission peak is in May, which is the first month of the year

when NH₃ emissions reach 2.0 Tg N month⁻¹ in both years. Emissions slightly decrease in June, but then reach the maximum in July. Wheat-related emissions are seen throughout the year and are the most significant emissions in most months, expect for August, September and October, in which rice contributes to the largest estimated emissions. Maize is also one of the most important crops that result in NH₃ emissions from May to August.

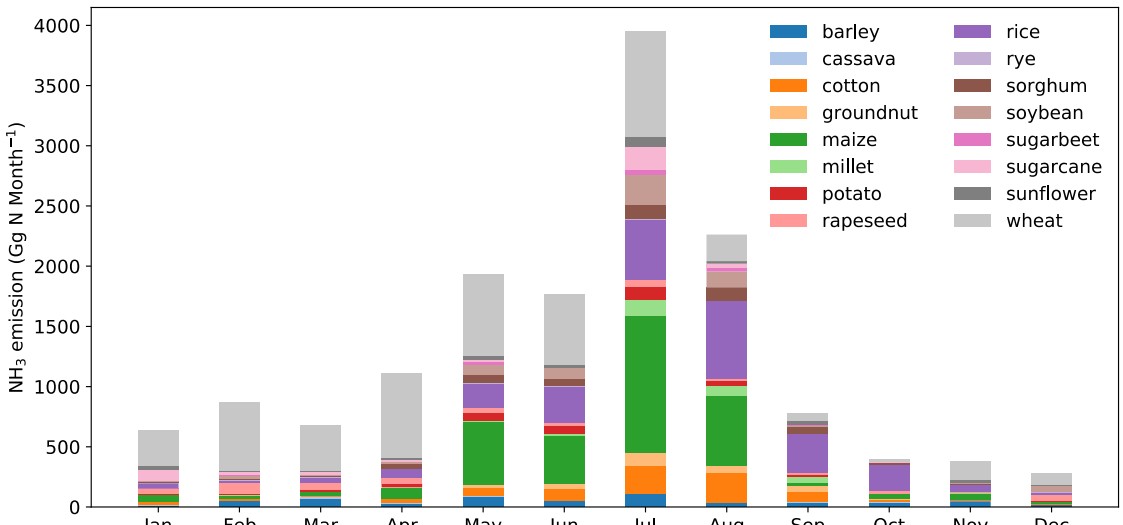

**Figure 11. Global monthly NH₃ emissions (Gg N month⁻¹) from ammonium and urea fertilizer applications for 16 major crops in 2010 simulated by AMCLIM–Land.**

The seasonality of NH₃ emissions differs across the globe between regions. Figure 12 and Figure A6 present monthly NH₃ emissions from 12 different geographical regions and the percentage of global monthly emissions that each region contributes. The map of the geographical regions defined in AMCLIM is given by Fig. A11 in the Appendix. The highest emissions are from South Asia and East Asia, with both regions responsible for roughly a quarter of global emissions. North America has the third highest emissions, accounting for over 17 % of global emissions. Southern Africa has the lowest emissions, which only accounts for about 1 % of the global total. In terms of country-level statistics, China results in the largest emissions of 3.7 Tg N yr⁻¹ in 2010 and 4.1 Tg N yr⁻¹ in 2018, followed by India which contributes to 3.4 Tg N yr⁻¹ and 3.5 Tg N yr⁻¹ in 2010 and 2018, respectively. US is the third largest emitter country with emissions of 1.9 Tg N yr⁻¹ in 2010 and 2.0 Tg N yr⁻¹ in 2018.

Regions in the NH, including East Asia, South Asia, North and Central Asia, Northern Europe and the Mediterranean, North America and the Middle East, generally show high emissions in JJA and MAM, as shown in Figs. 12 and A6. In particular, East Asia, Northern Europe and North America exhibit very similar monthly variations. North and Central Asia and South Asia also show similar monthly trends. The NH₃ emissions in the Mediterranean region are high in spring and

summer emissions, but much lower in autumn and winter. By contrast, South America and South Africa show higher winter
emissions and lower emissions in other seasons. Oceania has distinct seasonal emission patterns. The emission in Oceania peaks in May and is high in August and September. Figure 12 and Figure A6 show that different regions dominate NH$_3$ emissions in different seasons, with East Asia and South Asia being the largest contributors to spring and summer emissions, while South Asia and South America dominate autumn and winter emissions. North America also contributes significantly to summer emissions.

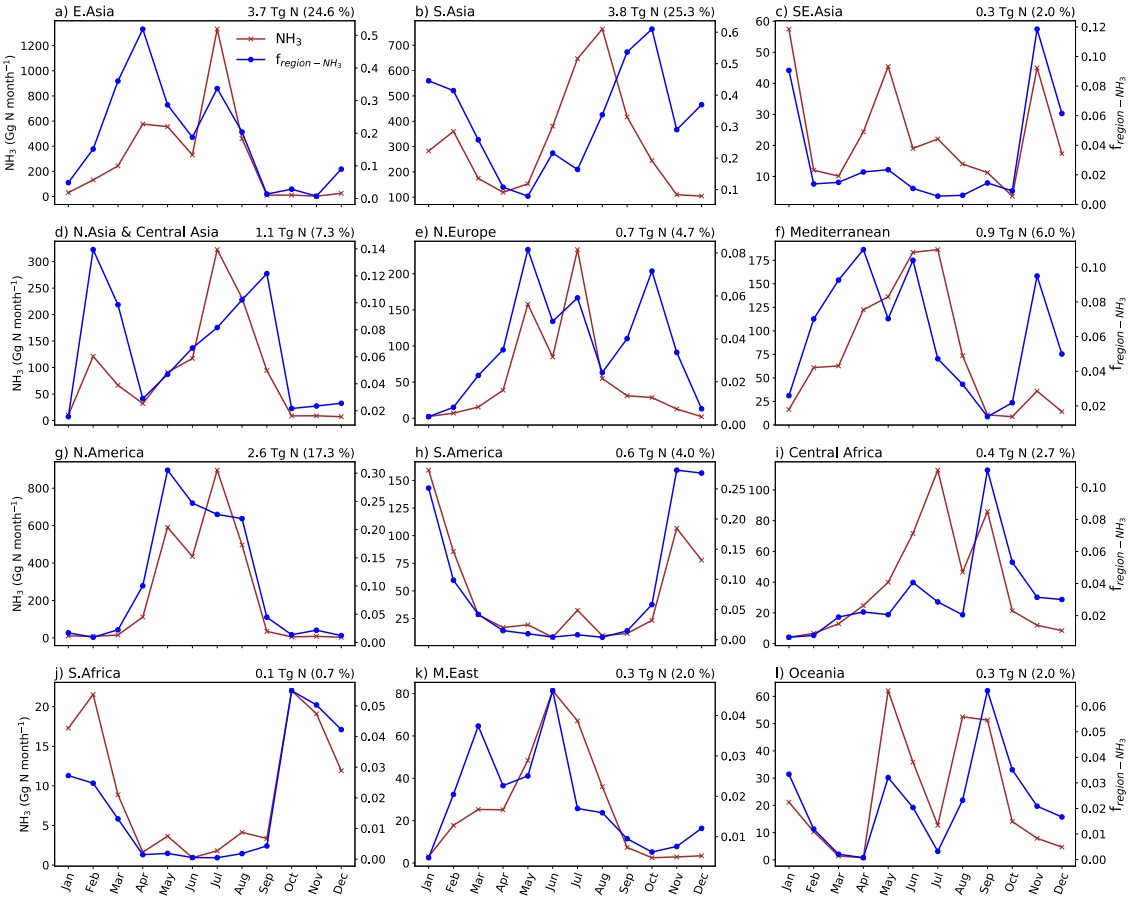

**Figure 12. Monthly NH₃ emissions from ammonium and urea fertilizer application in different regions of the world and the relative fraction of the global monthly emissions that are from the corresponding regions ($f_{region}$). Annual total NH₃ emissions of the region are given at the top right corner of each plot, with the percentage of emissions from this region. The figure is for 2010.**

Table 2 provides a summary of the regional volatilization rates from the use of synthetic fertilizers in both 2010 and 2018. The data show that ammonium application generally results in higher estimated volatilization rates than urea application according to the AMCLIM model, noting that the values would be half those shown if ammonia volatilisation is referenced

to total N input where ammonium nitrate is used. Africa and Oceania have the highest volatilization rates for both 2010 and 2018, with over 23 % to 25 % of N in ammonium and urea application volatilized as $NH_3$, while South America shows the lowest volatilization rates of less than 13 %. In the listed regions, the volatilization rates of both ammonium and urea application for 2018 are higher than 2010, except for Europe, North America and South Asia.

**Table 2. Volatilization rates of synthetic fertilizer use in different regions for the years 2010 and 2018 (percentage values). Values in the parentheses are volatilization rates when including nitrate application. *Volatilization rates are half the specified figure if referenced against total N input when applying ammonium nitrate.**

| Year | Fertilizer | Africa | East Asia | Europe | North America | South Asia | South America | Oceania | Other part of Asia | Global |
|------|-----------|--------|-----------|--------|---------------|------------|---------------|---------|--------------------|--------|
| 2010 | Ammonium* | 23.6 | 17.5 | 17.6 | 19.4 | 25.7 | 14.2 | 27.3 | 19.4 | 19.3 |
|      | Urea | 23.6 | 13.1 | 14.9 | 15.3 | 20.0 | 11.2 | 21.1 | 16.0 | 16.1 |
|      | All | 23.6 (19.5) | 14.5 (11.7) | 16.4 (11.6) | 17.9 (15.9) | 20.9 (19.9) | 12.3 (10.9) | 23.7 (22.2) | 16.8 (15.1) | 17.2 (14.6) |
| 2018 | Ammonium* | 25.8 | 17.8 | 15.1 | 18.0 | 23.3 | 13.6 | 31.1 | 17.9 | 18.1 |
|      | Urea | 24.5 | 12.7 | 13.8 | 14.5 | 17.4 | 12.5 | 26.9 | 16.6 | 15.7 |
|      | All | 25.0 (21.5) | 15.0 (11.2) | 14.5 (10.6) | 16.6 (14.8) | 18.3 (17.6) | 12.9 (11.7) | 28.3 (27.0) | 16.9 (15.3) | 16.7 (13.9) |

## 4 Discussion

### 4.1 Comparison with other studies

The GRAMINAE dataset proved especially useful for testing the model in detail given that a robust set of ammonia flux data is available (based on comparison of several instruments) together with extensive supporting data which allow AMCLIM to be applied at a high time resolution (15-min resolution). However, many published ammonia flux datasets do not have comparable detail to the GRAMINAE dataset. This highlights the importance of providing sufficient information on micrometeorological conditions, soil data and management information to allow detailed model simulations.

Nevertheless, it is possible to make a simpler overall comparison between the results of the global simulations of AMCLIM and the average volatilization rate of other published studies according to latitude and longitude of each study. In a simple way, this takes account of the effects of environment/climate, soil and management conditions as they are estimated

in AMCLIM. By doing this, the simulated volatilization rates from synthetic fertilizer use by AMCLIM were compared with measurements, focusing on experimental studies that measured cumulative $NH_3$ emissions from urea and ammonium fertilizer application to agricultural land (Fig. 13; detailed information can be found in Table A2 in Appendix). Modelled volatilization rates were extracted from the global simulations and compared with the reported volatilization rates from these experimental studies that were conducted in different regions across the globe. These experimental studies used for comparisons were from 17 sites in seven countries (as shown in Fig. A13). The sites included show a reasonable spatial distribution from various climatic conditions (representing by average temperature from 8.5 to 31.7 °C during measurements) and soil conditions (representing by soil pH between a range from 5.7 to 8.5), with five major crops being examined.

As shown in Fig. 13a, AMCLIM can either overestimate the volatilization rates of urea application compared with measured estimates (e.g., studies "E", "K", "M", "N", "P", "Q" and "R") or underestimate (e.g., study "A"). For ammonium fertilizer application (Fig. 13b), AMCLIM overestimated the volatilization rates, which has been also shown in the GRAMINAE simulations (Fig. 5d). The differences between the observed and modelled $P_V$ rates could be due to the diffusion and infiltration processes as irrigation was usually done in the experimental studies, and the complex soil pH dynamics after urea application. It is also worth noting that the results from global simulations are largely dependent on the timing of fertilization (crop calendar), which might not be reflected in or not consistent with the experimental studies. The limitations of such comparisons emphasize the urgent need for well-documented and good quality measurements to improve the development and performance of models.

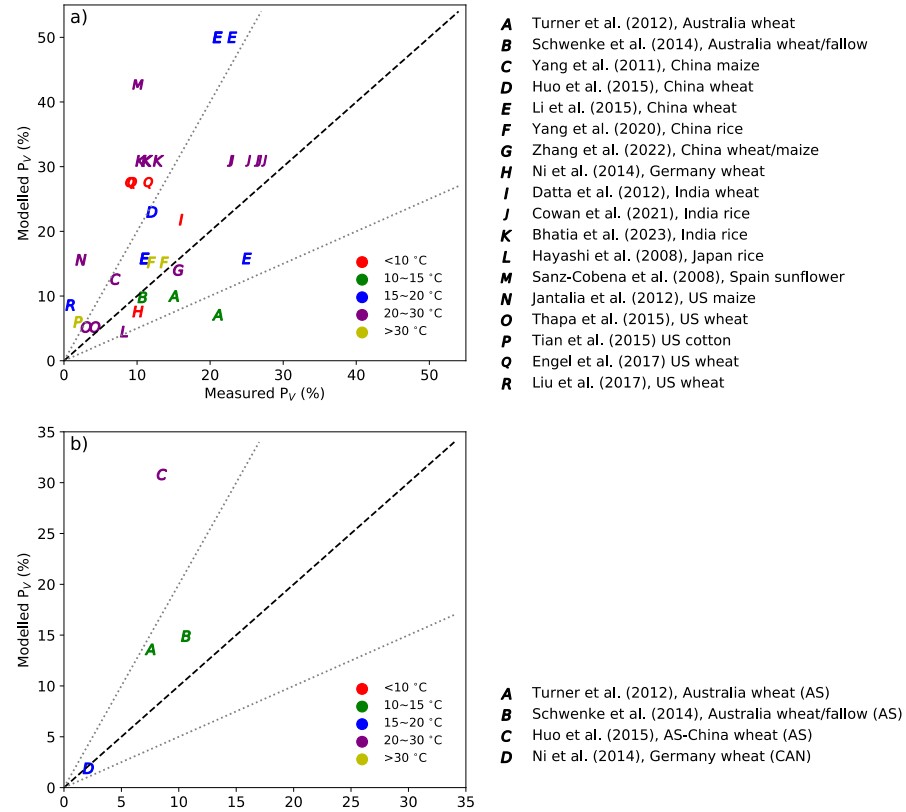

**Figure 13. Modelled percentage volatilization rates ($P_V$, %) compared with experimental studies (Hayashi et al., 2008; Sanz-Cobena et al., 2008; Yang et al., 2011; Datta et al., 2012; Jantalia et al., 2012; Turner et al., 2012; Ni et al., 2014; Schwenke et al., 2014; Huo et al., 2015; Li et al., 2015; Tian et al., 2015; Thapa et al., 2015; Engel et al., 2017; Liu et al., 2017; Cowan et al., 2021; Zhang et al., 2022; Bhatia et al., 2023). Measurement data were from literature that studied NH$_3$ volatilization from (a) urea application and (b) ammonium fertilizer application to field. Black dashed line is the 1:1 line, and grey dotted lines indicates the values within a factor of 2. AS: ammonium sulfate; CAN: calcium ammonium nitrate.**

As shown in Fig. 14a and Fig.14b, both measured and modelled $P_V$ tend to be larger as soil pH increases, with modelled

$P_V$ having a stronger response. Results of ammonium fertilizer application illustrate more obviously positive correlation between volatilization rates and soil pH compared with urea application, although it could be due to less data points of studies. The general trend is consistent with the fact that more alkaline soils can result in higher NH$_3$ volatilization. The soil pH used in AMCLIM for global simulations sometimes are lower than the measured values, and this indicates that the simulated volatilization rates would have been even larger if using higher pH. The difference in pH could be partly due to the

reason that the model still not having high enough resolution to deal with the heterogeneity of soil characteristics. Overall, the comparisons reflect that AMCLIM is likely to overestimate NH$_3$ emissions from fertilizer application as compared with the reported measurement results.

The comparatively high volatilization rates of ammonium fertilizers estimated by AMCLIM are possibly due to the following reasons:

1) AMCLIM does not simulate the dissolving process of ammonium fertilizers and instead it assumes that ammonium "pellets" instantly dissolves in soil moisture according to the soil pH specified in the global soil database, which result in large initial emission potential and might cause an overestimation of $NH_3$ emissions (Fig. 5d shows that the majority of overestimation by AMCLIM occurs in the first day).

2) The drainage and diffusion in AMCLIM might be underestimated so that more N in the ammonium fertilizer is available

for volatilization than may be the case in actuality.

3) The present version of AMCLIM does not include the potential for $NH_3$ recapture by an overlying plant canopy. This may be particularly relevant for tall canopies and those with small soil ammonia emissions in wet environments.

4) Measurements also have uncertainties, especially when using enclosure methods to measure $NH_3$ emissions (Kamp et al., 2024), which can significantly underestimate emissions. For example, companion studies at the GRAMINAE site suggested

that $NH_3$ fluxes measured by a cuvette system were only about a quarter of the net canopy fluxes estimated using micro-meteorological techniques, e.g., AGM and REA (David et al., 2009; Milford et al., 2009; Sutton et al., 2009b). Low levels of turbulent mixing and absorption to surfaces (especially if condensation occurs) are possible reasons for such underestimation of measured fluxes by chamber systems.

Each of these points indicates the need for further experimentation and comparison of published studies on the

dependence of ammonia emission on dissolution processes, soil properties, canopy structure and alternative fates of the added nitrogen (cf. Fig. 9), with the prospect of improved simulation of these interactions. For example, future development of AMCLIM is planned to include assessment of interactions with the overlaying plant canopy.

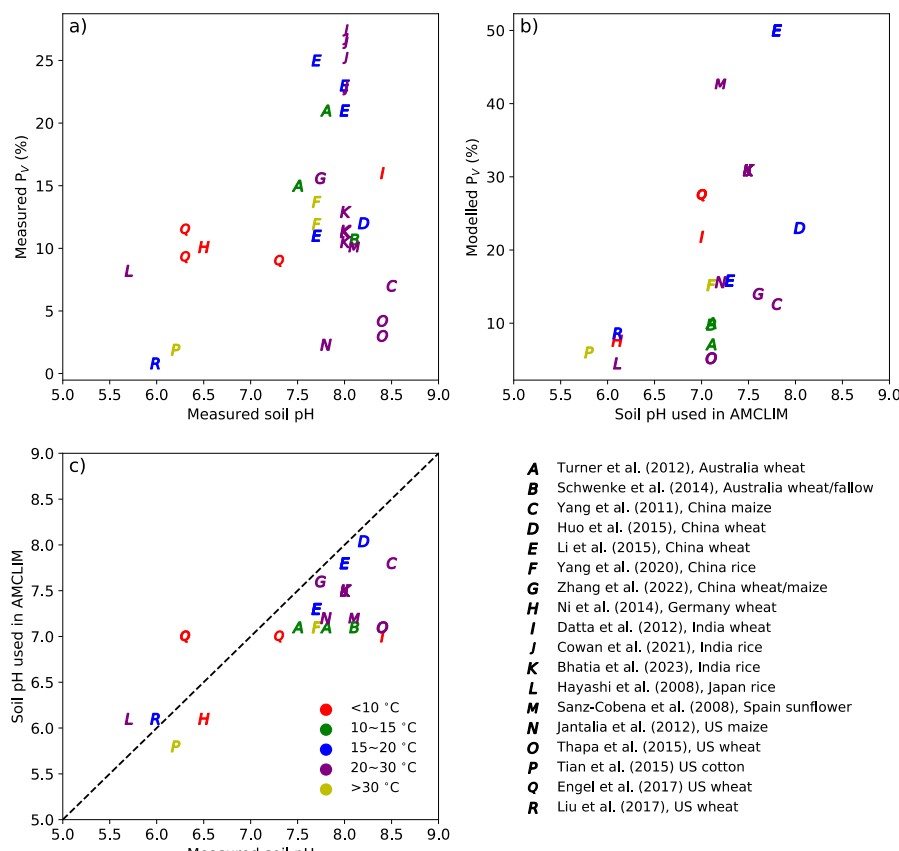

A Turner et al. (2012), Australia wheat
B Schwenke et al. (2014), Australia wheat/fallow
C Yang et al. (2011), China maize
D Huo et al. (2015), China wheat
E Li et al. (2015), China wheat
F Yang et al. (2020), China rice
G Zhang et al. (2022), China wheat/maize
H Ni et al. (2014), Germany wheat
I Datta et al. (2012), India wheat
J Cowan et al. (2021), India rice
K Bhatia et al. (2023), India rice
L Hayashi et al. (2008), Japan rice
M Sanz-Cobena et al. (2008), Spain sunflower
N Jantalia et al. (2012), US maize
O Thapa et al. (2015), US wheat
P Tian et al. (2015) US cotton
Q Engel et al. (2017) US wheat
R Liu et al. (2017), US wheat

**Figure 14. Measured percentage volatilization rates ($P_V$, %) vs. measured soil pH (a). Modelled $P_V$ vs. soil pH used in AMCLIM (b). Soil pH used in AMCLIM compared with measured soil pH (c), with black dashed line the 1:1 line.**

Table 3 compares the simulated $NH_3$ emissions from synthetic fertilizer use by AMCLIM–Land with those estimated by other studies, including models and inventories, on global, continental and national scales. Table 3 also includes volatilization rates, where available. Among all studies for comparisons, DLEM and FAN are both processed-based models (see Table 3 for references of all model descriptions/studies), while the other studies are inventories that mainly use EFs methods. DLEM incorporates a bi-directional exchange scheme for $NH_3$, and FAN is interactively coupled to an Earth System Model.

Table 3 shows AMCLIM–Land compared with DLEM and FAN at several spatial scales. Among models, estimated global $NH_3$ emissions by AMCLIM are the second largest, which are in close agreement with DLEM. For the year 2010, AMCLIM and DLEM estimate 15.0 Tg N yr$^{-1}$ and 16.7 Tg N yr$^{-1}$ of $NH_3$ emissions, respectively. By comparison, the FAN model provides lower estimates of 12 Tg N yr$^{-1}$ for 2000 by FANv1 and 11 Tg N yr$^{-1}$ for 2010–2015 by FANv2. The lower $NH_3$ emissions suggested by FAN is partially due to less total N application in FANv2, which is 79 – 87 Tg N yr$^{-1}$ compared

to 102 Tg N yr$^{-1}$ in AMCLIM and DLEM. The volatilization rates estimated by the three models are comparable, ranging between 13 to 16 %. For different types of fertilizers, it is estimated that about 16 % of N in urea fertilizers is lost as NH$_3$ compared with 19 % by FANv2, and NH$_3$ emissions from ammonium and nitrate fertilizer application account for 12 to 13 %, which is higher than 7 % estimated by FANv2, while DLEM does not specifically report NH$_3$ emissions from urea or ammonium fertilizers.

Global estimates of emissions from other studies vary significantly, ranging from 5.9 to 28.6 Tg N yr$^{-1}$. MASSAGE_NH$_3$ and NH$_3$_stat both suggest that annual global NH$_3$ emissions are less than 10 Tg N yr$^{-1}$ for 2008 and 2012. The NH$_3$_stat model estimates much lower emissions of 5.9 Tg N yr$^{-1}$, which is only about 35 % of AMCLIM's result. In contrast, estimated emissions by Yang et al. (2023) are the highest (28.6 Tg N yr$^{-1}$) for the 2010s. The large differences among the studies can be partly explained by the different agriculture activities included and different input data used in each study.

For NH$_3$ emissions from major continents and emitters (China, India and US), AMCLIM provides consistent estimates as compared with DLEM for regions including Africa, Asia, Europe and China, but higher emissions than FANv2. However, the volatilization rates of AMCLIM and FANv2 agree with each other in Africa, Asia, Oceania, China and India, indicating that the different NH$_3$ emissions can be partly explained by the different inputs of N fertilizer to the models. As shown in Table 3, AMCLIM has similar estimates for NH$_3$ emissions with other studies for most of regions expect for North America, 760 US and India. For the US, emissions estimated by AMCLIM are higher than EPA by 60 % and NH$_3$_stat and MASSAGE_NH$_3$ by two to four times. Meanwhile, NH$_3$ emissions from India are also higher in AMCLIM; approximately 10 % to 20 % higher than other models and inventories. Only Yang et al. (2023) suggested even higher NH$_3$ emissions from India than AMCLIM. However, the volatilization rate for India estimated by AMCLIM is similar to FANv2. The slightly lower values of AMCLIM than FANv2 indicate that the difference mainly results from different input data used. For 765 example, the total N application in India is 16 Tg N yr$^{-1}$ in AMCLIM compared to 10 Tg N yr$^{-1}$ in FANv2.

**Table 3. Comparisons of global, continental and national NH$_3$ emissions from fertilizer (Tg N yr$^{-1}$) and corresponding volatilization rates (%) between AMCLIM and other inventories, models and studies.**

| Model/ Study | Year | Global | Africa | Asia | Europe | North America | South America | Oceania | China | India | US |
|---|---|---|---|---|---|---|---|---|---|---|---|
| DLEM[a] | 2000s, 2010 | 13.6, 16.7 (16.3%) * | 0.5 | 9.0 | 1.9 | 2.0 | 0.9 | 0.2 | | | |
| DLEM[b] | 2000-2014 | | | | | | | | 4.1 | 2.8 | |
| FANv1[c] | 2000 | 12 | | | | | | | | | |
| FANv2[d] | 2010-2015 | 11 (13%) | 0.3 (20%) | 5.9 (15.6 %) * | 0.7 (6%) | 1.3 | 0.6 (17%) | 0.2 (22%) | 2.3 (11%) | 2.7 (26%) | |
| Literature[e] | 2000 | 11 | | | | | | | | | |

| Model | Year | | | | | | | | | | |
|---|---|---|---|---|---|---|---|---|---|---|---|
| Literature[f] | 2010s | 28.6 | | | 1.4 | | | | 5.5 | 6.9 | 1.3 |
| Literature[g] | 2008-2010 | | | | | | | | 2.4-5.2 | | |
| Literature[h] | 2003-2010 | | | | | | | | | 2.2-3.3 | |
| EPA[i] | 2011 | | | | | | | | | | 1.2 |
| MASSAGE_NH3[j] | 2008 | 9.4 | | | | | | | 3.0 | | 0.5 |
| NH3_stat[k] | 2012 | 5.9 | | | | | | | 0.7 | 0.6 | 0.8 |
| AMCLIM (this study) | 2010 | 15.0 (14.6%) | 0.6 (19.5%) | 9.2 (14.8%) | 1.6 (11.6%) | 2.6 (15.9%) | 0.6 (10.9%) | 0.3 (22.2%) | 3.7 (12.0%) | 3.4 (21.0%) | 1.9 (15.5%) |

[a] Xu et al. (2019) [b] Xu et al. (2018) [c] Riddick et al. (2016) [d] Vira et al. (2020) [e] Beusen et al. (2008) [f] Yang et al. (2023) [g] Kurokawa et al. (2013); Kang et al. (2016); Zhang et al. (2017, 2018) [h] Aneja et al. (2012); Kurokawa et al. (2013) [i] EPA, 2011 [j] Paulot et al. (2014) [k] Aneja et al. (2020) [*] Values are calculated based on on the results in the literature.

Key features of several models are summarised and listed in Table 4. Compared with existing models, AMCLIM is a dynamical emission model with an emphasis on the $NH_3$ volatilization. AMCLIM shows adequate level of complexity in terms of soil layering construction, simulations for N processes in soils, $NH_3$ volatilization simulation and soil pH dynamics. AMCLIM has relatively high temporal resolution, which provides implications in the temporal variations of $NH_3$ fluxes. The highly resolved outputs can be used by atmospheric transport/chemistry models. It is worth noting that AMCLIM is considered as a comprehensive emission model rather than a biogeochemical model.

**Table 4. Comparisons of model features between AMCLIM and other models for $NH_3$ emission simulations. [1*]Based on Gurung et al (2021) that developed a new $NH_3$ volatilization scheme for urea application in DayCent. [2*]Based on a model version DLEM-Bi-$NH_3$ (Xu et al., 2018).**

| Model | Model type | N processes in soils | Soil pH change/dynamics | $NH_3$ volatilization process | Vegetation interactions at the surface | Temporal resolution |
|---|---|---|---|---|---|---|
| AMCLIM | Dynamical $NH_3$ emission model | Four soil layers up to 28 cm depth; A+B+C+D+E+F+G+H+I | Yes; simple generalised scheme but buffering capacity not considered | M1 | No | Sub-hourly/hourly |
| CAMEO | $NH_3$ Emission module embedded to ORCHIDEE | 11 soil layers for hydrology; A+B+E+F+H+I | No | M1 | No | Sub-hourly, daily, yearly |

| DayCent[1*] | Biogeochemical model | 14 soil layers up to 210 cm depth; A+C+F+H+I | Yes; empirically derived formula with buffering capacity included | M2 | No | Daily |
|---|---|---|---|---|---|---|
| DLEM[2*] | Terrestrial ecosystem model | Unknown soil layering; soil N pools are not explicitly simulated but are derived from fertilizer application rate[2*] | No | M1 | Bi-directional exchange scheme | Daily |
| DNDC | Biogeochemical model | Five soil layers up to 50 cm depth; A+B+C+D+F+H+I | Yes; empirically derived formula with buffering capacity included | M2 | No | Daily |
| FANv2 | Process-based N model coupled to CESM | One soil layer 2 cm depth; A+B+C+D+E+F+G+H+K | Yes; pH varies based on age classes | M1 | No | Sub-hourly |

A–mass balance calculation of N pools

B–$NH_3/NH_4^+$ equilibrium

C–urea hydrolysis

D–TAN partition

E–surface runoff

F–leaching

G–diffusion in soils

H–nitrification

I–plant N uptake

J–microbial N uptake

K–mechanical N loss

M1–Fluxes are concentration gradient driven and constrained by resistances derived from well-established micrometeorological theory

M2–Empirically derived mass transfer coefficient

For modelling global scale $NH_3$ emissions, calibrations were not explicitly done at site scale for models such as FANv2 (Vira et al., 2020), CAMEO (Beaudor et al., 2023), DNDC (Yang et al., 2022) and DLEM (Xu et al., 2019). This possibly because global models tend to provide general representation and try to avoid over calibration. In contrast, DayCent is widely used for simulating $N_2O$ emissions rather than $NH_3$ emissions and is intensively calibrated using $N_2O$ measurements from fields. The parameters can be quite different between simulations for different places, e.g., North America vs. Europe. However, there are limited studies of applying DayCent at global scale, and little evidence was found that explicit model calibration was done in the global application of DayCent (e.g., Del Grosso et al., 2009).

In this study, the AMCLIM model was firstly applied to simulate the NH$_3$ emission from fertilized grassland at the GRAMINAE site and was intensively evaluated by a detailed comparison of timeseries between measured and modelled fluxes. AMCLIM was then applied at the global scale to simulate NH$_3$ emissions from 16 major crops. Admittedly, there is a gap resulted from using model settings for fertilized grassland to represent croplands. The development of AMCLIM was based on understanding at process-level. The management practices at the GRAMINAE site were not complicated, which provides a good test situation for the numerical representations of the physical and chemical processes. The model results show close agreement with the GRAMINAE measurements (Fig. 5). The multi-site comparison demonstrates that the model provides reasonable estimates for various crops under different climatic and soil conditions (Fig. 13), and the global estimates by AMCLIM are broadly consistent with existing models (as shown in Table 2). On the other hand, it has been found that the critical factor affects NH$_3$ emissions is the timing of fertilization and amount of fertilizer applied under current model settings (see Section 4.3, and Fig. A14 and A15). Therefore, considering the model structure and complexity, the processes included in AMCLIM are believed to be robust and representative for simulations for synthetic fertilizer use.

Table 5 summarizes the crop-specific NH$_3$ emissions from synthetic fertilizer applications estimated by AMCLIM–Land and other studies. Although there is limited data available for NH$_3$ emissions from individual crops, the results from each study are generally consistent in magnitude. Zhan et al. (2021) focuses on year 2000, and their values are the smallest. The results from Yang et al. (2023) are average values for the period from 2010 to 2018 and are generally higher. All studies agree that wheat, rice and maize are the top three crops that dominate the NH$_3$ emissions. AMCLIM has similar estimates for maize, but much higher wheat emissions and lower soybean emissions compared with DLEM. The emissions related to rice estimated by AMCLIM are nearly 50 % lower than estimates by DLEM, which might be partly due to the fact that AMCLIM does not include a flooded paddy scheme for rice simulations. Urea hydrolysis can be faster in flooded rice paddy, which can result in higher NH$_3$ emissions, depending on floodwater pH. There is no data for fertilization of this type of rice systems in the GGCMI3 dataset that was used in AMCLIM. But the simplification is considered reasonable for global application of the current model version as the comparisons (as shown in Fig. 13) indicate the simplification provide estimates that are not out of order of magnitude (mostly within a factor of two). The differences in crop-specific NH$_3$ emissions highlight the need for further research to improve the understanding of NH$_3$ emissions from different crops and fertilizer management practices. Future work should include the addressing of individual cropping systems in more detail.

**Table 5. Crop-specific NH$_3$ emissions (Tg N yr$^{-1}$) from synthetic fertilizer use simulated by AMCLIM and comparisons with other studies.**

| Study | Year | Barley | Cotton | Groundnuts | Maize | Potato | Rapeseed | Rice | Sorghum | Soybean | Sugarcane | Wheat |
|-------|------|--------|--------|------------|-------|--------|----------|------|---------|---------|-----------|-------|

| | | | | | | | | | | | | |
|---|---|---|---|---|---|---|---|---|---|---|---|---|
| DLEM[a] | 2000s | | | | 3.3 | | | 3.5 | | <1.5 | | 3.4 |
| Yang2023[b] | 2010s | 1.3 | | | 3.7 | | | 4.9 | | 0.9 | | 6.0 |
| Zhan2021[c] | 2000 | 0.30 | | 0.22 | 2.2 | 0.33 | 0.20 | 3.0 | 0.22 | 0.30 | 0.38 | 3.0 |
| AMCLIM | 2010 | 0.58 | 0.84 | 0.30 | 3.0 | 0.35 | 0.49 | 2.4 | 0.53 | 0.68 | 0.41 | 4.6 |

[a] Xu et al. (2019) [b] Yang et al. (2023) [c] Zhan et al. (2021)

## 4.2 Spatial and temporal variations in NH$_3$ emissions

The NH$_3$ emissions from synthetic fertilizer use are primarily determined by the amount of N applied and are strongly influenced by both environmental conditions and local management practices. High NH$_3$ emission regions are typically found in countries with intensive agricultural activities such as China, India, Pakistan and US, where large amount of synthetic fertilizer N has been used. The $P_V$ rate is an important indicator that shows the percentage of applied N volatilizes as NH$_3$. The regional pattern of the $P_V$ rate does not always match the distribution of NH$_3$ emissions due to the combined

effect of environmental factors and management practices. Since AMCLIM was applied in this study using the management practices that do not vary significantly for 2010 and 2018, this means that differences in $P_V$ between these years are mainly due to environmental differences between these years, as shown in Fig. 7 and Fig. A2.

When considering the environmental effects on NH$_3$ emissions, there are several factors that can cause volatilization rates to vary, including soil pH, soil temperature and moisture and wind speed. Alkaline soils tend to cause higher estimated NH$_3$

emissions in AMCLIM so that regions with high soil pH, such as western US, Argentina, the Middle East, Namibia, Mongolia and part of northern China show high volatilization rates (soil map as shown in Fig. A12 in the Appendix). Since the base soil pH distribution is fixed in AMCLIM–Land according to HWSD v1.2 (FAO and IIASA, 2012; Wieder et al., 2014) and does not vary over time, the similar geographical patterns of the volatilization rates in the two simulated years of 2010 and 2018 indicate clear climatic dependences (due to temperature, water and wind conditions) featured in NH$_3$

volatilization.

High temperature leads to faster rates and quicker processes, which can result in larger emissions. Soil moisture influences the concentrations of N species in soils. When the soil is dry, soil TAN concentrations can be high, which may result in a greater emission potential at the soil surface, especially under alkaline soil conditions. This effect could be further amplified for ammonium fertilizer application as AMCLIM does not simulate the initial dissolving of fertilizer pellets. Dry regions,

such as Mongolia, Namibia, western US and the Middle East show high volatilization rates. It is worth noting that these regions also have alkaline soils with high pH values, suggesting that the high volatilization may be due to a combined effects of soil dryness and alkalinity. Moreover, when the soil is dry and the subsurface percolation flux is small, there may be a lack of infiltration/drainage, which prevents N from moving from the surface to deeper soil layers. Instead, more N will

volatilize as $NH_3$ from the surface. It is notable that the estimated N fluxes to run off are lower in 2018, but the $NH_3$ volatilization rate is higher in 2018 than in 2010 (Fig. 9). Wind speed is also a critical factor that impacts $NH_3$ volatilization since it influences the turbulence which affects the atmospheric resistances. Emissions are higher under windy conditions because atmospheric resistances are smaller. Simulations for the GRAMINAE site indicate that the sub-hourly $NH_3$ emissions vary with temperature and the friction velocity (which is related to wind speed and atmospheric resistances; Fig. 5) and show strong diurnal cycles. Rainfall can also affect the $NH_3$ emission, mostly causing a reduction. Similarly, nitrogen species are washed off from the land surface during heavy rainfall event. Although the effects of rainfall are not explicitly included in AMCLIM, they are reflected implicitly by the runoff fluxes, and the magnitude of scavenged $NH_3$ is small compared with the emission flux. The above meteorological factors can interactively affect the $NH_3$ emission.

Management plays another key role in affecting the $NH_3$ emissions in the agricultural activities, specifically through the timing of fertilizer application during planting seasons, the type of fertilizer used and the application techniques. The temporal variations of $NH_3$ emissions are largely related to the timing of fertilizer application, since volatilization usually takes place soon after the fertilizers are applied. The regional monthly emissions are closely linked to the planting seasons, with large emissions being found in a few months throughout the year. On the global scale, $NH_3$ emissions are the highest in MAM and JJA, with the first peak of emissions in May and the largest emissions in July, corresponding to the typical planting seasons for crops in the NH.

The second factor is the type of fertilizer used. According to simulations using AMCLIM–Land, application of ammonium and urea have similar volatilization rates on the global scale. The comparable volatilization rates of the two fertilizer types are possibly because of the following reasons. Ammonium is a direct input to soil TAN pool which is readily to be volatilized as $NH_3$, while urea must be hydrolysed before it is converted to TAN. The hydrolysis process is limited by water availability. If the soil is very dry, the amount of urea that hydrolyses is reduced (Rodríguez et al., 2005). Furthermore, hydrolysis of urea can cause soil pH to increase, leading to more $NH_3$ emissions. As a result, $NH_3$ volatilization from urea application is controlled by two processes with opposite effects. When including both ammonium and nitrate fertilizers (e.g., ammonium nitrate, the N content doubles but volatilization rate of $NH_3$ halves because the nitrate part does not contribute to $NH_3$ emissions), urea application is found to result in higher $NH_3$ emission due to the elevated soil pH by hydrolysis.

The third critical factor is the application techniques. How fertilizers are applied on land can have huge impacts. Broadcasting is the most commonly used method and contributes to the largest fraction of $NH_3$ emissions, while both immediate incorporation and deep placement of fertilizers are effective methods that can reduce $NH_3$ emissions to a large extent. According to the assumptions specified in Table A1, only a small fraction of fertilizer was incorporated according to development level of each region (WB, 2022). In practice the adoption of increased used of incorporation and deep placement could substantially reduce $NH_3$ emissions as compared with the present estimates. The availability of national statistics on such practices is therefore a priority for improving estimates of the model, as well as of demonstrating the benefits of improved fertilizer placement practices.

Based on current estimates, less than 50 % of applied N is taken up by crops, which may be partly due to the very simple application techniques used. In addition to the three factors discussed above, irrigation can also influence $NH_3$ emissions as it has been found in literature that less $NH_3$ emissions occur after irrigation (Dawar et al., 2011; Yang et al., 2022; Zhang et al., 2022), although AMCLIM is not evaluated against observations for this effect because of insufficient input data. Irrigation leads to an increase of the soil moisture and reduction of emissions by diluting concentrations of N species and transporting them to deeper soil layers. Proper management practices, such as timely and precise application of fertilizers and adequate application techniques can help reduce $NH_3$ emissions and improve crop uptake of N.

## 4.3 Uncertainty and limitations

In general, uncertainty in $NH_3$ emissions arises from two main aspects: input data and model parameters. Input data uncertainty includes the N application rates, the crop calendars that determine the application dates and the soil characteristics. The crop calendars used in AMCLIM are static. Emissions could be influenced if using different crop calendars as the environmental conditions may also change. AMCLIM assumes that fertilizers are applied twice during the growing season, which is a moderate value used as a representation for fertilization. This value mostly varies between one to three or four times across the globe. For example, there can be two to four times of fertilizer application in North America or zero to three times for South America (Xu et al., 2019). Additional rounds of simulations were performed to test three possible scenarios, 1) 100 % fertilizer N applied at the beginning of planting season, 2) 75 % N applied at planting and 25 % midway, and 3) 40 % N at planting, 30 % at one third and 30 % at two thirds of the growing season. The global $NH_3$ emissions from synthetic fertilizer use based on the different scenarios were 10.8, 12.9 and 15.8 Tg N $yr^{-1}$, respectively, as compared with the base assumption of 15.0 Tg N $yr^{-1}$ (when applied at 50%:50%). In general, this shows that in AMCLIM adding a larger share of fertilizer later in the growing season is associated with increased emission, which can be linked to warmer temperatures as the growing season progresses. However, it should be noted that further testing of this effect would be warranted given the possible effect of tall crop canopies in reducing emissions, which is not addressed in the present version of AMCLIM reported in this study. The differences in spatial distribution and seasonal variation between different fertilization scenario are shown by Fig. A14 and A15.

In AMCLIM, only $NH_3$ emissions from grazed grassland were simulated, which will be described by the second part of the model in a forthcoming paper (for the livestock sector). Fertilized grasslands (with synthetic fertilizers) were not included and simulated because there is no data specified for this type of vegetation in the GGCMI3 dataset used. This addresses the need from future work on more completed compilation of statistical data and model input data. It is worth noting that the majority of synthetic fertilizer used globally was for croplands rather than grasslands (although in Europe and some other locations grasslands can receive significant amount of fertilizers). The total applied N from synthetic fertilizer

was 102.3 Tg N yr$^{-1}$ in AMCLIM, which is comparable to the 99.6 Tg N yr$^{-1}$ of consumed fertilizer suggested by the International Fertilizer Association.

The heterogeneity of the land during fertilization can also contribute to uncertainty in the overall estimates. In intensive farming countries like China and India, where fertilization can take up to a week or more, the assumption of fertilization being completed within a day in AMCLIM introduces uncertainty. Soil characteristics including soil pH, bulk density, soil constituents and organic matter content, etc., are assumed to remain constant in AMCLIM–Land, which can affect the chemical equilibrium and variables dependent on these data.

Uncertainty is also introduced from various parameters used in the AMCLIM model. First, the representation of soil pH evolution after urea application relies on an empirical relationship due to the complexity in simulating soil pH dynamics. Soil pH is assumed in AMCLIM to reach a maximum of 8.5 after urea application and will eventually decline to the original value. The duration of such perturbed soil pH is assumed to be around one week. In addition, long term trends of soil pH changes, i.e., soil acidification due to fertilization, are not included in AMCLIM. Instead, simulations for different years used

the same base soil pH. Potentially the most important uncertainty is the extent of variations in soil pH between microsites, as this can substantially affect simulated ammonia emissions, for example if dissolving fertilizer particles are not well coupled to soil pH.

  Second, AMCLIM assumes the atmospheric concentrations of $NH_3$ to be zero rather than a positive value for the background $NH_3$, which may cause some overestimation of emissions, although this is expected to be small (Fig. 6). Third,

the model uses linear relationships to calculate the diffusive and drainage pathways, including irrigation-related drainage, which is difficult to simulate accurately. The overestimation of night-time $NH_3$ at the GRAMINAE site by approximately a factor of two suggests that the diffusive fluxes were not well represented.

  Fourth, the relative fraction of techniques used in fertilizer application worldwide are assumed to be dependent on country income level as shown in Table A1. Wealthy countries are assumed to have better application techniques that result in less

$NH_3$ emissions, compared with less developed countries which tend to use simpler application methods. Such assumption is based on expert judgement due to the lack of statistical data so can introduce uncertainty. There are also sub-national-level variations in techniques used, which can affect the weighted sum and lead to different estimates. Application of urease inhibitor was also not included in the current version of AMCLIM due to insufficient statistical data, which may result in an overestimation of $NH_3$ emissions from urea application. However, this impact is limited because very few countries have

regulations on the use of urease inhibitor. Future work on incorporating existing regulations is possible once there is sufficient data.

  A systematic calculation of uncertainty associated with a process-based model can be very complicated. Instead, considering all the factors discussed above, a simple analysis for estimating the uncertainty of AMCLIM was performed. For ammonium fertilizer, the estimated uncertainty is suggested to be 33 % that was derived from the simulation of

GRAMINAE. For urea application, the uncertainty was estimated to be 20 % based on the multi-site comparison (as shown

in Fig. 13). As a result, the overall expected uncertainty of emissions from synthetic fertilizer use is 3.9 Tg N yr$^{-1}$ in 2010 and 4.3 Tg N yr$^{-1}$ in 2018, which accounts for 26 % of the emission in each year. It is worth noting that readers should only interpret the estimates and the uncertainty under the context of modelling.

The AMCLIM model only simulates $NH_3$ volatilization and does not include a bi-directional exchange scheme for $NH_3$, which may overestimate the $NH_3$ flux when there is enhanced deposition, especially for areas close to agricultural and semi-agricultural land. Another limitation of AMCLIM is the N uptake scheme. The current scheme for N uptake by crops has limited interactions with the carbon cycle, and the crop dynamics are represented by fixed empirical parameters that only account for temperature effects. There is no consideration of water stress constraining the uptake. Microbial N uptake is also not included in AMCLIM. However, these points are considered beyond the scope of this study due to the complexity 965 involved in simulating the relevant processes.

It must be acknowledged that the detailed evaluation of AMCLIM focused on the GRAMINAE dataset, where all needed inputs and parameters were available, as well as clearly quantified uncertainties in the measured fluxes. Other possible evaluation datasets that were accessible to the authors either had key gaps in data following fertilizer application or major measurement uncertainties. For this reason, the wider evaluation took an approach of comparing with multiple studies where 970 average flux estimates are available, but where the datasets are not sufficient for a detailed, hourly comparison with the model. This emphasizes the need to establish more high-quality and accessible ammonia flux datasets following fertilizer application that can be used for model development and evaluation. Considering the needs for detailed model testing, such measurements should meet the following requirements:

1) Information of the field site (coordinates, basic climatic and soil conditions).

2) Meteorological variables that are measured at high frequency and reported with high temporal resolution (ideally sub-hourly, e.g., 15 mins), including air temperature and wind speed at a reference height, atmospheric pressure, precipitation and humidity. Radiation and heat flux measurements are also very useful, especially to establish canopy temperature, humidity and wetness..

3) Soil temperature and soil moisture measured at a specified depth (better to have measurements at multiple depths) 980 with the same measured frequency as the meteorological variables.

4) Soil textures, bulk density and pH. If possible, the soil pH should be measured continuously or periodically following urea application.

5) Above-canopy fluxes of ammonia and atmospheric concentration of $NH_3$ at a reference height above the canopy (e.g. 1 m), together with reporting of uncertainties (especially where uncertainties vary over time), with clear 985 information on the $NH_3$, measurement method and the flux method used, including any assumptions.

6) Description of the field site, including estimates of surface displacement height, roughness height and single sided leaf area index (LAI, if vegetation presented).

7) Record of human management practices, such as fertilization information (date, time, amount and technique) and irrigation.

8) If possible, information on the temporal changes in atmospheric concentrations of other air pollutants, especially acidic compounds such as sulfur dioxide and nitric acid.

9) (Optional) Hydraulic conductivity and cation exchange capacity.

It is critical to avoid large gaps of measurement data, especially at key periods (such as immediately after fertilization events) if measurement data are to be used for detailed comparison with a model application.

**5 Conclusions**

This study presents the development and operation of AMCLIM–Land, a module in AMCLIM designed to simulate $NH_3$ emissions from synthetic fertilizer use at both the site scale and global scales. AMCLIM–Land simulates physical, chemical and biological processes in the soils and at the land surface that control $NH_3$ volatilization. It incorporates the effects of environmental conditions and important management practices on these processes. AMCLIM–Land employs a four-layer
soil structure, allowing for a detailed simulation of the soil processes and evaluation of various application techniques. Besides $NH_3$ volatilization, AMCLIM–Land also models other important N pathways in the agricultural systems, including surface runoff, nitrification, crop uptake and dissolved N to deep soils via leaching and diffusion. Although they are not the prime focus of the present modelling approach, these losses are relevant as they affect the amount of ammoniacal nitrogen that is available at the soil surface for volatilization as ammonia.

AMCLIM–Land was tested at the site scale (one field campaign with detailed measurements) and then applied on the global scale. Review of literature showed that many $NH_3$ flux measurement datasets do not have the requisite information included to allow full application of the model. Nevertheless, a broader comparison of average volatilization rates was possible with a wide range of published literature (Fig. 13). The comparison demonstrates close agreement with measurements in the GRAMINAE experiment as well as broad agreement from the global comparison of mean volatilization
rates. AMCLIM–Land accurately captures the major features of $NH_3$ fluxes from a post-cutting grassland after fertilization in the GRAMINAE study. On the global scale, using AMCLIM–Land, it is estimated that $NH_3$ emissions from synthetic fertilizer use are 15.0±3.9 Tg N $yr^{-1}$ in 2010 and 16.8±4.3 Tg N $yr^{-1}$ in 2018, which account for 14.6±3.8 % and 13.9±3.6 % of the total N in synthetic fertilizers in each year. The spatial and temporal variations of $NH_3$ emissions are significant, with high emissions occurring in regions with intensive agricultural activities, such as China, India and US. Global $NH_3$ emissions
are dominated by East and South Asia, and North America. AMCLIM highlights key factors that tend to cause larger $NH_3$ emissions, including hot temperatures, low soil moisture, windy conditions and high soil pH. The highest $NH_3$ emissions occur in July during both simulated years, and the seasonality of emissions was largely driven by planting seasons and temperature. Summer (JJA) contributed to over half of the annual $NH_3$ emissions.

Based on simulations using AMCLIM–Land, less than 50 % of applied N is absorbed by crops (Fig. 9), and $NH_3$ volatilization to the air is one of the major pathways for N losses. Broadcasting is the most commonly used methods for fertilizer application, but it results in a large fraction of N being lost due to $NH_3$ emissions. By comparison, incorporation and deep placement are effective methods that can be employed to mitigate $NH_3$ emissions, and which can be treated within the modelling framework of AMCLIM.

AMCLIM–Land can be used as a valuable tool for exploring the impacts of different agricultural management practices on $NH_3$ emissions and N pathways, despite the necessary simplifications of complex processes. The main advantages of AMCLIM–Land include: 1) the model is based on understanding at process-level and includes the most important N pathways, 2) responses to environmental variables are included, 3) there are representations of local management, 4) the model performs simulations at high temporal resolution which provides more reliable estimates of $NH_3$ emissions which are strongly influenced by environmental conditions. Overall, AMCLIM–Land provides insights on how environmental conditions and changes in agricultural management can affect $NH_3$ emissions and the N pathways.

Future research should focus on improving the soil pH dynamics and a better representative of the diffusion and drainage in the AMCLIM model in order to provide more accurate estimate for $NH_3$ emissions. Incorporating a bi-directional exchange scheme for $NH_3$ could be a potential future task so the model can simulate interactions between the soil and the vegetation. More testing and comparisons against site scale measurements will also be helpful to reduce uncertainty and improve the model performance.

**Appendix**

**A1 Global fertilizer use in the 21$^{st}$ century**

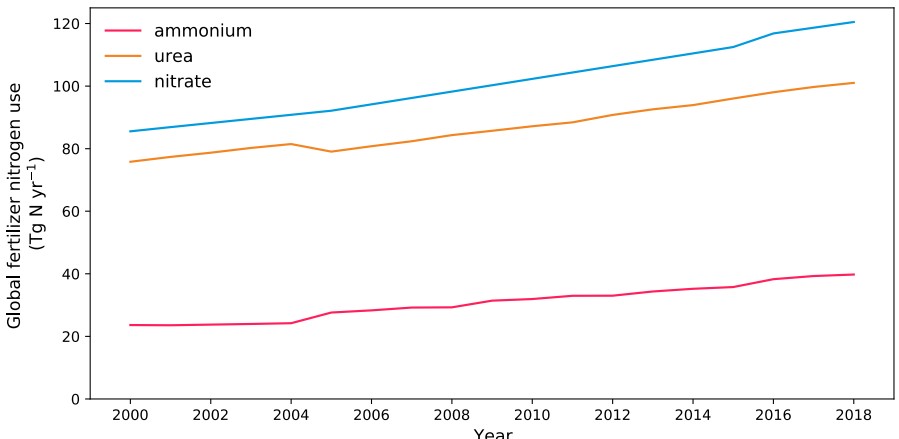

**Figure A1. Global total use of three types of fertilizers in the 21$^{st}$ century (IFA, 2021). The 16 major crops that are simulated by AMCLIM-Land are: 1) barley, 2) cassava, 3) cotton, 4) groundnut, 5) maize, 6) millet, 7) potato, 8) rapeseed, 9) rice, 10) rye, 11) sorghum, 12) soybean, 13) sugarbeet, 14) sunflower, 15), sugarcane and 16) wheat.**

## A2 Techniques used for chemical fertilizer application

**Table A1. Fraction of techniques used for chemical fertilizer application at country-level, based on the income classification (WB, 2022).**

|  | Broadcasting | Incorporation | Deep placement |
|---|---|---|---|
| High income | 0.7 | 0.2 | 0.1 |
| Upper middle income | 0.8 | 0.15 | 0.05 |
| Lower middle income | 0.95 | 0.05 | 0 |
| Low income | 1.0 | 0 | 0 |

## A3 Global simulations for year 2018

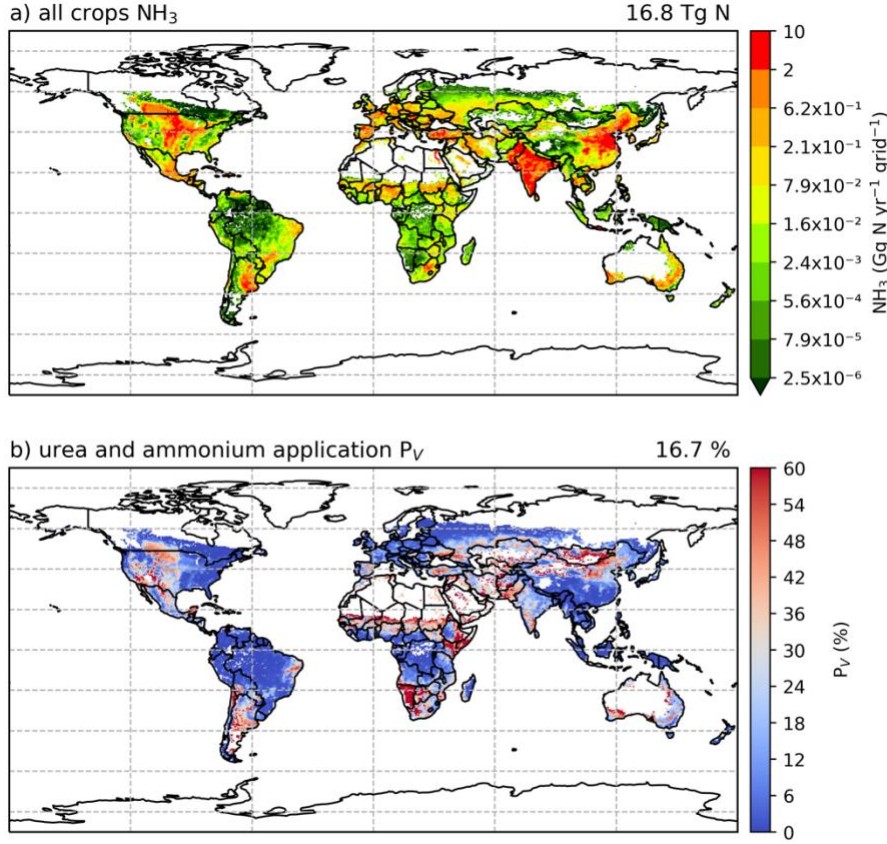

**Figure A2. Same as Fig. 7 but for 2018.**

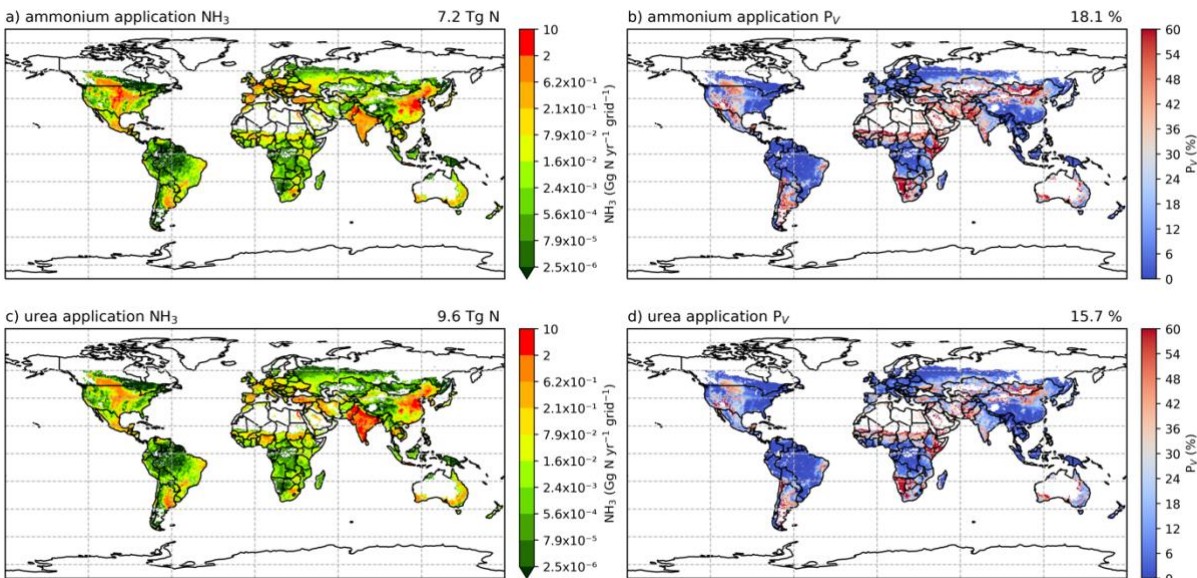


**Figure A3. Same as Fig. 8 but for 2018.**

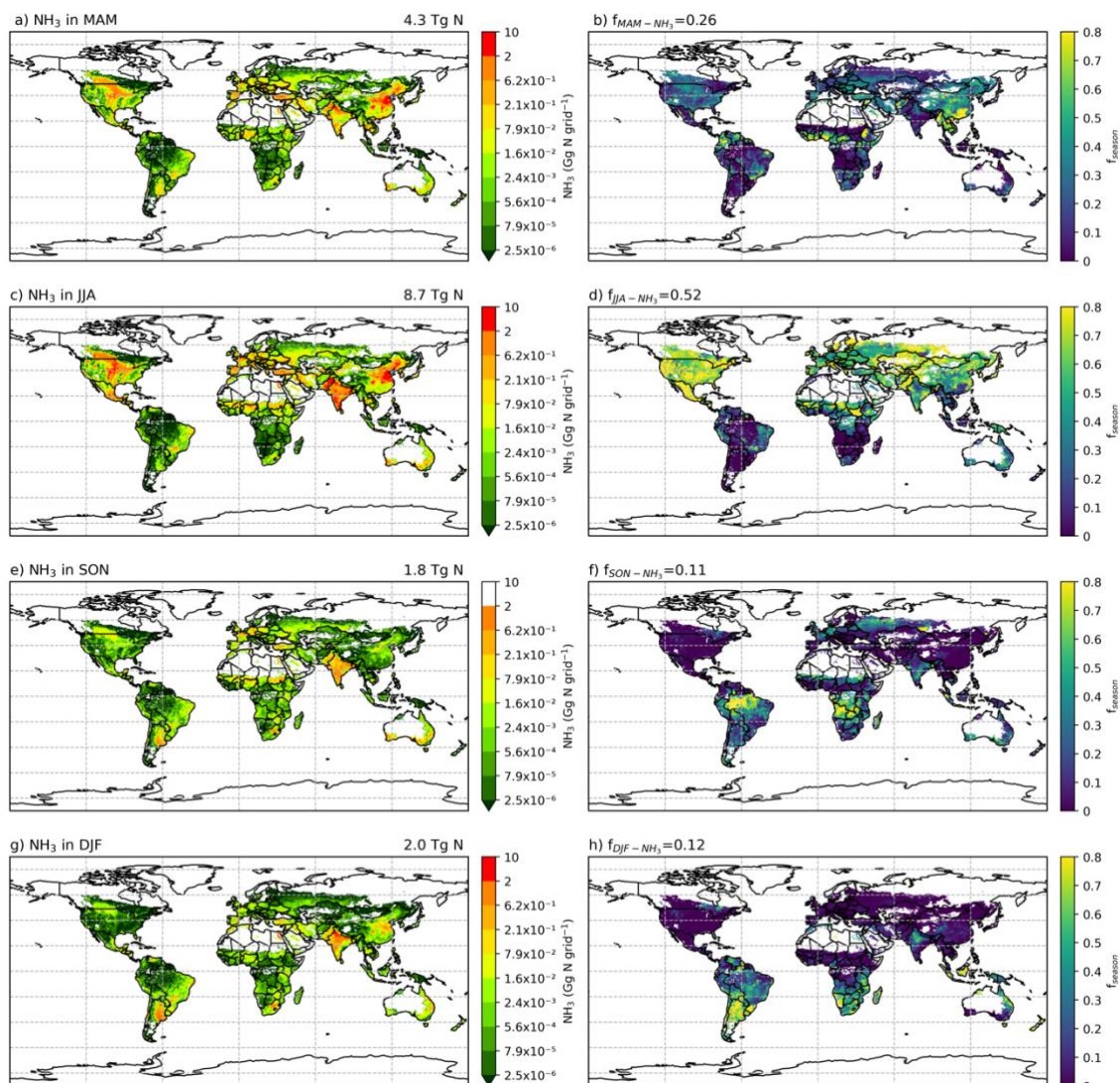

**Figure A4. Same as Fig. 10 but for 2018.**

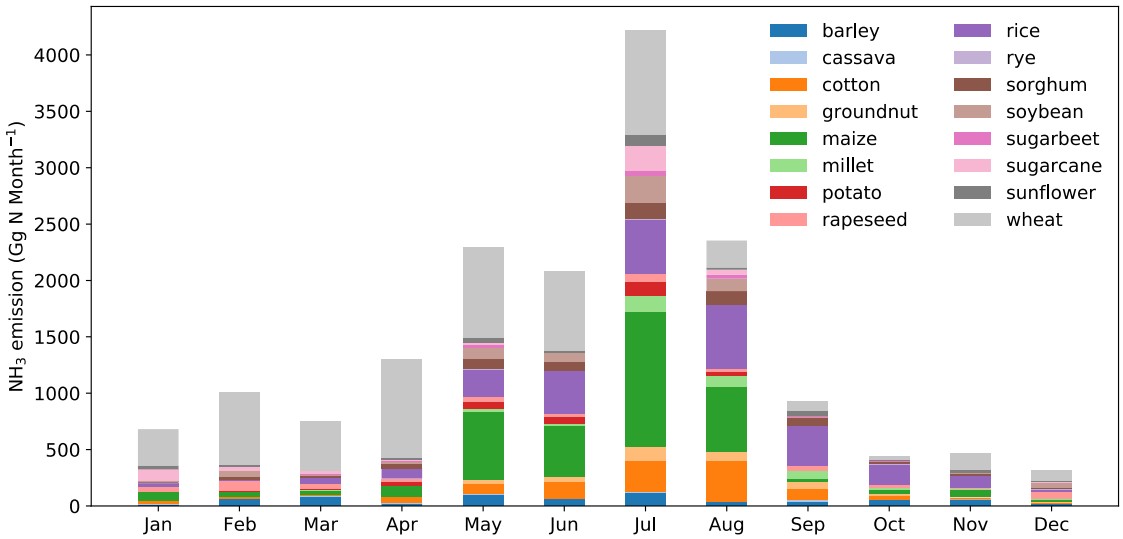


**Figure A5. Same as Fig. 11 but for 2018.**

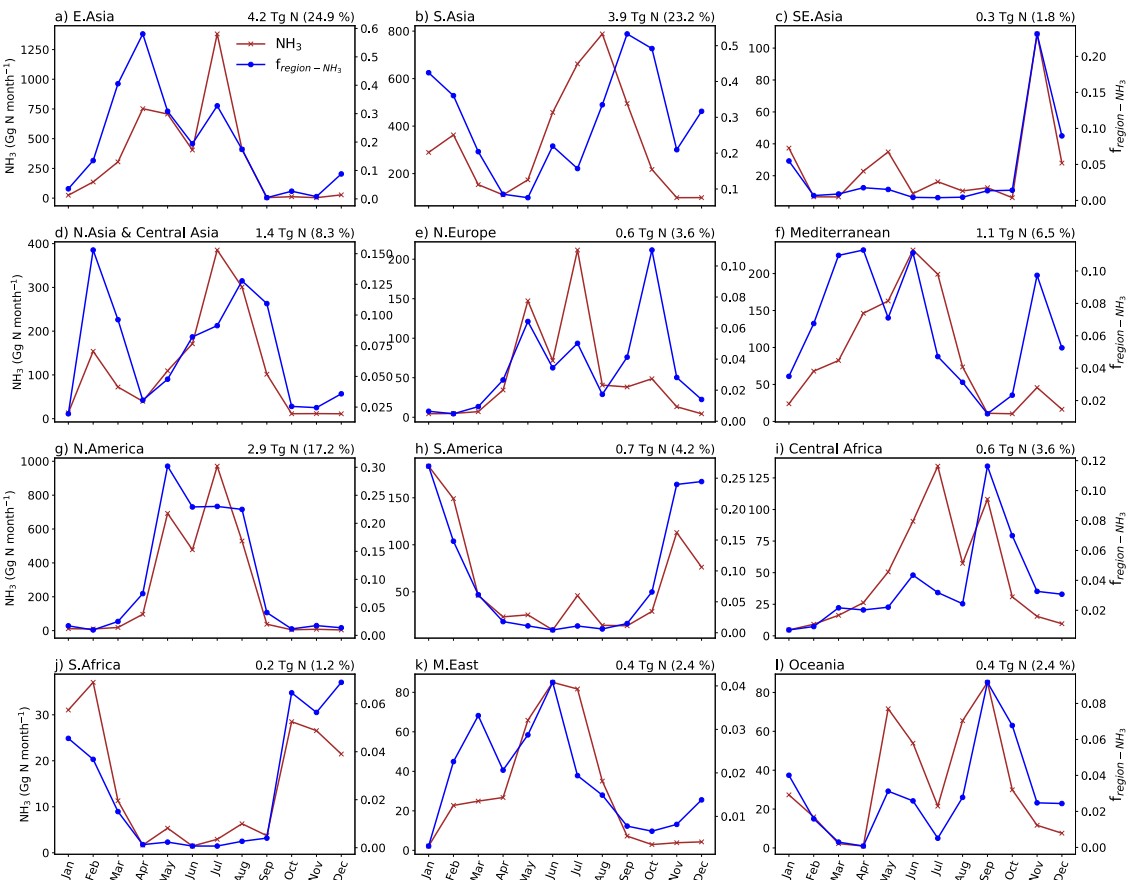

**Figure A6. Same as Fig. 12 but for 2018.**


## A4 Crop-specific NH₃ emissions

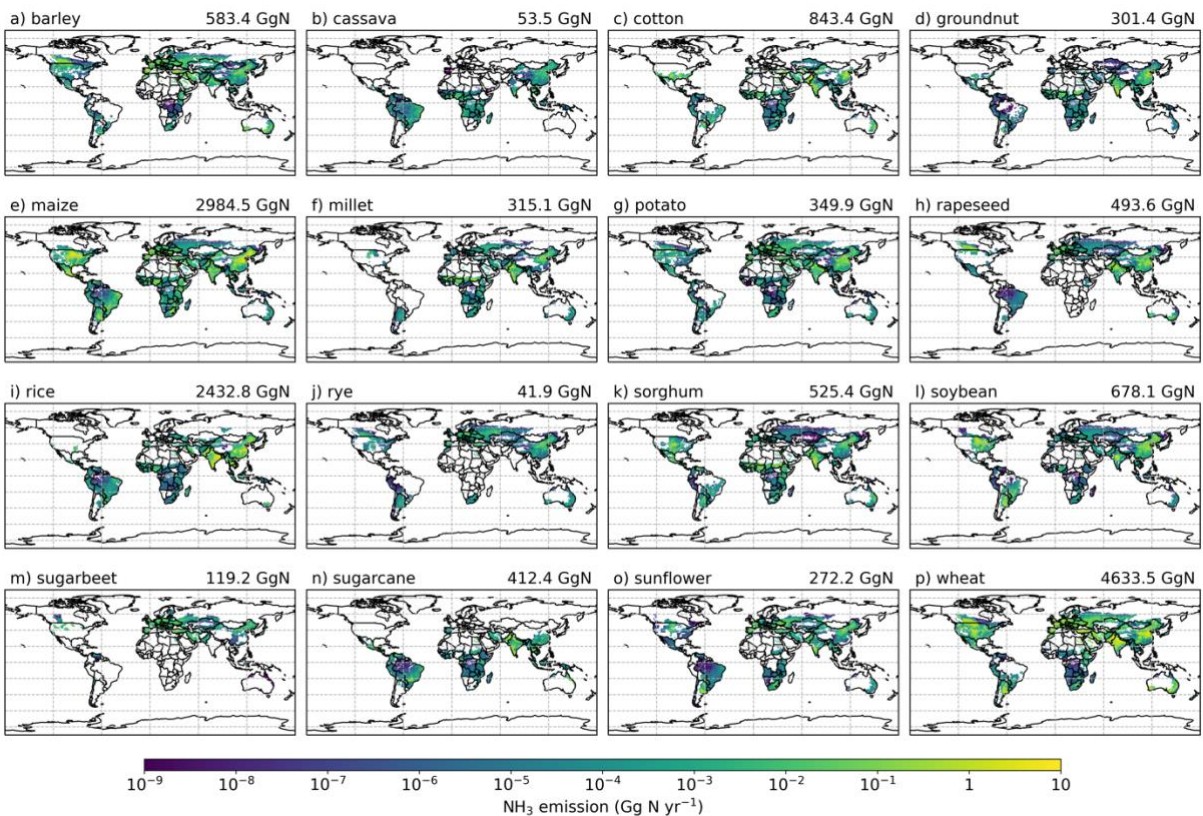

**Figure A7. Ammonia emissions (Gg N grid⁻¹) from 16 major crops for 2010 as simulated using AMCLIM.**

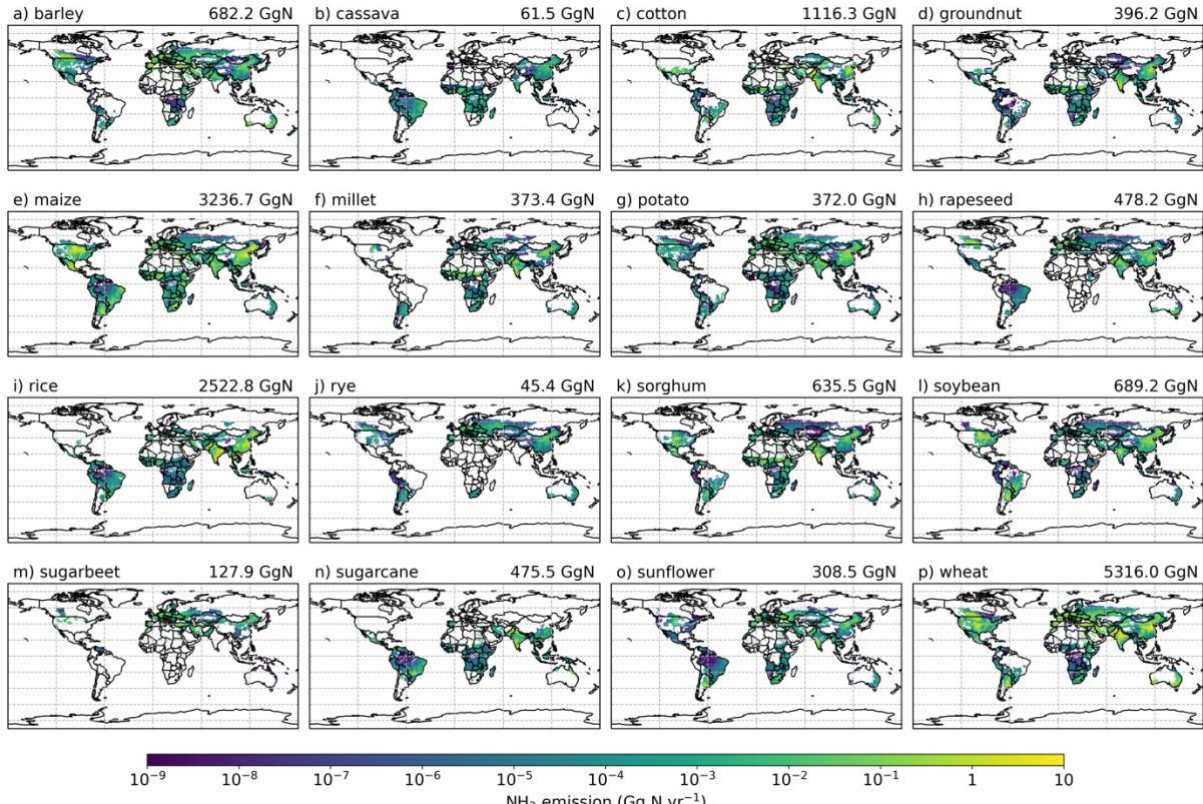

**Figure A8. Same as Fig. A7 but for 2018.**

## A5 Effects of application techniques

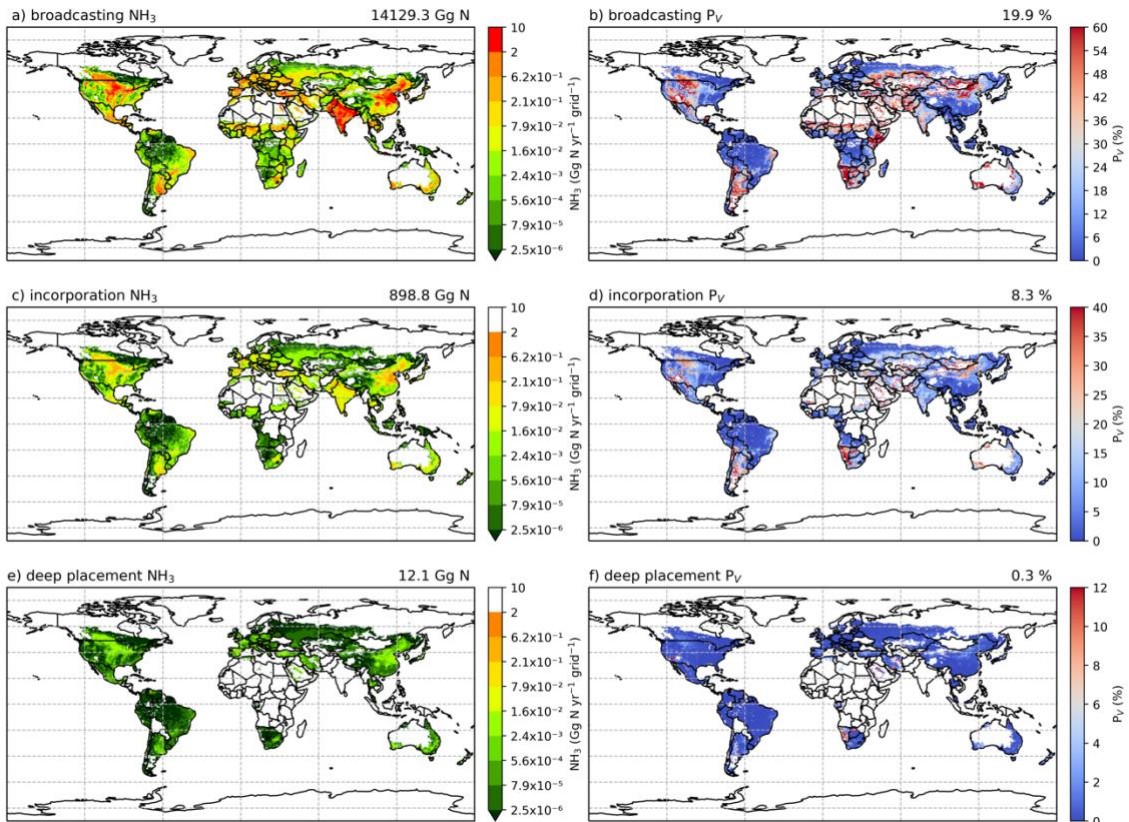

**Figure A9. Simulated NH₃ emissions (Gg N yr⁻¹) from synthetic fertilizer application by three techniques and the corresponding volatilization rates ($P_V$) in 2010. NH₃ emissions from (a) broadcasting, (c) incorporation and (e) deep placement. Percentage of applied N that volatilizes as NH₃ by (b) broadcasting, (d) incorporation and (f) deep placement.**

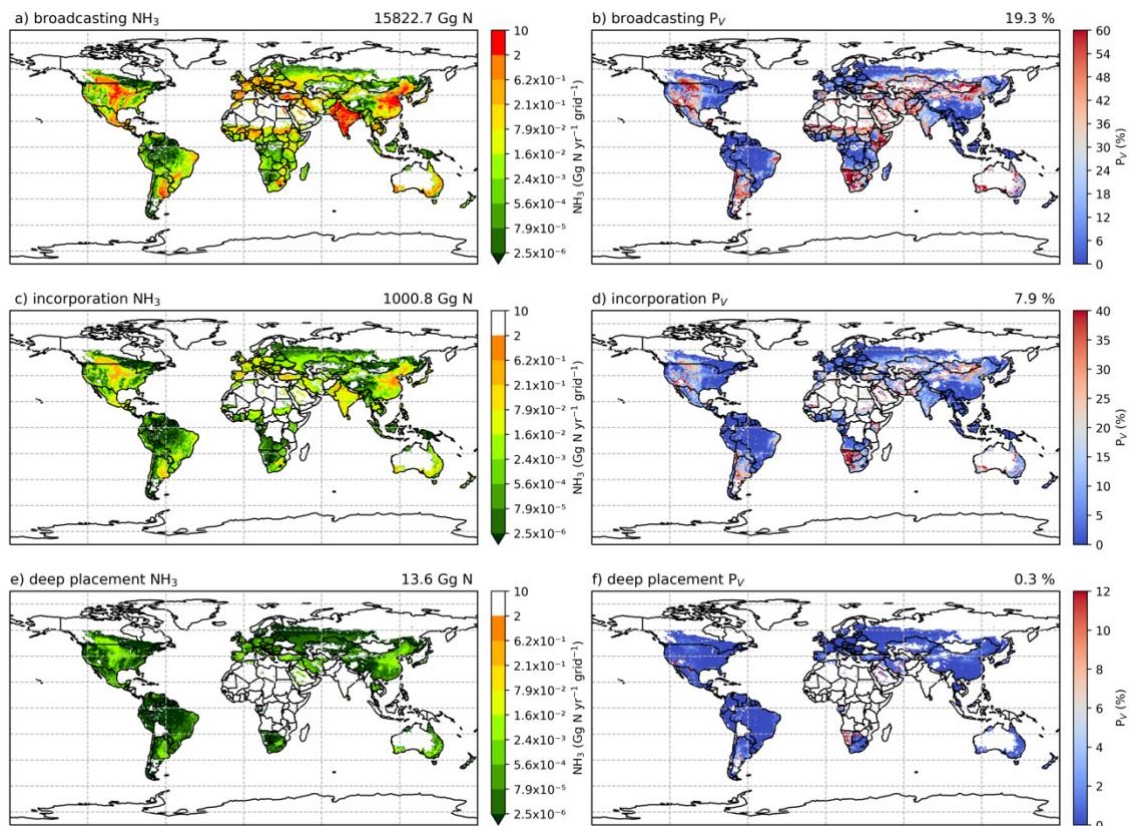

**Figure A10. Same as Fig. A9 but for 2018.**


 **A6 Geographical regions defined in AMCLIM**

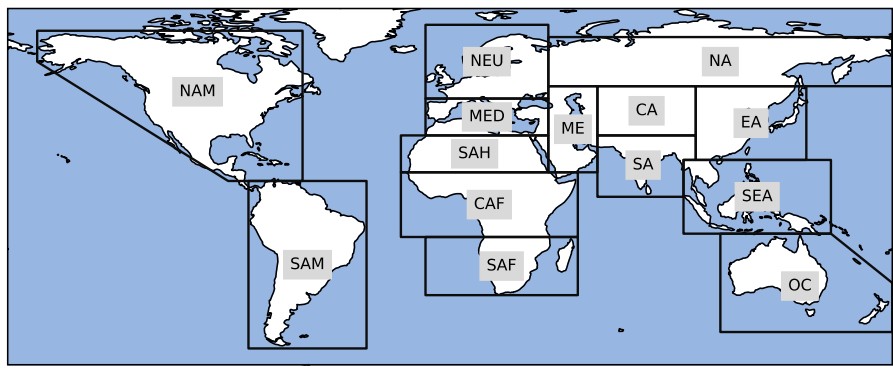

**Figure A11. Geographical regions (SREX scientific region) used in AMCLIM (Seneviratne et al., 2012).**

**A7 Global soil pH**

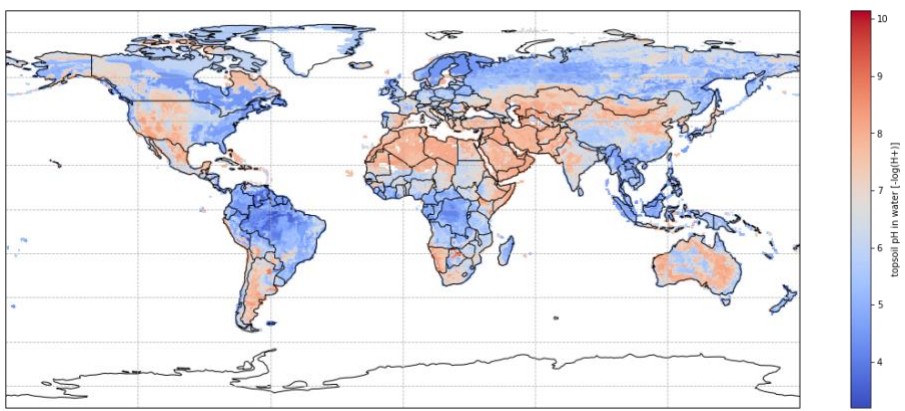

**Figure A12. Global soil pH. Data from HWSD v1.2 (Wieder et al., 2014).**

**A8 List of experimental studies used for model comparison**

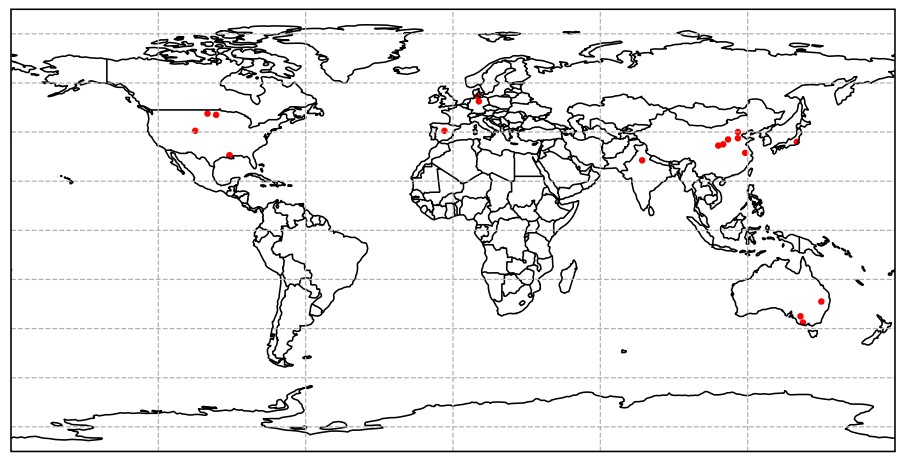

**Figure A13. Geographical locations of the measurement sites used for model comparison.**

**Table A2. Experimental studies used for model comparison, with information on fertilizer type, measured soil pH and techniques used for NH₃ measurements.**

| Study | Fertilizer | Soil pH | Technique |
|---|---|---|---|
| Turner et al. (2012) | Urea/AS | 7.8/7.5 | Micromet method |
| Schwenke et al. (2014) | Urea/AS | 8.1 | Passive flux method |
| Yang et al. (2011) | Urea | 8.5 | Micromet method |
| Huo et al. (2015) | Urea/AS | 8.2 | Micromet method |
| Li et al. (2015) | Urea | 8.0/7.7 | Dynamic chamber |
| Yang et al. (2020) | Urea | 7.7 | Ventilation method |
| Zhang et al. (2022) | Urea | 7.7 | Dynamic chamber |
| Ni et al. (2014) | Urea/CAN | 6.5 | Dynamic chamber |
| Datta et al. (2012) | Urea | 8.4 | Gas analyser |
| Cowan et al. (2021) | Urea | 8.0 | Static chamber |
| Bhatia et al. (2023) | Urea | 8.0 | Static chamber |
| Hayashi et al. (2008) | Urea | 5.7 | Wind tunnel |
| Sanz-Cobena et al. (2014) | Urea | 8.1 | Micromet method |
| Jantalia et al. (2012) | Urea | 7.8 | Semi-open chamber |

| Thapa et al. (2015) | Urea | 8.4 | Open chamber |
|---|---|---|---|
| Tian et al. (2015) | Urea | 6.2 | Passive and active closed chamber |
| Engel et al. (2017) | Urea | 6.3/6.3/7.3 | Gas samplers |
| Liu et al. (2017) | Urea | 6.0 | Dynamic chamber |

**A9 Global simulations for different fertilization scenarios**

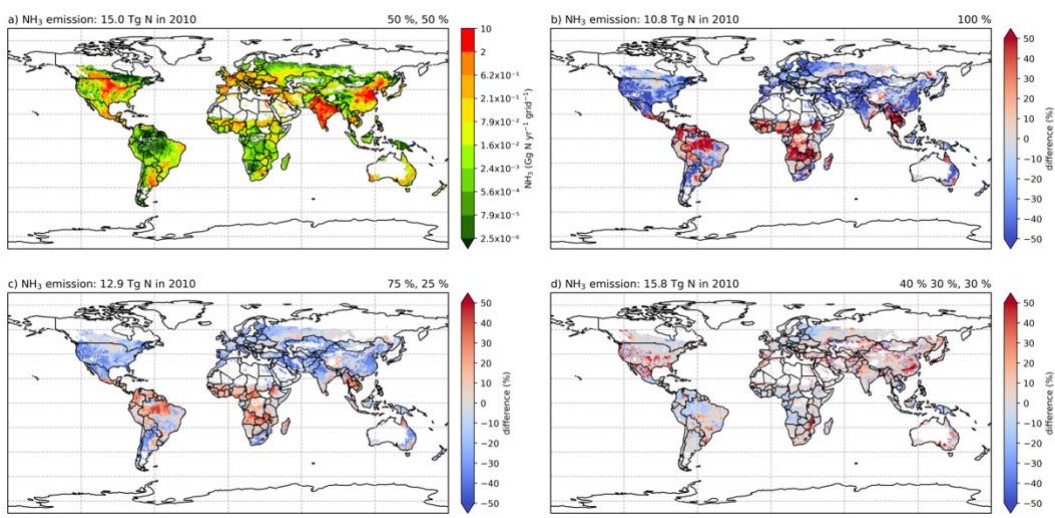

**Figure A14. Simulated (a) global NH₃ emissions (Gg N yr⁻¹ grid⁻¹) from synthetic fertilizer use in 2010 using the 50 %, 50 % fertilization scenario, and differences in NH₃ emissions (%) from simulations using (b) 100 % fertilization scenario, (c) 75 %, 25 % fertilization scenario, and (d) 40 %, 30 %, 30 % fertilization scenario.**


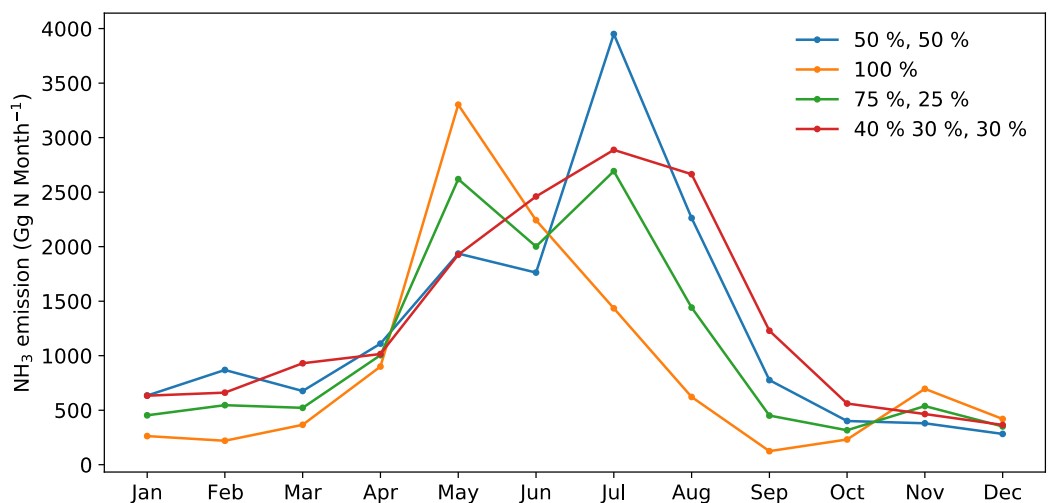

**Figure A15. Simulated global monthly NH₃ emissions (Gg N month⁻¹) from synthetic fertilizer use in 2010 using the four different fertilization scenarios.**

*Code and data availability*. Code of the model can be obtained at github (https://github.com/jjzwilliam/AMCLIM last access: 03 April 2024) and Zenodo (https://zenodo.org/records/10911886 last access: 03 April 2024). Model results presented in this study are in netCDF format and can be freely accessed from the Edinburgh DataShare (https://datashare.ed.ac.uk/handle/10283/8753, last access: 28 March 2024; Jiang et al., 2024).

*Supplementary Materials*. The Supplementary Materials related to this article has been submitted. A table of model variables and parameters is presented in the Supplementary Materials.

*Author contributions*. JJ, DSS and MAS designed the research. JJ developed the model, write the code, performed the simulations and wrote the paper. All authors contributed to analysis of the model outputs, interpretation of results and critical revision.

*Competing interests*. The authors declare that they have no conflict of interest.

*Acknowledgement.* Jize Jiang gratefully acknowledges support from University of Edinburgh, UK Centre for Ecology and Hydrology (UK CEH) and the UK national supercomputing ARCHER2. The authors are grateful for the support from the Global Environment Facility (GEF), through the UN Environment Programme (UNEP) for the project "Towards the International Nitrogen Management System (INMS)", and from the UKRI, under its Global Challenges Research Fund for support of the GCRF South Asian Nitrogen Hub (grant no. NE/S009019/2), and from NERC for National Capability support,

including through the CEH SUNRISE project. The authors sincerely thank two anonymous reviewers for their insightful and constructive comments and suggestions during the reviewing process. The authors thank Christoph Müller for being the topic editor.

*Financial support.* This research has been supported by the UK Natural Environment Research Council (grant no. NE/S009019/2).

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
