# Peer review of "A dynamical process-based model AMmonia-CLIMate v1.0 (AMCLIM v1.0) for quantifying global agricultural ammonia emissions – Part 1: Land module for simulating emissions from synthetic fertilizer use"

_EGUsphere, 2024_

## Author Comment (AC1)

**Response to Anonymous Referee #1**

**Overall response:** We would like to thank reviewer 1 for these insightful and useful comments. These help improve the manuscript. Here we outline the point-by-point responses below in blue, and the relevant figures are attached.

**Comment:** The manuscript reports a new process-based model for the emission of ammonia following fertiliser application, and then applies this on a global scale. The topic of agricultural ammonia emissions is an important one, and there is clearly a need for well-calibrated, process-based models as an aid to understanding the basic processes involved in ammonia volatilisation and as a means of upscaling to regional/global scales.

**Reply:** We appreciate that the reviewer recognizes the value of our study. We thank the reviewer for spending time reviewing the manuscript and the development of the AMCLIM model.

**Comment:** My understanding is that the AMCLIM model is a lightweight model that focuses on ammonia emissions in a period of 1-2 weeks following fertilisation, when the vast majority of ammonia is emitted. It uses a relatively simple Ohm's law like structure, which leads to most of the equations being linear. Non-linear contributions to N cycling, for example due to microbial activity, coupling between C and N and mineralisation of organic N compounds are ignored as being unimportant for predicting ammonia emissions shortly after fertilisation. Soil moisture and soil temperature are not explicitly modelled, but rely on measurement data. The relative simplicity makes the model a potentially valuable tool for experimental groups carrying out ammonia emission measurements. As compared to more detailed process based models (e.g. Daycent or DNDC type models) I assume the model is considerably easier to set-up, faster to run and doesn't require long spin up periods. Furthermore, the comparison to the GRAMINAE site shows that the model does a good job of capturing the variation in ammonia emissions in the days following fertilisation.

On the other hand, I think the application of the AMCLIM model to global ammonia emissions from croplands is premature. Insufficient evidence is provided to show that the model is well calibrated. The comparison to a single grassland site in Germany suggests that the model shows promise in capturing the diurnal cycle of ammonia emissions following fertilisation. However, before applying the model to global croplands I would like to see (see also calibration comments below):

**Reply:** The calibration process of a model aims to improve the model results by tuning the model parameters to better represent particular conditions. In this study, we performed a detailed timeseries comparison between measured and modelled $NH_3$

emissions at a high temporal resolution of 15 mins (Fig. 4), and adequate amount of parameter tuning and testing of model complexity have been done and discussed (as shown in Fig. 5). Meanwhile, we have done a multi-site model-measurement comparison (Fig. 12 and 13). More detailed responses are given at the reviewer's calibration comments section.

**Comment:** 1. Improved evidence that the pH dependence of ammonia emissions is well represented. Figure 13a,b provides some information in this direction, but it is hard to conclude from this that the model is well calibrated (for example a factor 2 difference in the y axis scale is needed to show the modelled Pv values as compared to the measured values).

**Reply:** As shown in Fig. 13a, b, volatilization rates were plotted against soil pH. The volatilization rates generally increase towards higher pH, which is reflected by both measurements (Fig. 13a) and the model (Fig. 13b). We kindly remind the reviewer that the comparisons between measured and modelled volatilization rates are shown in Fig. 12 (not Fig. 13). Overall (in the original Fig. 12), 18 out of 26 and 3 out of 4 modelled $P_V$ are within a factor of 2 for simulations of urea application and ammonium application, respectively.

**Comment:** 2. Evidence that the pH change and the impact this has on ammonia emissions following urea application is well calibrated.

**Reply:** We agree that pH is a critical factor that influences the $NH_3$ volatilization, especially for urea application. We are aware of the soil pH increase after urea application, which has been discussed in Section 2.2.1 under *Soil pH scheme in AMCLIM-Land* (line 284 to 299). Our soil pH dynamics follow a simple scheme, which is developed based on Chantigny et al. (2004), and Móring et al. (2016), as now shown in Fig. R1-1.

[Figure]

**Figure R1-1 Top panel: soil pH change after slurry application (Chantigny et al. (2004)) Mid panel: soil pH change after cattle urine deposition (Móring et al. (2016)) Bottom panel: Soil pH scheme used in AMCLIM–Land. Changes of soil pH for 192 hours (8 d) after urea application for soils with initial pH of six different values.**

Since there are no available datasets of NH₃ emission measurement following urea application that have sufficient input to drive AMCLIM, we conducted a multi-site

comparison between model results and measurements across the globe, as shown in Fig. 12a.

**Comment:** 3. Evidence that the total ammonia emissions are well represented across multiple crop, soil and climate conditions. For example, that the model can capture ammonia emissions from paddy rice fields in South East Asia. Figure 12 goes some way in this direction, but I find it hard to conclude from this figure that the model is performing well across multiple conditions.

**Reply:** We would like to once again address that there is lack of suitable measurement datasets that can be used for a detailed model-measurement comparison like what we did with the GRAMINAE data. Therefore, we conducted the multi-site comparison (Fig.12). These studies used for comparison were from 12 sites in seven different countries across the globe. We have now further improved the figure to show their climatic conditions and the geographic locations, with several additional studies from the US added. The updated figure shows that the selected sites for comparison have a reasonable spatial distribution for various climatic conditions (representing by average temperature from 8.5 to 31.7 degC) and soil conditions (representation by soil pH between a range from 5.7 to 8.5), with five major crops being examined (wheat, rice, maize, sunflower and cotton). Overall (in the updated Fig. 12), 20 out of 33 (originally 18 out of 26) and 3 out of 4 modelled $P_V$ are within a factor of 2 for simulations of urea application and ammonium application, respectively. One aspect that needs to be remembered is that measurements also have uncertainties, especially when using enclosure methods to measure $NH_3$ emissions (Kamp et al., 2024), with the finding that chambers may substantially underestimate emissions in some circumstances (e.g. when dealing with short vegetation and limited mixing in chambers). This points to the need for the future for more exact information being provided by experimentalists than has often been the case in the existing literature.

[Figure]

**Updated Figure 12. Modelled percentage volatilization rates ($P_V$, %) compared with experimental studies (Hayashi et al., 2008; Sanz665 Cobena et al., 2008; Yang et al., 2011; Datta et al., 2012; Jantalia et al., 2012; Turner et al., 2012; Ni et al., 2014; Schwenke et al., 2014; Huo et al., 2015; Li et al., 2015; Thapa et al., 2015; Tian et al., 2015; Engel et al., 2017; Liu et al., 2017; Yang et al., 2020; Cowan et al., 2021; Zhang et al., 2022; Bhatia et al., 2023). Measurement data were from literature that studied $NH_3$ volatilization from (a) urea application and (b) ammonium fertilizer application to field. Black dashed line is the 1:1 line, and grey dotted lines indicates the values within a factor of 2. Colours represent average temperature during the measurement periods.**

[Figure]

**Geographical distributions of the measurement sites used for model comparison.**

At the same time, the development of the AMCLIM model is based on the understanding at process level, and we tried not to "invent" parameters, of which the only purpose is to improve model performance. We think it is debatable that a model should be intensively calibrated (or this should be the only method), especially when the uncertain parameters do not physically (chemically and biologically) make sense. This hampers the identification of limitations in current models.

**Comment:** My opinion is that the model description and application to the GRAMINAE site would already make a valuable paper. I would suggest to leave out the application to global croplands and publish this at a later date, once more extensive calibration can be performed using site-scale data.

**Reply:** We thank the review for recognizing the value of the development of AMCLIM and the site application of the model. We try our best to address reviewer's comments and improve the manuscript, and wish to include the global simulations in this manuscript and its revised version.

It worth mentioning that one of the main goals of this study (and the following papers/manuscripts) is to provide global estimates of agricultural $NH_3$. Ideally, we would like to have more good quality measurement datasets that can be used to evaluate the AMCLIM model. Unfortunately, the GRAMINAE study appears to be the only one that is suitable. We admit that it is a limitation that AMCLIM was only intensively evaluated against one single measurement dataset, therefore the multi-site comparison is provided as complimentary evidence to demonstrate the capability of

the model. We think the global application is useful, and not having a lot of measurements for intensive comparison/evaluation should not be a reason not to apply the model to larger scales. It could take decades until there are multiple comparable datasets of the quality of the GRAMINAE experiment, and it would be unreasonable to prevent publication until such time.

We would like to also use this manuscript to address the importance of good quality measurements, and raise the awareness that it would be more helpful for process interpretation, model development and evaluation if future studies on measuring $NH_3$ emissions can have well-documented measured variables. Since nowadays "code availability" is a requirement for modelling studies, sharing the full sets of measurements should also be encouraged. In the revised manuscript we propose to outline a list of requirements for reporting of future measurements of agricultural $NH_3$ emissions:

1) Information of the field site (coordinates, basic climatic and soil conditions)
2) Meteorological variables that are measured at high frequency and reported with high temporal resolution (ideally sub-hourly, e.g., 15 mins), including air temperature and wind speed at a reference height, atmospheric pressure, precipitation and humidity. Radiation and heat flux measurements are also very useful.
3) Soil temperature and soil moisture measured at a specified depth (better to have measurements at multiple depths) with the same measured frequency as the meteorological variables.
4) Soil textures, bulk density and pH. If possible, the soil pH should be measured continuously or periodically following urea application.
5) Above-canopy fluxes of ammonia and atmospheric concentration of $NH_3$ at a reference height above the canopy (e.g. 1 m), together with reporting of uncertainties (especially where uncertainties vary over time), with clear information on the $NH_3$, measurement method and the flux method used, including any assumptions.
6) Description of the field site, including estimates of surface displacement height, roughness height and single sided leaf area index (LAI, if vegetation presented).
7) Record of human management practices, such as fertilization information (date, time, amount and technique) and irrigation.
8) (Optional) Hydraulic conductivity and cation exchange capacity.

It is critical to avoid large gaps of measurement data, especially at key periods if measurement data are to be used for detailed comparison with a model application.

**Comment:** More detailed questions/comments follow below:

** Model details **

The AMCLIM model includes N uptake by plants, but ignores N uptake by microbes (immobilisation). This may be an important process, especially for fertilisation events at or before planting, when plant N uptake is low. Please discuss why/under what circumstances it is reasonable to ignore microbial N uptake.

**Reply:** Immobilisation or microbial N uptake is a competing process against plant N uptake and is considered to be primarily regulated by available C in soils and gross ammonification ((Butterbach-Bahl et al., 2011). It remains uncertain that how microbial N uptake affects $NH_3$ emissions. Perhaps the clearest indication is provided by manipulation of ecosystems, such as by using nitrification inhibitors, the use of which has tended to lead to a small increase in $NH_3$ emissions, by increasing the lifetime of surface $NH_4^+$ pools (Snyder et al. 2009).

AMCLIM does not simulate soil C dynamics, and the explicit incorporation of microbial activities is beyond the scope of this study. In contrast, we used a simple N uptake scheme to estimate plant N uptake, which is used as an indicator to evaluate the nitrogen use efficiency. We propose to modify the manuscript to clarify that microbial N uptake is not simulated.

**Comment:** To what extent has the sensitivity of the model to changes in temporal and spatial resolution been tested? Ideally the spatial and temporal resolution is reduced until the ammonia emissions become relatively independent of further decreases (and I see no reason why the resolution cannot be made finer than the meteorological and soil inputs, e.g. by splitting each soil layer into sub-layers). Figure 5 suggests that changing the spatial resolution of the top soil layer leads to large changes in the model behaviour (comparing circles for z1=1,2,3 cm), and thus that the model behaviour has not yet converged.

**Reply:** We have tested the model performance at various temporal resolutions, including time-steps at 15 mins, 1 hour, 3 hours, 6 hours, 12 hours and 24 hours (proposed new Fig. R1-2). By decreasing the temporal resolution from 15 mins to 6 hours, the model was still able to capture the main temporal variations in fluxes and to reproduce the peak emissions of each day, while giving a reasonable estimate of cumulative $NH_3$ emissions. However, when the temporal resolution decreased to 8 hours and even less, the model started to underestimate $NH_3$ emissions and was not capable of reproducing the emission peaks. This is also the reason why global simulations were performed at an hourly time step.

[Figure]

**Figure R1-2 Comparisons between measured and modelled NH$_3$ emissions with modelling time-steps varied from 1min to 24h. Simulated cumulative NH$_3$ fluxes were 0.51 g m$^{-2}$ for 1min time-step, 0.36 g m$^{-2}$ for 3h time-step, 0.34 g m$^{-2}$ for 6h time-step, 0.25 g m$^{-2}$ for 8h time-step, 0.24 g m$^{-2}$ for 12h time-step and 0.18 g m$^{-2}$ for 24h time-step.**

The thickness of the top soil layer refers to the assumption of where the fertilizer is applied (by broadcasting, or surface spreading) for technical feasibility, and we found out that a 2cm layer is a reasonable assumption, which also gave a good model result when comparing model and measurements. Figure 5 shows that both increasing the thickness (to 3cm) and decreasing the thickness (to 1cm) of the top soil layer resulted in poorer model performances compared with a 2cm top soil layer setting. We tried not to make the soil layering over complicated for AMCLIM. Pursuing a model convergence behaviour is not practical given the fact that the overall performance of current model settings is reasonable.

**Comment:** I would expect the time-step to be important as the underlying processes have very different response times. In particular the chemical equilibrium reaction between NH3 and NH4+ is much faster than plant N uptake or nitrification. As such I think it is important to have some time-step control, especially in the minutes following broadcast fertilisation (is 15mins / 1hour short enough?). Similarly, the spatial scale will control the interaction between the concentration of NH3 and NH4+ in the top soil layer following broadcast fertilisation and the transport processes (is 4 soil layers enough?).

**Reply:** The GRAMINAE campaign has demonstrated that a 15 min time interval is short enough as the sub-hourly variations in NH$_3$ fluxes have been well captured by the measurements. Meanwhile, the meteorological inputs that drive the AMCLIM model have a temporal resolution of 15 mins. By simply increasing the temporal resolution of simulations (reduce the time-step) without higher resolution input, the model results are insignificantly different, but the computational costs will increase enormously (running the model at 1 min time-step but use the same meteorology for a 15 mins window leads to only a 4.1% difference, which does not justify the substantial increase in computational costs).

Regarding the soil profile, the most critical layer is the top soil layer where NH$_3$ volatilization takes place. As the main goal of this study is to model NH$_3$ emissions rather than simulating the soil C dynamics or N$_2$O fluxes, having a more detailed multi-layering of the vertical soil profile is not as important as getting a suitable value for the top model layer thickness.

**Comment:** As a related point, it would be useful to briefly mention how the coupled differential equations are solved. Is this by a Euler method or is a higher order method used?

**Reply:** The prognostics in AMCLIM are solved by the Euler method at each time step. We have added this point to the manuscript.

**Comment:** I think the section 'volatilisation of NH3' could be improved, in particular the description of how the surface NH3 concentration is calculated.

**Reply:** The calculation of the surface $NH_3$ concentration was explained in detail in the Supplementary material see Section S6. We have improved the manuscript accordingly.

**Comment:** It would also be useful to provide the recovery function for soil pH following urea fertilisation.

**Reply:** We have now added the equation for soil pH following urea application to land and a corresponding figure (Fig. R1-1).

**Comment:** For the plant N uptake, what values are used for W_r,i (SM14 and SM17)? I couldn't find this in the supplementary information. Also it would be useful to mention how perennial crops such as grass are treated, especially since this is relevant for the GRAMINAE site (the stages in Table S1 seem to be for annual crops).

**Reply:** The values of *W_r,i* were taken from Thornley et al. (1991), which are equivalent to 20, 40, 60, 80 g m$^{-2}$. While the results from the site testing are appropriate for cut grasslands and other agricultural crops, we focus the global upscaling in this study on arable crops. (Note that global upscaling of grazed grassland emissions is included in the second part the AMCLIM model for livestock, to be presented in a forthcoming manuscript). Considering cut/fertilized grassland, we note that there is no information from the GGCMI3 dataset on the spatial distribution of fertilizer amounts, which is much less well constrained than for arable crops. In the revised manuscript, we therefore highlight this gap in knowledge as an important research need.

**Comment:** ** Calibration **

As far as I understand, model parameters are taken from the literature, and are mostly not calibrated by comparing the AMCLIM model to measurements (a small number of model variations are shown in Figure 5, but are only compared to a single 10-day measurement at 1 site). Compare, for example, to Gurung et al. Nutr Cycl Agroecosyst

(2021) 119:259–273, where Bayesian techniques are used to perform a joint calibration of the 18 parameters relevant to ammonia volatilisation, comparing different levels of model complexity and by taking into account 8 different experimental sites with 42 site-year treatments. I understand that the authors cannot do everything in one paper, and am not expecting them to perform a full Bayesian calibration in this manuscript. However, I think that a comparable level of calibration is necessary before applying the model at a regional/global scale.

**Reply:** We thank the reviewer for pointing out the study by Gurung et al. 2021, which is an interesting study.

In the ACLIM model, there are three/four types of parameters.

1. Parameters that are measured or derived from lab and field experiments, e.g., Henry's law constant and dissociation constant.
2. Parameters from well-established theory and are widely used in sophisticated models, e.g., aerodynamic and boundary layer resistance for constraining the fluxes.
3. Parameters that are empirically derived or taken from other process-based models, e.g., coefficients for nitrification, coefficients for tortuosity correction that affect diffusion, coefficients for plant N uptake.
4. Parameters for AMCLIM setup, e.g., vertical soil layering, background $NH_3$ concentrations.

Tuning and testing was done for the last two categories. The model performances were then evaluated based on three statistics: the correlation coefficient, the error and deviation, as shown in the Taylor diagram Fig. 5. We have tested the model performance with different levels of complexity (Fig. 5), e.g., excluding drainage, runoff or nitrification. The current version of the model was found to be the optimal.

Gurung et al. (2021) developed an $NH_3$ emission module and implemented into the DayCent model. The new module is calibrated using a Bayesian method against measurements at several US sites. This is a novel study as it proposed a new $NH_3$ emission module (for urea application) for the DayCent model, in which the $NH_3$ volatilization was originally poorly represented. However, we noticed a few limitations:

1) There is no comparison for timeseries between the model results and measurements, given the fact that temporal variation is an important feature of $NH_3$ fluxes. To what extent the DayCent model is able to reproduce daily and diurnal variabilities in the $NH_3$ emissions remains unknown. In contrast, AMCLIM aims to replicate the temporal variations in $NH_3$ fluxes as well as to estimate the cumulative N loss due to $NH_3$ emissions.

2) DayCent is run at daily timestep. This further raised the question of excluding the temporal variations in $NH_3$ fluxes. The reviewer previously challenged the time step, questioning whether 15mins/1hour time-step is short enough. However, there is no evidence provided in Gurung et al. (2021), which could "fail" to address the reviewer's concern regarding the temporal solution based on the same reviewing standard.
3) Although Gurung et al. (2021) said input variables were reported in the publication, the reference where the model parameters were taken from are not provided and it is very difficult to trace back (as in Methods and Materials section and in Table 3). Many parameters used are empirically determined to formulate the processes. Since these parameters are not taken from well-established theory or process-based, the range of these parameters that was used as priori for the Bayesian calibration were mostly arbitrarily selected.
4) The comparisons were only for 8 US sites, possibly because the parameters are mainly applicable for US conditions. To what extent this can well represent other places (as the reviewer addressed, "across multiple crops, soil and climate conditions") is uncertain. The comparisons do not differentiate between sites, and it is difficult to assess which simulations perform better or worse.
5) It would be welcome if the underpinning measurement dataset used by Gurung et al. (2021) were to be made publicly available. It would provide a good opportunity for other modellers to benefit from the measurement effort summarized.

We emphasize that the focus of the present study is different from Gurung et al. (2021). We recognize that the two studies reflect two different modelling "philosophy", and it is difficult (and probably not possible) to say which is better. We definitely agree with the reviewer that the authors cannot do everything in one study, so we think it is crucial to fairly acknowledge the advantage and limitations of each study, which also reflects the rigor and inclusiveness of our science community.

**Comment:** I would find it useful to have a table of all model parameters and their values (e.g. in the supplementary material).

**Reply:** We agree and will add a table of all model parameters and their values in the supplementary material.

**Comment:** ** Global simulations **

I mentioned above that I believe the application to ammonia emissions from global croplands is premature, and requires additional calibration of the model. However, if

the authors choose to retain the global simulations in the manuscript, it would be useful to address the following points:

As discussed in the manuscript, correct fertiliser timing is important, due to the sensitivity of ammonia emissions to meteorological conditions (especially temperature). As such, the assumption that 50% is applied at planting and 50% midway through the growing season on a global scale seems a very crude approximation. Is no better data available? If not, how much do emissions change when these assumptions are varied?

**Reply:** As we have explained, we consider it important to retain the global simulations in the manuscript. Concerning the timing of fertilizer application, we agree that different assumptions could have an effect. The simulations rely on a static crop calendar from GGCMI3 dataset that estimates global planting and harvesting seasons for the major crops. There is no such statistical data for specific practices of fertilization time. We have highlighted this as a further research need in the revised manuscript.

In response to this comment, we have performed additional rounds of simulations to test three possible scenarios, 1) 100 % fertilizer N applied at the beginning of planting season, 2) 75 % N applied at planting and 25 % midway, and 3) 40 % N at planting, 30 % at one third and 30 % at two thirds of the growing season. The global $NH_3$ emissions from synthetic fertilizer use based on the different scenarios were 10.8, 12.9 and 15.8 Tg N yr$^{-1}$, respectively, as compared with the base assumption of 15.0 Tg N yr$^{-1}$ (when applied at 50%:50%). In general this shows that in AMCLIM adding a larger share of fertilizer later in the growing season is associated with increased emission, which can be linked to warmer temperatures as the growing season progresses. However, it should be noted that further testing of this effect would be warranted given the possible effect of tall crop canopies in reducing emissions, which is not addressed in the present version of AMCLIM reported in this study.

The differences in spatial distribution and seasonal variation between different fertilization scenario are shown by Fig. R1-3 and R1-4.

[Figure]

**Figure R1-3. Simulated (a) global NH₃ emissions (Gg N yr⁻¹ grid⁻¹) from synthetic fertilizer use in 2010 using the 50 %, 50 % fertilization scenario, and differences in NH₃ emissions (%) from simulations using (b) 100 % fertilization scenario, (c) 75 %, 25 % fertilization scenario, and (d) 40 %, 30 %, 30 % fertilization scenario.**

[Figure]

**Figure R1-4. Simulated global monthly NH₃ emissions (Gg N month⁻¹) from synthetic fertilizer use in 2010 using the four different fertilization scenarios.**

**Comment:** The lack of model adaption for paddy rice systems means it is likely unreliable for these systems. I would suggest to either adapt the model to paddy rice, or to leave rice out of the global simulation.

**Reply:** On the one hand, there is no data for fertilization of paddy rice systems in GGCMI3 dataset that was used in AMCLIM. On the other hand, we consider simplification is reasonable for global application of the current model version.

Meanwhile, the comparisons in Fig. 12 indicate the simplification provide estimates not out of order of magnitude. We agree future work should include the addressing of individual cropping systems in more detail.

**Comment:** An uncertainty estimate is given in the discussion section, but no details are provided as to how this was calculated. Please provide details so that the reader can judge how seriously to take this estimate.

**Reply:** A systematic estimate of uncertainty associated with a process-based model is very complicated. Therefore, we performed a simple analysis for estimating the uncertainty. For ammonium fertilizer, the number was derived from the simulation of GRAMINAE. For urea application, the uncertainty was estimated based on the multi-site comparison (as shown in Fig. 12). It is worth noting that readers should only interpret the estimates and the uncertainty under the context of modelling. We have modified the manuscript to clarify the uncertainty calculations.

**Comment:** ** Discussion **

I would find it useful to discuss:

How the model differs at a process level from other process-based models such as FAN, DLEM, Daycent or DNDC type models. What are the advantages and disadvantages of the AMCLIM model with respect to these established models and what do the authors see as the future role of the AMCLIM model?

**Reply:** In response to this comment, we have summarised the features of each model, as listed in Table R1-1. Compared with other models, AMCLIM is a dynamical emission model with an emphasis on the $NH_3$ volatilization. AMCLIM shows adequate level of complexity in terms of the soil layering construction, and simulations for N processes in soils, the volatilization simulation and soil pH dynamics. AMCLIM has relatively high temporal resolution, which provides implications in the temporal variations of $NH_3$ fluxes. The highly resolved outputs can be used by atmospheric transport/chemistry models. AMCLIM is considered as a comprehensive emission model rather than a biogeochemical model like the other models shown. However, other advantages reflect in the livestock part of the AMCLIM model, to be published in a forthcoming paper.

**Table R1-1 Comparisons of model features between AMCLIM and other models for $NH_3$ emission simulations. [1]\*DayCent does not have an official version that explicitly include $NH_3$ emissions. The summary is based on Gurung et al. (2021) pointed out by the reviewer. [2]\*DLEM is not an open-source model. We are not able to find a model description paper or use manual of the full DLEM model. The summary is based on a model version DLEM-Bi-$NH_3$ (Xu et al., 2018).**

| Model | Model type | N processes in soils | Soil pH change/dynamics | NH₃ volatilization process | Vegetation interactions at the surface | Temporal resolution |
|---|---|---|---|---|---|---|
| AMCLIM | Dynamical $NH_3$ emission model | Four soil layers up to 28 cm depth; A+B+C+D+E+F+G+H+I | Yes; simple generalised scheme but buffering capacity not considered | M1 | No | Sub-hourly/hourly |
| CAMEO | $NH_3$ Emission module embedded to ORCHIDEE | 11 soil layers for hydrology; A+B+E+F+H+I | No | M1 | No | Sub-hourly, daily, yearly |
| DayCent[1*] | Biogeochemical model | 14 soil layers up to 210 cm depth; A+C+F+H+I | Yes; empirically derived formula with buffering capacity included | M2 | No | Daily |
| DLEM[2*] | Terrestrial ecosystem model | Unknown soil layering; soil N pools are not explicitly simulated but are derived from fertilizer application rate[2*] | No | M1 | Bi-directional exchange scheme | Daily |
| DNDC | Biogeochemical model | Five soil layers up to 50 cm depth; A+B+C+D+F+H+I | Yes; empirically derived formula with buffering capacity included | M2 | No | Daily |
| FANv2 | Process-based N model coupled to CESM | One soil layer 2 cm depth; A+B+C+D+E+F+G+H+K | Yes; pH varies based on age classes | M1 | No | Sub-hourly |

A–mass balance calculation of N pools
B–$NH_3$/$NH_4^+$ equilibrium
C–urea hydrolysis
D–TAN partition
E–surface runoff
F–leaching
G–diffusion in soils
H–nitrification
I–plant N uptake
J–microbial N uptake
K–mechanical N loss
M1–Fluxes are concentration gradient driven and constrained by resistances derived from well-established micrometeorological theory
M2–Empirically derived mass transfer coefficient

**Comment:** How does the calibration procedure compare to these other models, and what are the consequences for the level of confidence we should have in the AMCLIM model results as compared to established models?

**Reply:** For modelling at global scale, we do not find explicit model calibrations from FANv2 (Vira et al., 2020), CAMEO (Beaudor et al., 2023), DNDC (Yang et al., 2022) and DLEM (Xu et al., 2019) at site scale simulations. We think this is largely because global models tend to provide general representation and try to avoid over calibration. The

management practices at the GRAMINAE site were not complicated, which provides a good test situation for the numerical representations of the physical and chemical processes. In contrast, DayCent is widely used for simulating $N_2O$ emissions (not $NH_3$ emissions) and is intensively calibrated using $N_2O$ measurements from fields. The parameters can be quite different between simulations for different places, e.g., US vs. Switzerland. There are limited studies of applying DayCent at global scale, and we did not find explicit model calibration in the global application of DayCent (e.g., De Grosso et al., 2009).

The global estimates by AMCLIM are broadly consistent with existing models (as shown in Table 2). Combining with the generally close agreement with the GRAMINAE measurements, the AMCLIM model is considered to be robust and capable in estimating agricultural $NH_3$ emissions.

**Comment:** Why has urease inhibition not been considered? This is required by many countries when broadcast spreading urea fertiliser and has important consequences for ammonia emissions. To what extent does this limit the usefulness of the model?

**Reply:** There is no statistical data of the application of urease inhibitor that can be used in the model. To our knowledge, only Germany has regulations on the use of urease inhibitor (or incorporation is required). The use of urea inhibitor can be incorporated into the AMCLIM in the future work, once there is sufficient data.

**Reference**

Beaudor, M., Vuichard, N., Lathière, J., Evangeliou, N., Van Damme, M., Clarisse, L., and Hauglustaine, D.: Global agricultural ammonia emissions simulated with the ORCHIDEE land surface model, Geosci. Model Dev., 16, 1053–1081, https://doi.org/10.5194/gmd-16-1053-2023, 2023.

Butterbach-Bahl, K., Gundersen, P., Ambus, P., Augustin, J., Beier, C., Boeckx, P., Dannenmann, M., Gimeno, B. S., Ibrom, A., Kiese, R., Kitzler, B., Rees, R. M., Smith, K. A., Stevens, C., Vesala, T., and Zechmeister-Boltenstern, S.: Nitrogen processes in terrestrial ecosystems, in: The European Nitrogen Assessment, vol. Chapter 6, Cambridge University Press, 99– 125, 2011

Chantigny, M. H., Rochette, P., Angers, D. A., Massé, D., and Côté, D.: Ammonia Volatilization and Selected Soil Characteristics Following Application of Anaerobically Digested Pig Slurry, Soil Sci. Soc. Am. J., 68, 306–312, https://doi.org/10.2136/sssaj2004.3060, 2004

Del Grosso, S. J., Ojima, D. S., Parton, W. J., Stehfest, E., Heistemann, M., DeAngelo, B.and Rose, S.: Global scale DAYCENT model analysis of greenhouse gas emissions and mitigation strategies for cropped soils, 67, https://doi.org/10.1016/J.GLOPLACHA.2008.12.006, 2009.

Gurung, R., Ogle, S. M., Breidt, F. J., Williams, S. A., Zhang, Y., Del Grosso, S. J., Del Grosso, S. J., Parton, W. J.and Paustian, K.: Modeling ammonia volatilization from urea application to agricultural soils in the DayCent model, 119, https://doi.org/10.1007/S10705-021-10122-Z, 2021.

Kamp, J. N., Hafner, S. D., Huijsmans, J., van Boheemen, K., Götze, H., Pacholski, A., & Pedersen, J.: Comparison of two micrometeorological and three enclosure methods for measuring ammonia emission after slurry application in two field experiments, Agric. For. Meteorol., 354, 110077, https://doi.org/10.1016/j.agrformet.2024.110077, 2024.

Móring, A., Vieno, M., Doherty, R. M., Laubach, J., Taghizadeh-Toosi, A., and Sutton, M. A.: A process-based model for ammonia emission from urine patches, GAG (Generation of Ammonia from Grazing): description and sensitivity analysis, Biogeosciences, 13, 1837–1861, https://doi.org/10.5194/bg-13-1837-2016, 2016.

Snyder, C. S., Bruulsema, T. W., Jensen, T. L.and Fixen, P. E.: Review of greenhouse gas emissions from crop production systems and fertilizer management effects, 133, https://doi.org/10.1016/J.AGEE.2009.04.021, 2009.

Thornley, J. H. M.: A Transport-resistance Model of Forest Growth and Partitioning, Annals of Botany, 68, 211–226, https://doi.org/10.1093/oxfordjournals.aob.a088246, 1991.

Vira, J., Hess, P., Melkonian, J., and Wieder, W. R.: An improved mechanistic model for ammonia volatilization in Earth system models: Flow of Agricultural Nitrogen version 2 (FANv2), Geosci. Model Dev., 13, 4459–4490, https://doi.org/10.5194/gmd-13-4459-2020, 2020

Xu, R., Tian, H., Pan, S., Prior, S. A., Feng, Y., Batchelor, W. D., Chen, J., and Yang, J.: Global ammonia emissions from synthetic nitrogen fertilizer applications in agricultural systems: Empirical and process-based estimates and uncertainty, Glob Change Biol, 25, 314–326, https://doi.org/10.1111/gcb.14499, 2019.

Yang, Y., Liu, L., Bai, Z., Xu, W., Zhang, F., Zhang, X., Liu, X., and Xie, Y.: Comprehensive quantification of global cropland ammonia emissions and potential abatement, Science of The Total Environment, 812, 151450, https://doi.org/10.1016/j.scitotenv.2021.151450, 2022

---

## Author Comment (AC2)

**Response to Anonymous Referee #2**
**Overall response:** We would like to thank reviewer 2 for these invaluable comments and detailed checking of the equations. These help improve the manuscript. Here we outline the point-by-point responses below in blue, and the relevant figures are attached.

A dynamical process-based model AMmonia–CLIMate v1.0

(AMCLIM v1.0) for quantifying global agricultural ammonia

emissions – Part 1: Land module for simulating emissions from synthetic fertilizer use by Jiang et al.

**Comment:** This paper describes the ammonia emissions from synthetic fertilizer. First of all I want to compliment you with this article. It is nicely written and the results and methods are clear. However I also have some concerns.

**Reply:** We thank the reviewer for these kind words.

**Comment:** 1.   There is a nice description of the validation/calibration of the model on the GRAMINAE database. These are all observations on fertilized grassland. After that the application of the model is on a global scale with 16 crops. But none of these crops represent grassland. Is it reasonable to assume that grassland is a good representative for all 16 crops?

**Reply:** AMCLIM is developed based on the understanding at process level. The management practices at the GRAMINAE site were not complicated, which provides a suitable test context for the numerical representations of the physical and chemical processes. On the one hand, the model results show close agreement with the GRAMINAE measurements (Fig. 5), and the multi-site comparison demonstrates that the AMCLIM model have reasonable estimates for various crops under different climatic and soil conditions (Fig. 12). On the other hand, we found that the critical factor affects $NH_3$ emissions is the timing of fertilization and amount of fertilizer applied under current model settings (See replied and new figures Fig. R1-3, Fig. R1-4 in response to #Reviewer 1). Therefore, we think the processes included in AMCLIM are robust and representative for simulations for synthetic fertilizer use.

**Comment:** 2.   And from the other perspective. The GRAMINAE database shows that fertilizer is used on grassland (which is common practice in for example Europe and US). None of this fertilizer is mentioned in the global estimate of ammonia? What is the

role of grassland in the global emission? Can you elaborate on this uncertainty (of not taking this load)?

**Reply:** In AMCLIM, only $NH_3$ emissions from grazed grassland were simulated, which will be described by the second part of the model in a forthcoming paper (for the livestock sector). Fertilized grasslands (with synthetic fertilizers) were not included/simulated because there is no data specified for this type of crop in the dataset we used. This is a significant gap, which we highlight in the revised manuscript as needing further work. We understand that globally the majority of synthetic fertilizer was used for croplands rather than grasslands (although in Europe and some other locations grasslands can receive significant amount of fertilizers). The total applied N from synthetic fertilizer was 102.3 Tg N yr$^{-1}$ in AMCLIM, which is comparable to the 99.6 Tg N yr$^{-1}$ of consumed fertilizer suggested by the International Fertilizer Association. More details are given by the reply to the reviewer's next comment.

**Comment:** 3.   I have the impression that the global figures of the ammonia emissions (for example figure 6, but actually all maps) have a coverage of grassland and cropland. So I am wondering which land use database is used and whether the assumption is valid that all grassland is fertilized. Perhaps I am wrong, but I would expect that maps have more white area. Can you please elaborate on this as well?

**Reply:** The areas of croplands used in AMCLIM were from the Farming the Planet 2 (FTP2) dataset (Monfreda et al., 2008). The data for fertilization rates and crop calendars were from Global Gridded Crop Model Intercomparison Phase 3 (GGCMI3). However, fertilized grasslands were not included in the GGCMI3 datasets. We have now modified the manuscript to clarify this point.

Specific comments

**Comment:** Line 84: AMCLIM-Fertilizer is nowhere else mentioned. I was sometimes confused whether it should be AMCLIM or AMCLIM-Land or it should be AMCLIM-Fertilizer. Please check the document to verify that the right name is used. I think I would prefer to use AMCLIM-Fertilizer in most cases (as suggestion).

**Reply:** We thank the reviewer for pointing out the unclear naming in the manuscript. There are three modules in AMCLIM, namely 1) AMCLIM–Housing, 2) AMCLIM–MMS and 3) AMCLIM–Land. AMCLIM–Land is described in this study and was used to simulate $NH_3$ emissions from synthetic fertilizer use. We have now removed "AMCLIM–Fertilizer" and used "Fertilizer simulations" to avoid confusion.

**Comment:** Line 85: "AMCLIM Livestock". Not clear to me. Not mentioned in Figure 1 and not mentioned in line 92. But the contents of this must be very clear, because this determines whether a topic is described here or in the other article.

**Reply:** We removed "AMCLIM Livestock" and use "Livestock simulation" to avoid confusion.

**Comment:** Figure 1: "chemical" fertilizers is used, but mostly in the text "synthetic" is used.

**Reply:** We updated Fig. 1 and Fig. 2 to have all "chemical fertilizer" changed to "synthetic fertilizer".

**Comment:** Figure 2, line 122: Why is ammonification not included? This is input TAN.

**Reply:** The ammonification of nitrate (or DNRA) was not simulated in AMCLIM, while the decomposition of organic nitrogen (or mineralization) was included in the model but only for the livestock simulations. We have now clarified this in the revised manuscript.

**Comment:** Line 158: I think here m denotes mass. But in line 152 it is meter..... Please change to make it clear.

**Reply:** The unit of $V_{\text{H2O}}$ is millilitre per unit area.

**Comment:** Line 165: eq 3: Explain in the text the names in the right hand side of the equation (so s, aq and g are not explained).

**Reply:** The abbreviations s, aq and g represent solid phase, aqueous phase and gaseous phase of a substance, respectively. We improved the text as the follows:

"The most important aggregated N species simulated in AMCLIM is total ammoniacal nitrogen (TAN = $NH_3$+ $NH_4^+$), which can either be partitioned into gaseous $NH_3$ ($M_{NH_3,g}$) aqueous TAN ($M_{\text{TAN,aq}}$) or adsorbed $NH_4^+$ ($M_{NH_4^+,s}$), as shown in Eq. (3):

$$M_{\text{TAN}} = M_{NH_3,g} + M_{\text{TAN,aq}} + M_{NH_4^+,s} \ . \qquad\qquad (3)"$$

**Comment:** Line 168: Where is H+ coming from? Can you elaborate on this?

**Reply:** The H⁺ was given by Eq. 9 and Eq. 10. The emission potential $\Gamma$ is defined as $[NH_4^+]/[H^+]$. We moved sentence to from line 168 to follow Eq. 9 and Eq. 10 to improve clearness.

**Comment:** Line 168: Reference format of Sutton
**Reply:** We corrected the citation format.

**Comment:** Line 187 – 190: Here square brackets are used. But why? Is [NH3(g)] not the same as M_nh3,g as mentioned in eq. 3? Please make clear what your intention is here.

**Reply:** Square brackets are used to represent the concentrations. For example, $[NH_3(g)]$ is the gaseous concentration of $NH_3$, and $[NH_4^+(aq)]$ is the aqueous concentration of $NH_4^+$. By comparison, M denotes the mass of the species, i.e., $M_{NH_3,g}$ is the mass of gaseous $NH_3$.

**Comment:** Line 190: Why NH3(g) instead of TAN(g)?

**Reply:** The total ammoniacal nitrogen is represented by TAN, which is the aggregate of ammonia ($NH_3$) and ammonium ($NH_4^+$). Therefore, the gas phase should only include ammonia gas $NH_3(g)$.

**Comment:** Line 210: No explanation of constants or parameters is given in the text.

**Reply:** These constants are derived from the Henry's Law constant and dissociation constant, which have been fully explained in Sutton et al. (1994).

**Comment:** Line 303: "applied to cropland" What about grassland?

**Reply:** As mentioned, fertilized grasslands (by synthetic fertilizer) were not simulated and included in the global upscaling of AMCLIM model due to lack of global estimates of the distribution of fertilizer to grasslands.

**Comment:** Line 575: explain MAM. SON and DJF. I see they are explained in the caption of figure 9, but this text is before the figure.

**Reply:** We improved the text as the follows.

"The seasonal emissions in both years are similar, with over 50 % of $NH_3$ occurring in the Northern Hemisphere (NH) summer months and about 25 % in March-April-May (MAM). September-October-November (SON) and December-January-February (DJF) both contribute slightly over 10 % of the annual emissions. In the NH, more than 70 %

of annual emissions are from June-July-August (JJA), while emissions in SON and DJF are significant in the Southern Hemisphere (SH)."

**Comment:** Figure 9: end of caption is text missing

**Reply:** We corrected the caption as the follows.

"Percentage of annual emissions in the season of (b) MAM, (d) JJA, (f) SON and (h) DJF."

**Comment:** Figure 9: Caption says something about percentage, but figure shows fractions. I would suggest to use everywhere the Pv (%).

**Reply:** We changed the caption to "Figure 9. Seasonal NH3 emissions (Gg N grid$^{-1}$) from ammonium and urea fertilizer application and the relative fraction of annual emissions that are from the corresponding season ($f_{season}$) in 2010 simulated by AMCLIM–Land." We would like to use a different symbol to represent the contributions from each season.

**Comment:** Figure 9. Add per grid cell to the unit.

**Reply:** We changed the caption to "Figure 9. Seasonal NH3 emissions (Gg N grid$^{-1}$) from ammonium and urea fertilizer application and the relative fraction of annual emissions that are from the corresponding season ($f_{season}$) in 2010 simulated by AMCLIM–Land."

**Comment:** Figure 9: The y axes of NH3 per grid is confusing. Looks like it was for the right column. Can you put this on the left hand side of the figure or make it more clear that it belongs to the left column? This remark is for the maps figures.

**Reply:** We addressed this point by clarifying the unit in the figure caption. The position of colour map label is more of a default setting in the data visualization.

**Comment:** Line 599: End of caption misses text.

**Reply:** We corrected the caption to "Figure 10. Global monthly NH3 emissions (Gg N month$^{-1}$) from ammonium and urea fertilizer applications for 16 major crops in 2010 simulated by AMCLIM–Land."

**Comment:** Line 623: In figure a small f is used in F_region.

**Reply:** We corrected the caption to "Figure 11. Monthly NH$_3$ emissions from ammonium and urea fertilizer application in different regions of the world and the

relative fraction of the global monthly emissions that are from the corresponding regions ($f_{region}$). Annual total NH3 emissions of the region are given at the top right corner of each plot, with the percentage of emissions from this region. The figure is for 2010."

**Comment:** Line 628: AMLIM -> AMCLIM

**Reply:** We corrected the typo.

**Comment:** Line 810: Assumptions -> assumptions

**Reply:** We corrected the typo.

**Comment:** Figure A4: same remarks as for Figure 9.

**Reply:** We corrected the caption as for Fig. 9.

**Comment:** Caption figure A7: Mention unit in Gg N per grid cell.

**Reply:** We corrected the caption.

Supplementary information

**Comment:** Eq SM 2 and 3: in the main text (line 184: "In addition, diffusive and drainage fluxes
considered as losses in the soil layer above become sources of nitrogen for the layer underneath."). I don't see this in these equations.

**Reply:** These fluxes are considered to be included by $I_{TAN}$ in the original manuscript. We updated the equations to make it explicit.

**Comment:** Line 35: Unit of K_d?

**Reply:** The unit of $K_d$ is $m^3 m^{-3}$.

**Comment:** Line 36: "fractional soil clay content": Is this determined per grid cell. So here it is the upper soil layer? From what is this a fraction?

**Reply:** The fraction of clay, sand and silt (soil texture) of each grid is from the Regridded Harmonized World Soil Database (HWSD) v1.2 (Wieder et al., 2014).

**Comment:** Line 44: K_Knitrif,opt -> K_nitrif,opt

**Reply:** We corrected the symbol.

**Comment:** Line 45: Is the unit of K_nitrif,opt in percentage??

**Reply:** The unit is percentage per time.

**Comment:** Line 49: small t was reserved for time, but now it is temperature. Make the T (T_opt, T_max)

**Reply:** We updated the symbols.

**Comment:** SM8: change k_nitrif,T -> K_nitrif,T

**Reply:** We updated the symbol.

**Comment:** In line 50, K is used for Kelvin (correct)   Perhaps it is an idea to change all K variable into small k variables (also in main text) to avoid confusing.

**Reply:** We propose to include a table of all model parameters and variables in the revised manuscript.

**Comment:** Line 82: Unit of J_C,N?

**Reply:** It is a dimensionless parameter.

**Comment:** Lines 87-88: What is the unit of 4 and 40?

**Reply:** The units are g C m$^{-2}$ and g N m$^{-2}$, respectively, as specified in line 85. We updated the units to make it clear.

**Comment:** Equation SM17: use the alfa_root and J_C,N in this formula.

**Reply:** We updated Eq. SM17.

**Comment:** Line 92 -99: "There are four ….. [end of table]" I would move this under equation SM14. Now it is coming too late.

**Reply:** We moved this under Eq. SM14.

**Comment:** Line 102: unit of W_uptake?

**Reply:** The unit is m s$^{-1}$. We added the unit.

**Comment:** Line 110: remove one of the closing brackets

**Reply:** We removed one closing bracket.

**Comment:** Line 114: What is 20.1 and 14.9?

**Reply:** These are the atomic diffusion volumes of air and NH$_3$, respectively.

**Comment:** Line 114: unit of pressure?

**Reply:** The unit is Pa. We added in the manuscript.

**Comment:** I stopped here. Please check whether the units of all parameters are given and whether they are explained in the text.

**Reply:** We would like to thank the reviewer again for these useful comments.

**Reference**

Monfreda, C., Ramankutty, N., and Foley, J. A.: Farming the planet: 2. Geographic distribution of crop areas, yields, 1105 physiological types, and net primary production in the year 2000: GLOBAL CROP AREAS AND YIELDS IN 2000, Global Biogeochem. Cycles, 22, n/a-n/a, https://doi.org/10.1029/2007GB002947, 2008.

Sutton, M. A., Asman, W. A. H., and Schørring, J. K.: Dry deposition of reduced nitrogen, Tellus B, 46, 255–273, 1994.

Wieder, W. R., Boehnert, J., and Bonan, G. B.: Evaluating soil biogeochemistry parameterizations in Earth system models with observations: Soil Biogeochemistry in ESMs, Global Biogeochem. Cycles, 28, 211–222, https://doi.org/10.1002/2013GB004665, 2014.

---

## Author Response (AR2)

**Response to Anonymous Referee #1**

**Comment:**

I find the manuscript considerably improved and am happy to recommend that it is published without additional changes.

Previously I stated that I was unconvinced about the validity of performing a global upscaling. While I remain somewhat skeptical, I think that the methods and shortcomings have now been well enough described that readers can make up their own minds in an informed way on how much to trust the global results. On the global scale I believe it would be useful to perform a more rigorous uncertainty analysis, but I understand this is a big task and am content if it is postponed to a future publication.

**Reply:** We appreciate that the reviewer recognizes the improvement of the revised manuscript. We thank the reviewer for all the insightful and critical comments/suggestions in the review.

**Response to Anonymous Referee #2**

**Comment:**

First of all I want to thank the authors for their extensive reply on the concerns and remarks of both reviewers. I think they addressed a lot of issues. I also think the manuscript has improved a lot and also recognized some of the reviewers concerns. I also noticed that reviewer #1 has a lot of critical and good notes and that the question whether this is the good time to apply this model on a global scale. For what I have seen in this manuscript, the authors did everything to calibrate and validate the model on as much observations they could use. They tried to address the spatial distribution in climate zones, as well as different fertilizer application moments. Additional information to improve their model results are not yet available, so therefore, I think, this lightweight model is useful at a global scale. Like any other model, there are still improvements possible, when useful data or observations are available.

**Reply:** We thank the reviewer for recognizing the significance of our work. We thank the reviewer for all the constructive suggestions and detailed comments in the review.

Other comments:
I am looking at figure R1-2. There are simulations where the internal time step has been varied between 1 minute and 1 day. I think it is a pity that this figure is not added to the final manuscript (in the Appendix). Could that be done?

**Reply:** Fig. R1-2 (in Author's response) has been added to Sect.S10 in the Supplementary materials. This is an independent section, and there is an accompanying paragraph discussing the model sensitivity to the temporal resolution.

I also want to make a suggestion (probably not for this version of the model). When I see a lot of peaks which should be reproduced by the model, I always consider to use a dynamic time step method for the Euler method (for example: Adaptive Runge–Kutta methods on https://en.wikipedia.org/wiki/Runge%E2%80%93Kutta_methods). In case a lot of change is happening in the system, small time steps are taken, and when the change is limited, large time steps are taken. The largest time step taken, is dependent on the reporting time (times that the model write results to output files). This avoids missing any kind of peaks and it does not always need more computing time. This is a suggestion.

**Reply:** We thank the reviewer for pointing out this method/technique. We think this dynamic time-step method can be very useful to address the challenge that higher temporal resolution would require much more computational resources, especially for large-scale simulations. We added this point to the discussion of future work in Sect.S10, together with Fig. R1-2.

The last point from reviewer #1 about the details of the uncertainty estimates. I see that new runs are performed to give more insights in the uncertainty. I appreciate that. But the reviewer is asking to provide more details. I didn't see that in the revised manuscript. I see in the revised manuscript that there are two main aspects of uncertainty: input data and model parameters. I did find the input data. I am missing more information about the basic assumption of the one to four points that are described. There is a conclusion of 33% and 20% uncertainty. But still it is not clear what you assumed and what parameters are changed to reach this conclusion. Can you provide this information?

**Reply:** We improved Sect.4.3 to clarify the assumptions of the summarised points in the model.

As discussed in the manuscript, systematically quantifying model uncertainty, especially on the global scale, is challenging and difficult because we only have very limited datasets for explicit model-measurement comparisons. The value of 33 % was derived from the comparison with the GRAMINAE experiments, indicating the reduction in the cumulative modelled $NH_3$ emission to match the measured emission. The value of 20 % came from the multi-site model-measurement. Given that chamber measurements often estimate lower emissions, and global simulations can show both overestimates and underestimates, a rather conservative estimate of 20 % uncertainty is proposed. In contrast to systematic analyses, these two values serve as an approximation or "back-of-the-envelope" calculations for the uncertainty estimation to give readers a useful point of "reference". We have emphasized that "It is worth noting that readers should only interpret the estimates and the uncertainty under the context of modelling" as highlighted during the previous round of review.

I liked the inclusion of the list of parameters needed to be used in this modelling exercise. The question is whether this should be in the main text or in the Appendix. I think the bullet points could be moved to the Appendix. Feel free to keep it or move it.

**Reply:** We would like to raise a point that high quality measurements are urgently needed and can be valuable for model development and evaluation. Therefore, we would like to have a list of suggested measurements in the main text.